# Transfer Learning for High-dimensional Quantile Regression with Statistical Guarantee

**Sheng Qiao**  *sqiao@ucsd.edu*
*Department of Mathematics*
*University of California San Diego*
*San Diego, CA 92093, USA*

**Yong He**  *heyong@sdu.edu.cn*
*Institute for Financial Studies*
*Shandong University*
*Jinan, 250100, China*

**Wen-Xin Zhou**  *wenxinz@uic.edu*
*Department of Information and Decision Sciences*
*University of Illinois at Chicago*
*Chicago, IL 60607, USA*

**Reviewed on OpenReview:** *https://openreview.net/forum?id=d3xwrfAG4V*

## Abstract

The task of transfer learning is to improve estimation/inference of a target model by migrating data from closely related source populations. In this article, we propose transfer learning algorithms for high-dimensional Quantile Regression (QR) models with the technique of convolution-type smoothing. Given the transferable source populations, we derive $\ell_1/\ell_2$-estimation error bounds for the estimators of the target regression coefficients under mild conditions. Theoretical analysis shows that the upper bounds are improved over those of the classical penalized QR estimator with only the target data, as long as the target and the sources are sufficiently similar to each other. When the set of informative sources is unknown, a transferable source detection algorithm is proposed to detect informative sources from all available sources. Thorough simulation studies justify our theoretical analysis.

## 1 Introduction

Transfer learning (Torrey & Shavlik, 2010) has been growing popular and drawing increasing attention in machine learning, which achieves great success in a wide range of real applications with limited available training data. Transfer learning aims to transfer knowledge from related source tasks/domains to enhance the learning or performance of the target task/domain, which typically involves two main subproblems. First, some criteria should be come up with to quantify the relatedness/similarity among target and source tasks. Intuitively, a high similarity would enhance the performance, while a low similarity would be harmful for the target task, which is known as "negative transfer" in the literature (Torrey & Shavlik, 2010). Second, a transfer procedure should be carefully designed to transfer the "critical" knowledge from source domains, just like the human intelligence of leveraging prior experiences to tackle novel problems. A well designed transfer algorithm should not only identify the positive transfer sources thereby enlarging their impact, but also avoid the negative transfer in any case. All in all, transfer learning has become an active and promising research area, and substantial contributions has also been made recently to the theoretical guarantee for transfer learning in both supervised, semi-supervised, and unsupervised settings, see for example the context of classification by Cai & Wei (2021); Reeve et al. (2021), high-dimensional (generalized) linear regression by Li et al. (2022); Tian & Feng (2023); Lin & Li (2022), graphical model by Li et al. (2023); He et al. (2022).

As far as we know, there exist no work on transfer learning for quantile regression and we aim to fill this gap in this paper.

**Comparison with the existing work and our contribution**

A few works explore transfer learning under the high-dimensional setting. Bastani (2021) studied the transfer learning problem under a high-dimensional generalized linear models (GLM) with one single known transferable source data and the dimensionality $p$ is assume to be larger than the sample size of the target dataset $n_{\text{target}}$ while smaller than that of the source dataset $n_{\text{source}}$. A two-step transfer learning algorithm was developed and the $\ell_1$-estimation error bound was derived when the difference between the target and source coefficient is $\ell_0$-sparse. More specifically, their estimator requires $n_{\text{target}} = \mathcal{O}(s^2 \log^2(p/\xi)/\xi^2)$ as long as $n_{\text{source}} \gtrsim \mathcal{O}(s^2 p^2 \log^2(p/\xi)/\xi^2)$, where $\xi$ denotes a parameter which is less than the $\ell_1$-norm of the difference vector between the coefficients of the target and source, $p$ is the number of features, and $s$ is the sparsity of the difference in coefficients between the target and source. Li et al. (2022) studied the high-dimensional linear regression problem under some weaker assumptions, where both target and source samples are high-dimensional. Multiple source datasets are available and the transferable set may even be unknown in their paper. With $\ell_q$-sparse difference vector between the coefficients of the target and source for $q \in [0, 1)$ and $\ell_0$-sparse target parameter, the $\ell_2$-estimation error bound was derived and proved to be minimax optimal under some conditions. In the setting where the transferable set is unknown, a source detection algorithm was proposed to consistently select the informative sources. Tian & Feng (2023) further investigated multi-source transfer learning on high-dimensional generalized linear models (GLM). They assumed both target and source data to be high-dimensional and the disparity in coefficients between the target and source to be $\ell_1$-sparse. Given the informative sources to transfer, the $\ell_1/\ell_2$-estimation error was derived and proved to be minimax optimal under mild conditions. Tian & Feng (2023) also established a transferable source detection algorithm to identify the informative sources. In addition, they constructed the corresponding confidence interval for individual regression parameter. Li et al. (2021) proposed a federated transfer learning approach to consolidate data from different populations and from multiple medical associations. The target and source data are both high-dimensional in their discussion and they characterized the vector of disparities between the target and source parameters to be $\ell_0$-sparse. Compared with Tian & Feng (2023), their approach achieves a faster convergence rate under some conditions and has weaker requirements on the level of heterogeneity for data from diverse populations.

Inspired by the two-step algorithm in Bastani (2021), Li et al. (2022) and Tian & Feng (2023), we propose a multi-source transfer learning method under high-dimensional quantile regression. To overcome the non-smoothness and non-convexity of the quantile loss, motivated by He et al. (2021) and Tan et al. (2022), we employ the convolution-type smoothed quantile regression. With the help convolution smoothing, Tan et al. (2022) proposed a gradient-based algorithm that is more scalable to large-scale problems with either large sample size or high dimensionality compared with other methods for fitting high-dimensional quantile regression. Assuming the difference vector between the target and each source coefficients to be $\ell_1$-sparse, we establish the $\ell_1/\ell_2$-estimation error bounds that are proved to be sharper than the bounds of the classical $\ell_1$-penalized quantile regression (Belloni & Chernozhukov, 2011) under some conditions.

In this paper, we propose transfer learning algorithms for quantile regression with high-dimensional data and we assume the difference between target and source coefficients to be $\ell_0$-sparse or $\ell_1$-sparse. In the setting where the sources are sufficiently close to the target, our theoretical analysis and simulation results show that the estimation error bound of the target coefficients is improved compared to the classical $\ell_1$-penalized quantile regression model (Belloni & Chernozhukov, 2011) using only the target data under mild conditions. To overcome the lack of smoothness and convexity of the check loss, we employed the convolution-type smoothed quantile regression and analyzed the (local) restricted strong convexity of the empirical smoothed quantile loss functions in the transferring and debiasing steps. We also extended the source detection algorithm in Tian & Feng (2023) to the quantile regression setting. Simulation results show that the algorithm works well in discovering useful sources. In contrast to the case with $\ell_1$-sparse difference vector between the target and source coefficients, the algorithm with $\ell_0$-sparse one learns the source coefficients independently, which greatly reduces the communications cost across different sources. Furthermore, the algorithm with $\ell_0$-sparse difference vector has fewer assumptions on the level of heterogeneity for data from different sources.

The most related work is a concurrent paper by Zhang & Zhu (2022), which also considered the smoothed quantile regression models under transfer learning framework. They proposed a smoothed two-step transfer learning algorithm as well as a new source detection method based on the $K$-means clustering algorithm, which does not need the input of a threshold in contrast to the source detection algorithm in Tian & Feng (2023). In addition, they further extended their work to the distributed quantile regression and model averaging setup. However, compared with Zhang & Zhu (2022), our work doesn't require the restrictive conditions on the kernels that $\sup_{|h|\leq 1} K(u/h)/h < M_k$ almost everywhere in $u$. In addition, given that the disparity vector is characterized in $\ell_0$-norm instead of $\ell_1$-norm, we introduce an algorithm which is motivated from Li et al. (2022) and Li et al. (2021). The $\ell_1/\ell_2$-estimation error bounds are also established and proved to be sharper than the bounds of the classical $\ell_1$-penalized quantile regression (Belloni & Chernozhukov, 2011) under some mild conditions.

Before ending this section, we introduce the notations used throughout the paper. For every integer $k \geq 1$, we use $\mathbb{R}^k$ to denote the $k$-dimensional Euclidean space, and write $[k] = \{1,\ldots,k\}$. For $k \geq 2$, $\mathbb{S}^{k-1} = \{\boldsymbol{u} \in \mathbb{R}^k : ||\boldsymbol{u}||_2 = 1\}$ denotes the unit sphere in $\mathbb{R}^k$. For any symmetric, positive semidefinite matrix $\boldsymbol{A} \in \mathbb{R}^{k\times k}$, if its vector of eigenvalues is denoted by $\gamma(\boldsymbol{A})$ and ordered as $\gamma_1(\boldsymbol{A}) \geq,\ldots,\geq \gamma_p(\boldsymbol{A}) \geq 0$, the operator norm of $\boldsymbol{A}$ is $||\boldsymbol{A}||_2 = \gamma_1(\boldsymbol{A})$. Moreover, the vector norm induced by $\boldsymbol{A}$ is $||\boldsymbol{u}||_A = ||\boldsymbol{A}^{1/2}\boldsymbol{u}||_2$ for any $\boldsymbol{u} \in \mathbb{R}^k$. For any real numbers $s$ and $t$, $s \vee t$ denotes $\max(s,t)$ and $s \wedge t$ denotes $\min(s,t)$. For two sequences $\{a_n\}_{n\geq 1}$ and $\{b_n\}_{n\geq 1}$, which consist of non-negative numbers, $a_n \lesssim b_n$ means that there exists a constant $C > 0$ such that $a_n \leq Cb_n$. $a_n \asymp b_n$ is equivalent to $a_n \lesssim b_n$ and $b_n \lesssim a_n$. For $r, l > 0$, define the $\ell_2$-ball and $\ell_1$-cone as

$$\mathbb{B}_A(r) = \{\boldsymbol{\delta} \in \mathbb{R}^p : ||\boldsymbol{\delta}||_A \leq r\} \text{ and } \mathbb{C}_A(l) = \{\boldsymbol{\delta} \in \mathbb{R}^p : ||\boldsymbol{\delta}||_1 \leq l||\boldsymbol{\delta}||_A\}.$$

## 2 Methodology

### 2.1 Problem setup

Given the predictors $\boldsymbol{x} \in \mathbb{R}^p$ and a scalar response variable $y \in \mathbb{R}$, the $\tau$-th conditional quantile functions of $y$ given $\boldsymbol{x}$ is written as

$$F_{y|x}^{-1}(\tau) = \inf\{y : F_{y|x}(y) \geq \tau\},$$

where $F_{y|x}(\cdot)$ is the conditional distribution function of $y$ given $\boldsymbol{x}$. Consider the following linear quantile regression model at a given $\tau \in (0,1)$:

$$F_{y|x}^{-1}(\tau) = \boldsymbol{x}^{\mathrm{T}}\boldsymbol{\beta}^*(\tau),$$

where $\boldsymbol{\beta}^*(\tau) = (\beta_1^*(\tau),\ldots,\beta_p^*(\tau))^{\mathrm{T}} \in \mathbb{R}^p$ is the true quantile regression coefficient.

Let $\{(y_i,\boldsymbol{x}_i)\}_{i=1}^n$ be a random sample from $(y,\boldsymbol{x})$. The preceding model assumption is equivalent to the following model

$$y_i = \boldsymbol{x}_i^{\mathrm{T}}\boldsymbol{\beta}^* + \epsilon_i \text{ and } \mathbb{P}(\epsilon_i \leq 0|\boldsymbol{x}_i) = \tau.$$

The $\ell_1$-penalized quantile regression estimator (Belloni & Chernozhukov, 2011) is generally defined as one of the solution to the optimization problem

$$\underset{\boldsymbol{\beta}=(\beta_1,\ldots,\beta_p)^{\mathrm{T}}\in\mathbb{R}^p}{\text{minimize}} \left\{ \underbrace{\frac{1}{n}\sum_{i=1}^n \rho_\tau(y_i - \boldsymbol{x}_i^{\mathrm{T}}\boldsymbol{\beta})}_{=:\hat{Q}(\boldsymbol{\beta})} + \lambda||\boldsymbol{\beta}||_1 \right\}, \tag{1}$$

where $\rho_\tau(u)$ is defined as $\rho_\tau(u) = u\{\tau - \mathbb{1}(u < 0)\}$, also referred to as the $\tau$-quantile check loss function. Let $\hat{F}(\cdot;\boldsymbol{\beta})$ be the empirical cumulative distribution function of the residuals $\{r_i(\boldsymbol{\beta}) := y_i - \boldsymbol{x}^{\mathrm{T}}\boldsymbol{\beta}\}_{i=1}^n$, i.e., $\hat{F}(u;\boldsymbol{\beta}) = (1/n)\sum_{i=1}^n \mathbb{1}\{r_i(\boldsymbol{\beta}) \leq u\}$ for any $u \in \mathbb{R}$. Then the empirical quantile loss $\hat{Q}(\boldsymbol{\beta})$ in (1) can be written as

$$\hat{Q}(\boldsymbol{\beta}) = \int_{-\infty}^\infty \rho_\tau(u)d\hat{F}(u;\boldsymbol{\beta}). \tag{2}$$

As the empirical cumulative distribution function $\hat{F}(\cdot;\boldsymbol{\beta})$ is discontinuous, the empirical quantile loss is non-differentiable, which brings great challenges to both computation and statistical theory establishment. The

kernel smoothing method (Horowitz, 1998) is commonly utilized to tackle this issue. However, the smoothed loss is still non-convex, thereby we further consider the convolution-type smoothed quantile loss function, which is not only convex but also differentiable and brings great convenience in terms of both computation and theoretical analysis. In the following, we briefly introduce the convolution-type smoothed quantile loss function, which was firstly introduced by Tan et al. (2022).

Let $K(\cdot)$ be a non-negative kernel function that is symmetric around 0 and integrates to 1, and $h > 0$ be a bandwidth. That is

$$K_h(u) = (1/h)K(u/h), \ \bar{K}(u) = \int_{-\infty}^{u} K(v)dv \text{ and } \bar{K}_h(u) = \bar{K}(u/h), \ u \in \mathbb{R}.$$

The empirical smoothed loss function can be defined as

$$\hat{Q}_h(\boldsymbol{\beta}) = \frac{1}{n}\sum_{i=1}^{n} l_h(y_i - \boldsymbol{x}_i^{\mathrm{T}}\boldsymbol{\beta}) \text{ with } l_h(u) = (\rho_\tau * K_h)(u) = \int_{-\infty}^{\infty} \rho_\tau(v)K_h(v - u)dv,$$

where $*$ denotes the convolution operator. Therefore, the $\ell_1$-penalized convolution smoothed estimator is given by

$$\hat{\boldsymbol{\beta}} \in \arg\min_{\boldsymbol{\beta} \in \mathbb{R}^p} \left\{ \hat{Q}_h(\boldsymbol{\beta}) + \lambda||\boldsymbol{\beta}||_1 \right\},$$

where the smoothing bandwidth $h$ adapts to the sample size $n$ and the dimension $p$ while $\hat{\boldsymbol{\beta}}$ depends on the quantile index $\tau$, bandwidth $h$, and penalty level $\lambda$.

**Remark 2.1.** To better understand this smoothing mechanism, we compute the smoothed loss $l_h(u)$ explicitly for several widely used kernel functions. Recall that $\rho_\tau(u) = |u|/2 + (\tau - 1/2)u$.

(i) (Gaussian kernel) The Gaussian kernel $K(u) = \phi(u)$, where $\phi(\cdot)$ is the density function of a standard normal distribution. The resulting smoothed loss is $l_h(u) = (h/2)G(u/h) + (\tau - 1/2)u$, where $G(u) = (2/\pi)^{1/2}e^{-u^2/2} + u\{1 - 2\Phi(-u)\}$.

(ii) (Uniform kernel) The uniform kernel is $K(u) = (1/2)\mathbb{1}(|u| \leq 1)$, which is the density function of the uniform distribution on $[-1, 1]$. the resulting smoothed loss is $l_h(u) = (h/2)U(u/h) + (\tau - 1/2)u$, where $U(u) = (u^2/2 + 1/2)\mathbb{1}(|u| \leq 1) + |u|\mathbb{1}(|u| > 1)$ is a Huber-type loss. Convolution plays a role of random smoothing in the sense that $l_h(u) = (1/2)\mathbb{E}(|Z_u|) + (\tau - 1/2)u$, where for every $u \in \mathbb{R}$, $Z_u$ denotes a random variable uniformly distributed between $u - h$ and $u + h$.

(iii) (Laplacian kernel) The Laplacian kernel is $K(u) = e^{-|u|}/2$. We have $l_h(u) = \rho_\tau(u) + he^{-|u|/h}/2$.

(iv) (Logistic kernel) The logistic kernel is $K(u) = e^{-u}/(1 + e^{-u})^2$. The resulting smoothed loss is $l_h(u) = \tau u + h\log(1 + e^{-u/h})$.

(v) (Triangular kernel) The triangular kernel is $K(u) = (1 - |u|)\mathbb{1}(|u| \leq 1)$. The resulting smoothed loss is $l_h(u) = (h/2)l_{tr}(u/h) + (\tau - 1/2)u$, where $l_{tr}(u) := (u^2 - |u|^3/3 + 1/3)\mathbb{1}(|u| \leq 1) + |u|\mathbb{1}(|u| > 1)$.

(vi) (Epanechnikov kernel) The Epanechnikov kernel is $K(u) = (3/4)(1 - u^2)\mathbb{1}(|u| \leq 1)$. The resulting smoothed loss is $l_h(u) = (h/2)E(u/h) + (\tau - 1/2)u$, where $E(u) := (3u^2/4 - u^4/8 + 3/8)\mathbb{1}(|u| \leq 1) + |u|\mathbb{1}(|u| > 1)$.

One can easily check that all the empirical smoothed loss functions above are convex. See Figure 1 for a visualization of Horowitz's and convolution smoothing methods.

In the following, we consider the multi-source transfer learning scenario, where we have a target data set $(\boldsymbol{X}^{(0)}, \boldsymbol{y}^{(0)})$ and $\mathcal{K}$ source data sets with the $k$-th source denoted as $(\boldsymbol{X}^{(k)}, \boldsymbol{y}^{(k)})$, where $\boldsymbol{X}^{(k)} \in \mathbb{R}^{n_k \times p}$, $\boldsymbol{y}^{(k)} \in \mathbb{R}^{n_k}$ for $k = 0, \ldots, \mathcal{K}$. The $i$-th row of $\boldsymbol{X}^{(k)}$ and the $i$-th element of $\boldsymbol{y}^{(k)}$ are denoted as $\boldsymbol{x}_i^{(k)}$ and $y_i^{(k)}$, respectively. The goal is to transfer useful information from the source datasets to improve the estimation

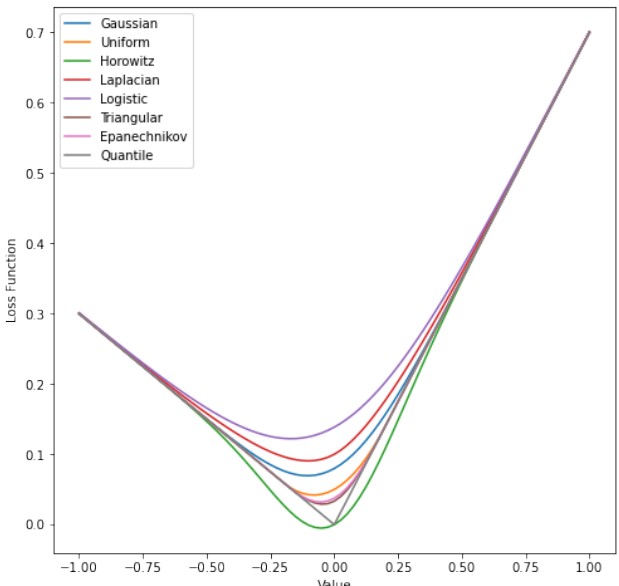

Figure 1: Plots of a standard quantile loss, Horowitz's smoothed quantile loss (Horowitz, 1998) and some proposed convolution-type smoothed quantile loss with different widely used kernel functions.

accuracy of the target parameters. Denote the true target parameter as $\boldsymbol{\beta}^* = \boldsymbol{\omega}^{(0)}$. We assume the responses in the target and source data all follow the linear quantile regression model, that is,

$$y_i^{(k)} = (\boldsymbol{x}_i^{(k)})^{\mathrm{T}} \boldsymbol{\omega}^{(k)} + \epsilon_i^{(k)} \text{ and } \mathbb{P}(\epsilon_i \leq 0 | \boldsymbol{x}_i^{(k)}) = \tau, \ k = 0, \dots, \mathcal{K}.$$

We build our quantile regression transfer learning procedure in the high-dimensional regime with a sparsity assumption. In other words, we assume the dimension $p$ is much larger than the sample size $n_k$ for all $k$ while the target model is $s$-sparse, which satisfies $||\boldsymbol{\beta}^*||_0 = s$. Define the $k$-th contrast as $\boldsymbol{\delta}^{(k)} = \boldsymbol{\beta}^* - \boldsymbol{\omega}^{(k)}$ and $||\boldsymbol{\delta}^{(k)}||_q$ is referred to as the transferring level of source $k$ in the literature, where $q \in \{0, 1\}$. Define the level-$m$ transferring set $\mathcal{A}_m = \{k : ||\boldsymbol{\delta}^{(k)}||_q \leq m\}$ as the set of sources which has transferring level lower than $m$. Denote $n_{\mathcal{A}_m} = \sum_{k \in \mathcal{A}_m} n_k$, $\alpha_k = n_k / (n_{\mathcal{A}_m} + n_0)$ for $k \in \{0\} \cup \mathcal{A}_m$ and $\mathcal{K}_{\mathcal{A}_m} = |\mathcal{A}_m|$.

As stated in the introduction, we will consider two types of transferring level, corresponding to $q \in \{0, 1\}$ respectively. In the case of $q = 0$, the transferring set corresponds to the source data whose contrast vectors have at most $m$ nonzero elements. In the case of $q = 1$, all the coefficients of the contrast vectors can be nonzero, but their absolute magnitude decays at a relatively rapid rate. It will be seen later that as long as $m$ is relatively small, the source data in $\mathcal{A}_m$ can be useful in improving the estimation accuracy of $\boldsymbol{\beta}^*$. In addition, the logic of the algorithm with $\ell_1$-normed $\mathcal{A}_m$ and the algorithm with $\ell_2$-normed $\mathcal{A}_m$ are quite different and we will elaborate on these two different algorithms in the following sections.

## 2.2   The proposed algorithm with an $\ell_1$-norm constrained transferring set

In this section, we propose the transfer learning algorithm with $\ell_1$-norm constrained transferring set, which is motivated by Tian & Feng (2023). This algorithm involves two steps. The first step of our algorithm is to transfer the information from useful sources by pooling all the data in transferable set $\mathcal{A}_m$ and target set $\mathcal{A}_0$ to obtain a primal estimator. We also call it the transferring step. To be more precise, we define a total smoothed loss function for the target and source datasets in the transferable set $\mathcal{A}_m$, i.e.,

$$\hat{Q}_h(\boldsymbol{\omega}) = \frac{1}{n_{\mathcal{A}_m} + n_0} \sum_{k \in \mathcal{A}_m} \sum_{i=1}^{n_k} l_h \big( y_i^{(k)} - (\boldsymbol{x}_i^{(k)})^{\mathrm{T}} \boldsymbol{\omega} \big),$$

where

$$l_h(u) = (\rho_\tau * K_h)(u) = \int_{-\infty}^{\infty} \rho_\tau(v) K_h(v - u) dv.$$

Then for the transferring step, we aim to find the minimizer to the following optimization problem with respect to $\boldsymbol{w} \in \mathbb{R}^p$:

$$\underset{\boldsymbol{\omega}}{\text{minimize}} \left\{ \hat{Q}_h(\boldsymbol{\omega}) + \lambda_\omega ||\boldsymbol{\omega}||_1 \right\}.$$

We denote the minimizer as $\hat{\boldsymbol{\omega}}^{\mathcal{A}_m}$, i.e., $\hat{\boldsymbol{\omega}}^{\mathcal{A}_m} = \arg\min_{\boldsymbol{\omega}} \left\{ \hat{Q}_h(\boldsymbol{\omega}) + \lambda_\omega ||\boldsymbol{\omega}||_1 \right\}$. By selecting an appropriate bandwidth $h$, the iteratively reweighted $\ell_1$-penalized SQR estimator proposed by Tan et al. (2022) shares the same upper bounds for both $\ell_1$ and $\ell_2$ errors as the $\ell_1$-QR estimator, as indicated by Belloni & Chernozhukov (2011). Furthermore, they introduced coordinate descent and ADMM-based algorithms for solving $\ell_1$-penalized quantile regression, which are computationally efficient especially for large-scale problems.

Denote the true parameter in the first step as $\boldsymbol{\omega}^{\mathcal{A}_m}$, and $\boldsymbol{\omega}^{\mathcal{A}_m}$ has the following explicit form:

$$\boldsymbol{\omega}^{\mathcal{A}_m} = \boldsymbol{\beta} + \boldsymbol{\delta}^{\mathcal{A}_m},$$

where $\delta^{\mathcal{A}_m} = \sum_{k \in \mathcal{A}_m} \alpha_k \boldsymbol{\delta}^{(k)}$ and $\alpha_k = n_k/(n_{\mathcal{A}_m} + n_0)$. For the second step (the debiasing step), we correct the bias, $\boldsymbol{\delta}^{\mathcal{A}_m}$, based on the estimator $\hat{\boldsymbol{\omega}}^{\mathcal{A}_m}$ acquired in the transferring step. The smoothed loss function for the target data with respect to $\boldsymbol{\delta}$ is defined as

$$\hat{Q}_g^{(0)}(\hat{\boldsymbol{\omega}}^{\mathcal{A}_m} + \boldsymbol{\delta}) = \frac{1}{n_0} \sum_{i=1}^{n_0} l_g \big( y_i^{(0)} - (\boldsymbol{x}_i^{(0)})^{\mathrm{T}} (\hat{\boldsymbol{\omega}}^{\mathcal{A}_m} + \boldsymbol{\delta}) \big).$$

The error of the debiasing step is under control for a relatively small $m$, since $\boldsymbol{\delta}^{\mathcal{A}_m}$ is a $\ell_1$-sparse high-dimensional vector.

We call this algorithm Oracle $\ell_1$-Trans-SQR as we first assume that all useful sources are known as a priori. Algorithm 1 formally presents the Oracle $\ell_1$-Trans-SQR algorithm.

---

**Algorithm 1:** Oracle $\ell_1$-Trans-SQR

**Input:** Target data $(\boldsymbol{X}^{(0)}, \boldsymbol{y}^{(0)})$, source data $\{(\boldsymbol{X}^{(k)}, \boldsymbol{y}^{(k)})\}_{k=1}^{\mathcal{K}}$, penalty parameters $\lambda_\omega$ and $\lambda_\delta$, transferring set $\mathcal{A}_m$.

**Output:** The estimator $\hat{\boldsymbol{\beta}}$.

**1** Transferring step:

$$\hat{\boldsymbol{\omega}}^{\mathcal{A}_m} \leftarrow \arg\min_{\boldsymbol{\omega}} \left\{ \hat{Q}_h(\boldsymbol{\omega}) + \lambda_\omega ||\boldsymbol{\omega}||_1 \right\}.$$

**2** Debiasing step:

$$\hat{\boldsymbol{\delta}}^{\mathcal{A}_m} \leftarrow \arg\min_{\delta} \left\{ \hat{Q}_g^{(0)}(\hat{\boldsymbol{\omega}}^{\mathcal{A}_m} + \boldsymbol{\delta}) + \lambda_\delta ||\boldsymbol{\delta}||_1 \right\}.$$

**3 return** $\hat{\boldsymbol{\beta}} = \hat{\boldsymbol{\omega}}^{\mathcal{A}_m} + \hat{\boldsymbol{\delta}}^{\mathcal{A}_m}$.

---

If $\mathcal{A}_m$ is unknown, then we need a detection algorithm to find useful transferable sets in practice. We propose a transferrable source detection algorithm which is inspired from the Algorithm 2 in Tian & Feng (2023). Firstly, partition the target data into $q$ subsets. Secondly, fit the penalized smoothed quantile regression on each combination of $(q - 1)$ target subsets and calculate the loss on the remaining target subset. In the following, consider the average cross-validation loss $\hat{L}_0^{(0)}$. Run the transferring step on each combination of $(q - 1)$ target subsets and each source data, and evaluate the loss function on the remaining target subset. Similarly compute the average cross-validation loss $\hat{L}_0^{(k)}$ for each source. Thirdly, calculate the difference between $\hat{L}_0^{(0)}$ and $\hat{L}_0^{(k)}$ for each $k$ and compare it with a predefined threshold. Finally select the sources

---

**Algorithm 2:** Transferable Source Detection

---

**Input:** Target data $(\boldsymbol{X}^{(0)}, \boldsymbol{y}^{(0)})$, all source data $\{(\boldsymbol{X}^{(k)}, \boldsymbol{y}^{(k)})\}_{k=1}^{\mathcal{K}}$, a threshold $C_0$, penalty parameters $\{\{\lambda^{(k)[a]}\}_{k=0}^{\mathcal{K}}\}_{a=1}^{q}$, where $q$ is the number of folds chosen.

**Output:** The set of transferable sources $\hat{\mathcal{A}}$.

**1** Randomly divide $(\boldsymbol{X}^{(0)}, \boldsymbol{y}^{(0)})$ into $q$ equal-sized sets $\{(\boldsymbol{X}^{(0)[i]}, \boldsymbol{y}^{(0)[i]})\}_{i=1}^{q}$.

**2 for** $a = 1$ **to** $q$ **do**

**3** $\quad$ $\hat{\boldsymbol{\beta}}^{(0)[a]} \leftarrow$ fit the penalized quantile regression on $\{(\boldsymbol{X}^{(0)[i]}, \boldsymbol{y}^{(0)[i]})\}_{i=1}^{q} \setminus (\boldsymbol{X}^{(0)[a]}, \boldsymbol{y}^{(0)[a]})$ with penalty parameter $\lambda^{(0)[a]}$.

**4** $\quad$ $\hat{\boldsymbol{\beta}}^{(k)[a]} \leftarrow$ run the transferring step in Algorithm 1 with $\{(\boldsymbol{X}^{(0)[i]}, \boldsymbol{y}^{(0)[i]})\}_{i=1}^{q} \setminus (\boldsymbol{X}^{(0)[a]}, \boldsymbol{y}^{(0)[a]}) \cup (\boldsymbol{X}^{(k)}, \boldsymbol{y}^{(k)})$ and penalty parameter $\lambda^{(k)[a]}$ for all $k \neq 0$.

**5** $\quad$ Calculate the loss function $\hat{L}_0^{[a]}(\hat{\boldsymbol{\beta}}^{(k)[a]})$ on $(\boldsymbol{X}^{(0)[a]}, \boldsymbol{y}^{(0)[a]})$ for $k = 1, \ldots, K$.

**6** $\hat{L}_0^{(k)} \leftarrow \sum_{a=1}^{q} \hat{L}_0^{[a]}(\hat{\boldsymbol{\beta}}^{(k)[a]})/q, \quad \hat{L}_0^{(0)} \leftarrow \sum_{a=1}^{q} \hat{L}_0^{[a]}(\hat{\boldsymbol{\beta}}^{(0)[a]})/q,$
$\hat{\sigma} = \sqrt{\sum_{a=1}^{q} (\hat{L}_0^{[a]}(\hat{\boldsymbol{\beta}}^{(k)[a]}) - \hat{L}_0^{(0)})^2/(q-1)}.$

**7** $\hat{\mathcal{A}} \leftarrow \{k \neq 0 : \hat{L}_0^{(k)} - \hat{L}_0^{(0)} \leq C_0(\hat{\sigma} \vee 0.01)\}.$

**8 return** $\hat{\mathcal{A}}$.

---

whose difference is less than the threshold and include them in the set $\hat{\mathcal{A}}$. The detailed transferable source detection procedure is summarized in Algorithm 2.

With the transferrable source detection algorithm, we propose a feasible Algorithm 3 in practice, in which we first detect useful source datasets $\hat{\mathcal{A}}$ by Algorithm 2 and then run Algorithm 1 using datasets $\{(\boldsymbol{X}^{(k)}, \boldsymbol{y}^{(k)})\}_{k \in \{0\} \cup \hat{\mathcal{A}}}$.

---

**Algorithm 3:** Trans-SQR

---

**Input:** Target data $(\boldsymbol{X}^{(0)}, \boldsymbol{y}^{(0)})$, all source data $\{(\boldsymbol{X}^{(k)}, \boldsymbol{y}^{(k)})\}_{k=1}^{\mathcal{K}}$, a threshold $C_0$ and penalty parameters $\{\{\lambda^{(k)[a]}\}_{k=0}^{\mathcal{K}}\}_{a=1}^{q}$.

**Output:** The estimator $\hat{\boldsymbol{\beta}}$.

**1** Run Algorithm 2 (Transferable Source Detection Algorithm) and output $\hat{\mathcal{A}}$.

**2** Run Algorithm 1 (Oracle Trans-SQR) using data $\{(\boldsymbol{X}^{(k)}, \boldsymbol{y}^{(k)})\}_{k \in \{0\} \cup \hat{\mathcal{A}}}$.

**3 return** $\hat{\boldsymbol{\beta}}$.

---

### 2.3 The proposed algorithm with an $\ell_0$-norm constrained transferring set

In this section we consider a more strict transferable set $\mathcal{A}'_m = \{k : ||\boldsymbol{\delta}^{(k)}||_0 \leq m\}$, where the $\ell_1$-norm discussed in Section 2.2 is replaced by $\ell_0$-norm. Compared with the $\ell_1$-norm, the theoretical analysis of the transfer learning procedure under $\ell_0$-norm is free of the restrictive Assumption 3.4 below, which requires "sufficient" similarity between the target covariance matrix and transferable source covariance matrices. However, as $\ell_0$-norm is not additive, it is not easy to combine target and source data to estimate a primary estimator for the true target parameter. Instead, we correct each source data independently and incorporate the corrected source and target data to make predictions. Certain adjustments need to be made on the proposed transfer learning procedure in Algorithm 1.

This $\ell_0$-norm constrained transfer algorithm is inspired by the idea in Li et al. (2021). Unlike the transferring step in Algorithm 1, the first step of the algorithm in this section is to train each source separately to get primal estimators of $\boldsymbol{\omega}^{(k)}$, $k \in \{1, \ldots, \mathcal{K}\}$, where the smoothed loss function for each source $k$ is

$$\hat{Q}_h^{(k)}(\boldsymbol{\omega}) = \frac{1}{n_k} \sum_{i=1}^{n_k} l_h\big(y_i^{(k)} - (\boldsymbol{x}_i^{(k)})^{\mathrm{T}} \boldsymbol{\omega}\big).$$

In the second step, as the debiasing step in Algorithm 1, we adjust for the differences $\hat{\boldsymbol{\delta}}^{(k)}$ for all $k$ using the target data, which is obtained via

$$\hat{\boldsymbol{\delta}}^{(k)} = \arg\min_{\delta} \left\{ \hat{Q}_g^{(0)}(\hat{\boldsymbol{\omega}}^{(k)} + \boldsymbol{\delta}) + \lambda_\delta ||\boldsymbol{\delta}||_1 \right\},$$

where the smoothed loss function with respect to $\boldsymbol{\delta}$ is defined as

$$\hat{Q}_g^{(0)}(\hat{\boldsymbol{\omega}}^{(k)} + \boldsymbol{\delta}) = \frac{1}{n_0} \sum_{i=1}^{n_0} l_g\big(y_i^{(0)} - (\boldsymbol{x}_i^{(0)})^{\mathrm{T}}(\hat{\boldsymbol{\omega}}^{(k)} + \boldsymbol{\delta})\big).$$

Then a threshold for each $\hat{\boldsymbol{\delta}}^{(k)}$ is computed by only keeping the largest $\sqrt{n_0/\log p}$ elements of $\hat{\boldsymbol{\delta}}^{(k)}$ and letting all the other elements be zero. In the third step, with the estimated "bias" from the second step, the corrected source data has the following form:

$$\left\{ \left( \boldsymbol{X}^{(k)}, \boldsymbol{y}^{(k)} + \boldsymbol{X}^{(k)}\tilde{\boldsymbol{\delta}}^{(k)} \right) \right\}_{k=1}^{\mathcal{K}}.$$

Then, we combine all the corrected sources and target data to estimate the parameter $\boldsymbol{\beta}$ which is of our interest. The above algorithm estimate the source parameters and the contrast vectors individually, while in the Oracle $\ell_1$-Trans-SQR proposed in Section 2.2, a pooled analysis is conducted with data from target and sources, which relies on the homogeneous designs of the covariance matrices among target and source data.

---

**Algorithm 4:** Oracle Trans-SQR with $\ell_0$-norm constrained transferring set

**Input:** Target data $(\boldsymbol{X}^{(0)}, \boldsymbol{y}^{(0)})$, source data $\{(\boldsymbol{X}^{(k)}, \boldsymbol{y}^{(k)})\}_{k=1}^{\mathcal{K}}$, penalty parameters $\lambda_\omega$, $\lambda_\delta$ and $\lambda_\beta$, transferring set $\mathcal{A}_m'$. Let $n = n_0 + n_{\mathcal{A}_m'}$.

**Output:** The estimator $\hat{\boldsymbol{\beta}}$.

**1** For each $k \in \mathcal{A}_m'$,

$$\hat{\boldsymbol{\omega}}^{(k)} \leftarrow \arg\min_{\boldsymbol{\omega}} \left\{ \hat{Q}_h(\boldsymbol{\omega}) + \lambda_\omega^{(k)} ||\boldsymbol{\omega}||_1 \right\}.$$

**2** For each $k \in \mathcal{A}_m'$,

$$\hat{\boldsymbol{\delta}}^{(k)} \leftarrow \arg\min_{\delta} \left\{ \hat{Q}_g^{(0)}(\hat{\boldsymbol{\omega}}^{(k)} + \boldsymbol{\delta}) + \lambda_\delta ||\boldsymbol{\delta}||_1 \right\}.$$

**3** Threshold $\hat{\boldsymbol{\delta}}^{(k)}$ via $\tilde{\boldsymbol{\delta}}^{(k)} = \mathcal{H}_{\sqrt{n_0/\log p}}(\hat{\boldsymbol{\delta}}^{(k)})$, where $\mathcal{H}_k(\boldsymbol{b})$ is formed by setting all but the largest $k$ elements of $\boldsymbol{b}$ to zero.

**4** Joint estimation using source and target data:

$$\hat{\boldsymbol{\beta}} \leftarrow \arg\min_{\beta} \left\{ \frac{1}{n} \sum_{i=1}^{n_0} l_w\big(y_i^{(0)} - (\boldsymbol{x}_i^{(0)})^{\mathrm{T}}\boldsymbol{\beta}\big) \right.$$
$$\left. + \frac{1}{n} \sum_{k \in \mathcal{A}_m'} \sum_{i=1}^{n_k} l_w\big(y_i^{(k)} - (\boldsymbol{x}_i^{(k)})^{\mathrm{T}}(\boldsymbol{\beta} - \tilde{\boldsymbol{\delta}}^{(k)})\big) + \lambda_\beta ||\boldsymbol{\beta}||_1 \right\}.$$

**5 return** $\hat{\boldsymbol{\beta}}$.

---

## 3 Statistical theory

In this section, we establish theoretical guarantees on the algorithms in the above section.

**Assumption 3.1.** There exists $\bar{f} \geq \underline{f} > 0$ such that the conditional density of $\epsilon^{(k)}$ given $\boldsymbol{x}^{(k)}$ satisfies $\underline{f} \leq f_{\epsilon^{(k)}|\boldsymbol{x}^{(k)}}(0) \leq \bar{f}$ almost surely (over $\boldsymbol{x}^{(k)}$) for all $k = 0, \ldots, \mathcal{K}$. Moreover, there exists $l_0 > 0$ such that $|f_{\epsilon^{(k)}|\boldsymbol{x}^{(k)}}(u) - f_{\epsilon^{(k)}|\boldsymbol{x}^{(k)}}(v)| \leq l_0|u - v|$ for all $u, v \in \mathbb{R}$ almost surely (over $\boldsymbol{x}^{(k)}$), and $\boldsymbol{z}^{(k)} = (\Sigma^{(k)})^{-1/2}\boldsymbol{x}^{(k)}$ for $k = 0, \ldots, \mathcal{K}$, where $\Sigma^{(k)}$ denote the covariance matrix of $\boldsymbol{x}^{(k)}$, and

$$\inf_{t \in [0,1], \boldsymbol{v} \in \mathbb{S}^{p-1}} \mathbb{E}\left[f_{\epsilon^{(k)}|\boldsymbol{x}^{(k)}}\left(t(\boldsymbol{z}^{(k)})^{\mathrm{T}}\boldsymbol{v}\right)\left((\boldsymbol{z}^{(k)})^{\mathrm{T}}\boldsymbol{v}\right)^2\right] \geq \underline{f}.$$

**Assumption 3.2.** The kernel function $K : \mathbb{R} \to [0, \infty)$ is symmetric, that is, $K(u) = K(-u)$, and satisfies that $\int_{-\infty}^{\infty} K(u)du = 1$ and $\int_{-\infty}^{\infty} u^2 K(u)du < \infty$. For $a = 1, 2, \ldots$, let $\kappa_a = \int_{-\infty}^{\infty} |u|^a K(u)du$ be the $a$-th absolute moment of $K(\cdot)$. Assume $\sup_{u \in \mathbb{R}} K(u) \leq \bar{\kappa}$ for some $\bar{\kappa} \in (0, 1]$.

**Assumption 3.3.** For $k = 0, \ldots, \mathcal{K}$, $\Sigma^{(k)} = \mathbb{E}[\boldsymbol{x}^{(k)}(\boldsymbol{x}^{(k)})^{\mathrm{T}}]$ is positive definite and $\boldsymbol{z}^{(k)} = (\Sigma^{(k)})^{-1/2}\boldsymbol{x}^{(k)} \in \mathbb{R}^p$ is sub-exponential: there exist constants $v_0, c_0 \geq 1$ such that $\mathbb{P}(|(\boldsymbol{z}^{(k)})^{\mathrm{T}}\boldsymbol{u}| \geq v_0\|\boldsymbol{u}\|_2 \cdot t) \leq c_0 e^{-t}$ for all $\boldsymbol{u} \in \mathbb{R}^p$ and $t \geq 0$. For convenience, we assume $c_0 = 1$, and write $\sigma_x^2 = \max_{1 \leq j \leq p} \mathbb{E}(x_j^2)$.

Under Assumption 3.3, the $a$-th ($a \geq 3$) absolute moments of all the one-dimensional marginals of $\boldsymbol{z}$ are uniformly bounded: $\mu_a := \sup_{\boldsymbol{u} \in \mathbb{S}^{p-1}} \mathbb{E}|(\boldsymbol{z}^{(k)})^{\mathrm{T}}\boldsymbol{u}|^a \leq a!v_0^a$. In particular, $\mu_1 \leq \mu_2^{1/2} = 1$.

Meanwhile, for every $\delta \in (0, 1]$, define

$$\eta_\delta = \inf\left\{\eta > 0 : \mathbb{E}\left[\left((\boldsymbol{z}^{(k)})^{\mathrm{T}}\boldsymbol{v}\right)^2 \mathbb{1}\left(|(\boldsymbol{z}^{(k)})^{\mathrm{T}}\boldsymbol{v}| > \eta\right)\right] \leq \delta \text{ for all } \boldsymbol{v} \in \mathbb{S}^{p-1}\right\}. \tag{3}$$

Since $\mathbb{E}[(\boldsymbol{z}^{(k)})^{\mathrm{T}}\boldsymbol{v}]^2 = 1$ for any $\boldsymbol{v} \in \mathbb{S}^{p-1}$, $\eta_\delta$ is well-defined for each $\delta$, and depends implicitly on the distribution of $\boldsymbol{z}^{(k)}$.

**Assumption 3.4.** Denote

$$\tilde{\Sigma} = \sum_{k=0}^{\mathcal{K}} \alpha_k \int_0^1 \nabla^2 Q^{(k)}((1-t)\boldsymbol{\beta}^* + t\boldsymbol{\omega}^*)dt$$

$$\tilde{\Sigma}^{(k)} = \int_0^1 \nabla^2 Q^{(k)}((1-t)\boldsymbol{\beta}^* + t\boldsymbol{\omega}^{(k)})dt,$$

where $\nabla^2 Q^{(k)}((1-t)\boldsymbol{\beta}^* + t\boldsymbol{\omega}) = \mathbb{E}\{f_{\epsilon|\boldsymbol{x}}(t\boldsymbol{\omega} - t\boldsymbol{\beta}^*) \cdot \boldsymbol{x}^{(k)}(\boldsymbol{x}^{(k)})^{\mathrm{T}}\}$. Define

$$C_1 = \sup_{0 \leq k \leq \mathcal{K}} \|\tilde{\Sigma}^{-1}\tilde{\Sigma}^{(k)}\|_1.$$

Let $C_1$ be bounded, that is $C_1 < \infty$.

Assumption 3.1 imposes the Lipschitz continuity on the conditional density $f_{\epsilon|\boldsymbol{x}}(\cdot)$. Assumption 3.2 holds for most commonly used kernel functions, for instance, uniform kernel, Gaussian kernel, etc.

Assumption 3.3 assumes a sub-exponential condition on the random covariates characterized by a well-behaved covariance structure. In particular, $\mu_4$ can be regarded as the uniform kurtosis parameter.

Assumption 3.4 restricts the difference between the target covariance matrix and transferable source covariance matrix in some sense, which guarantees the estimator at the transferring step is close to the true parameter $\boldsymbol{\beta}^*$. This assumption is commonly used in other transfer learning works, Tian & Feng (2023); Li et al. (2022); Zhang & Zhu (2022); Huang et al. (2022).

Formally, we consider the parameter space

$$\Theta(s, m) = \left\{\boldsymbol{\beta}^*, \{\boldsymbol{\omega}^{(k)}\}_{k \in \mathcal{A}_m} : \|\boldsymbol{\beta}^*\|_0 \leq s, \sup_{k \in \mathcal{A}_m} \|\boldsymbol{\omega}^{(k)} - \boldsymbol{\beta}^*\|_1 \leq m\right\}.$$

## 3.1 Estimation with an $\ell_1$-norm constrained transferring set

**Proposition 3.1.** (Local Restricted Strong Convexity) Assume Assumptions 3.1 - 3.3 hold. Let $\Delta = \boldsymbol{\omega} - \boldsymbol{\omega}^*$, $n = n_{\mathcal{A}_m} + n_0$ and $\underline{\kappa} = \min_{|u| \leq 1} K(u) > 0$. If $(r, h, n, d)$ satisfies

$$\max\{4\eta_{1/4}r, 32v_0\gamma_1^{1/2}d\mu_4^{1/2}\} \leq h \leq \underline{f}/l_0 \text{ and } nh \gtrsim \bar{f}\underline{f}^{-2}\eta_{1/4}^2\mu_4\sigma_x^2 \log p,$$

for any $\boldsymbol{\omega} \in \boldsymbol{\omega}^* + \mathbb{B}_\Sigma(r)$ and $\boldsymbol{\omega}^* \in \boldsymbol{\omega}^{(k)} + \mathbb{B}_1(d)$,

$$\hat{Q}_h(\boldsymbol{\omega}) - \hat{Q}_h(\boldsymbol{\omega}^*) - \left(\nabla \hat{Q}_h(\boldsymbol{\omega}^*)\right)^{\mathrm{T}}(\boldsymbol{\omega} - \boldsymbol{\omega}^*) \geq \phi_1 ||\Delta||_\Sigma^2 - \phi_2 \sqrt{\frac{\log p + \log n}{nh}} ||\Delta||_1 ||\Delta||_\Sigma, \tag{4}$$

with probability at least $1 - (pn)^{-1}$, where $\phi_1 = \underline{\kappa} \cdot \underline{f}/10$ and $\phi_2 > 0$ is a constant depending only on $(\underline{\kappa}, \underline{f})$.

**Proposition 3.2.** Assume Assumptions 3.1 - 3.3 hold. Let $\boldsymbol{v} = \boldsymbol{\beta}_1 - \boldsymbol{\beta}_2, \phi_1' = \underline{\kappa} \cdot \underline{f}/25$ and $\phi_2' = C_{\underline{\kappa},\underline{f}}^2/(2\phi_1')$. If $\max\{4\eta_{1/4}r, 8v_0 r\mu_4^{1/2}\} \leq g \leq \bar{f}/l_0$ with $\eta_{1/4}$ defined in (3) and $n_0 g \gtrsim \bar{f}\underline{f}^{-2}\eta_{1/4}^2\mu_4 s$, then for any $\boldsymbol{v} \in \mathbb{B}_\Sigma(r)$,

$$\hat{Q}_g^{(0)}(\boldsymbol{\beta}_1) - \hat{Q}_g^{(0)}(\boldsymbol{\beta}_2) - \left(\nabla \hat{Q}_g^{(0)}(\boldsymbol{\beta}_2)\right)^{\mathrm{T}}(\boldsymbol{\beta}_1 - \boldsymbol{\beta}_2) \geq \alpha_1 ||\boldsymbol{v}||_\Sigma^2 - C_{\underline{\kappa},\underline{f}}\sqrt{\frac{\log p + \log n_0}{n_0 g}} ||\boldsymbol{v}||_1 ||\boldsymbol{v}||_\Sigma,$$

with probability at least $1 - (pn_0)^{-1}$, where $C_{\underline{\kappa},\underline{f}} > 0$ is a constant depending only on $(\underline{\kappa}, \underline{f})$. By the arithmetic mean-geometric mean inequality

$$C_{\underline{\kappa},\underline{f}}\sqrt{\frac{\log p + \log n_0}{n_0 g}} ||\boldsymbol{v}||_1 ||\boldsymbol{v}||_\Sigma \leq \frac{\phi_1'}{2} ||\boldsymbol{v}||_\Sigma^2 + \frac{C_{\underline{\kappa},\underline{f}}^2}{2\phi_1'} \frac{\log p + \log n_0}{n_0 g} ||\boldsymbol{v}||_1^2,$$

we have

$$\hat{Q}_g^{(0)}(\boldsymbol{\beta}_1) - \hat{Q}_g^{(0)}(\boldsymbol{\beta}_2) - \left(\nabla \hat{Q}_g^{(0)}(\boldsymbol{\beta}_2)\right)^{\mathrm{T}}(\boldsymbol{\beta}_1 - \boldsymbol{\beta}_2) \geq \frac{\phi_1'}{2} ||\boldsymbol{v}||_\Sigma^2 - \phi_2' \frac{\log p + \log n_0}{n_0 g} ||\boldsymbol{v}||_1^2,$$

with probability at least $1 - (pn_0)^{-1}$.

In the debiasing step, we need another restricted strong convexity condition with both $|| \cdot ||_1^2$ and $|| \cdot ||_\Sigma^2$ in the lower bound. Proposition 3.2 provides that kind condition.

Finally, with the above establishments of restricted strong convexity, we are able to obtain the main result for the two-step transfer learning algorithm on quantile regression.

**Theorem 3.1.** Assume Assumptions 3.1 - 3.4 hold. Suppose $m \leq s\sqrt{\log(p)/n_0}$, $n_0 \geq Cs^2 \log p$ and $n_{\mathcal{A}_m} \gtrsim n_0$, where $C > 0$ is a constant. Also let

$$\log(p)/(n_{\mathcal{A}_m} + n_0) \lesssim h \leq \min\{\underline{f}/(2l_0\kappa_1), (s^{1/2}\lambda_{\boldsymbol{\omega}})^{1/2}\}$$
$$s \log(p)/n_0 \lesssim g \leq \left(\log(p)/n_0\right)^{1/4}.$$

We take $\lambda_{\boldsymbol{\omega}} = C_{\boldsymbol{\omega}}\sqrt{\log(p)/(n_{\mathcal{A}_m} + n_0)}$, $\lambda_{\boldsymbol{\delta}} = C_{\boldsymbol{\delta}}\sqrt{\log(p)/n_0}$, where $C_{\boldsymbol{\omega}}$ and $C_{\boldsymbol{\delta}}$ are sufficiently large constants, then

$$||\hat{\boldsymbol{\beta}} - \boldsymbol{\beta}^*||_\Sigma \lesssim \sqrt{m}\left(\frac{\log p}{n_0}\right)^{1/4} + \sqrt{s}\left(\frac{\log p}{n_0}\right)^{1/4}\left(\frac{\log p}{n_{\mathcal{A}_m} + n_0}\right)^{1/4}, \tag{5}$$

$$||\hat{\boldsymbol{\beta}} - \boldsymbol{\beta}^*||_1 \lesssim s\sqrt{\frac{\log p}{n_{\mathcal{A}_m} + n_0}} + \left(\frac{\log p}{n_{\mathcal{A}_m} + n_0}\right)^{\frac{1}{4}}\sqrt{sm} + m, \tag{6}$$

with probability at least $1 - p^{-1}$.

**Remark 3.1.** In the trivial case where $\mathcal{A}_m$ is an empty set, the upper bound in (5) is $\mathcal{O}_P(\sqrt{s\log(p)/n_0})$. When $\mathcal{A}_m$ is non-empty, the upper bound in (5) is sharper than $\sqrt{s\log(p)/n_0}$ and the upper bound in (6) is sharper than $s\sqrt{\log(p)/n_0}$, if $n_{\mathcal{A}_m} \gtrsim n_0$ and $m < s(\log(p)/n_0)^{1/2}$.

The above theorem gives the convergence rate of the Trans-SQR estimator under $\ell_1/\ell_2$-errors. As the above remarks stated, if the total sample size of the transferable sources is significantly larger than the target sample size, the Trans-SQR estimator could even achieve a sharper convergence rate with some proper choices of the transferable level of the contrasts and the smoothing bandwidth in the debiasing step. As some previous works show, our theorem shares similar estimation error bounds as the results in Tian & Feng (2023) and Li et al. (2022).

### 3.2 Estimation with an $\ell_0$-norm constrained transferring set

**Assumption 3.5.** For $k = 0, \ldots, \mathcal{K}$, the covariate vector $\boldsymbol{x}^{(k)}$ is compactly supported with

$$\zeta_p^{(k)} := \sup_{\boldsymbol{x}^{(k)} \in \mathbb{R}^p} \left\| (\Sigma^{(k)})^{-1/2} \boldsymbol{x}^{(k)} \right\|_2 < \infty,$$

and $\|\boldsymbol{x}^{(k)}\|_\infty \leq B$ almost surely for some $B \geq 1$, where $\Sigma^{(k)}$ is positive definite. Without loss of generality, assume $B = 1$. In addition, $\mu_a = \sup_{\boldsymbol{u} \in \mathbb{S}^{p-1}} \mathbb{E} |(\boldsymbol{z}^{(k)})^{\mathrm{T}} \boldsymbol{u}|^a < \infty$ for $a = 1, \ldots, 4$.

**Remark 3.2.** Assumption 3.5 is a stronger version of Assumption 3.3. Note that quantile regression has Hessian matrix $\nabla^2 \hat{Q}_h(\boldsymbol{\beta}) = (1/n) \sum_{i=1}^n K_h(\boldsymbol{x}_i^{\mathrm{T}} \boldsymbol{\beta} - y_i) \boldsymbol{x}_i \boldsymbol{x}_i^{\mathrm{T}}$, where $\hat{Q}_h(\boldsymbol{\beta})$ is the smoothed empirical quantile loss and $K_h(u) = (1/h) K(u/h)$. Unlike the generalized linear regression, there is a smoothing bandwidth $h$ in the denominator. We import Assumption 3.5 to provide convenience for bounding the difference $\nabla \hat{Q}_h(\boldsymbol{\beta}) - \nabla \hat{Q}_h(\boldsymbol{\beta}^*)$.

**Proposition 3.3.** (RSC in Step 2) Assume Assumptions 3.1, 3.2, 3.5 hold. Let $\boldsymbol{v} = \boldsymbol{\beta}_1 - \boldsymbol{\beta}_2$. If $\max\{4r\eta_{1/4}, 32v_0 r \mu_4^{1/2}\} \leq g \leq \bar{f}/l_0$ and $n_0 g \gtrsim \bar{f} \underline{f}^{-2} \eta_{1/4}^2 \mu_4 s$, then for any $\boldsymbol{v} \in \mathbb{B}_\Sigma(r)$,

$$\hat{Q}_g^{(0)}(\boldsymbol{\beta}_1) - \hat{Q}_g^{(0)}(\boldsymbol{\beta}_2) - \left( \nabla \hat{Q}_g^{(0)}(\boldsymbol{\beta}_2) \right)^{\mathrm{T}} (\boldsymbol{\beta}_1 - \boldsymbol{\beta}_2) \geq 0.1 \underline{\kappa} \cdot \underline{f} \|\boldsymbol{v}\|_\Sigma^2,$$

with probability at least $1 - (pn_0)^{-1}$.

**Theorem 3.2.** Assume Assumptions 3.1, 3.2, 3.5 hold. Let

$$\log(p)/n_0 \lesssim h \leq \min\{\underline{f}/(2l_0 \kappa_1), (s^{1/2} \lambda_\omega)^{1/2}\}$$

$$(s + m) \log(p)/n_0 \lesssim g \leq \left( (s + m) \log(p)/n_0 \right)^{1/4}$$

$$m \log(p)/n \lesssim w \leq \left( m \log(p)/n \right)^{1/4},$$

where $n = n_0 + n_{\mathcal{A}'_m}$. Meanwhile, suppose $m \leq s$, $n_k \geq n_0$ and $n_0 \geq Cs^2 \log p$, where $C > 0$ is a constant. We take $\lambda_\omega^{(k)} = C_\omega \sqrt{\log(p)/n_k}$, $\lambda_\delta = C_\delta \sqrt{\log(p)/n_0}$ and $\lambda_\beta = C_\beta \sqrt{\log(p)/n}$, where $C_\omega$, $C_\delta$ and $C_\beta$ are sufficiently large constants, then

$$\|\hat{\boldsymbol{\beta}} - \boldsymbol{\beta}^*\|_\Sigma \lesssim \sqrt{\frac{s \log p}{n}} + \sqrt{\frac{sm \log p}{n_0}}, \tag{7}$$

$$\|\hat{\boldsymbol{\beta}} - \boldsymbol{\beta}^*\|_1 \lesssim s\sqrt{\frac{\log p}{n}} + s\sqrt{\frac{m \log p}{n_0}}, \tag{8}$$

with probability at least $1 - p^{-1}$.

The above theorem gives the convergence rate of the Trans-SQR estimator under $\ell_1/\ell_2$-errors, where the contrast vectors are characterized in terms of the $\ell_0$-norm. If the sample size of the target data is large enough and the total sample size of the transferable sources is significantly larger than the target sample size, the Trans-SQR estimator could achieve a sharp convergence rate with some proper choices of the transferable level of the contrasts.

**Remark 3.3.** As mentioned above, Assumption 3.4 is to make sure that the estimation error in the transferring step is small enough. However, Theorem 3.2 does not require Assumption 3.4 because Algorithm 4 learns the parameter $\boldsymbol{w}^{(k)}$ independently in Step 1 and reduces the bias in Step 2. For Step 1, the upper bound of the difference between the estimator $\hat{\boldsymbol{w}}^{(k)}$ and true parameter $\boldsymbol{w}^{(k)}$ can be controlled by the sample size of the each source data and the $\ell_0$ transferable level $m$. For Step 2, the estimated $\hat{\boldsymbol{\delta}}$ could also be closed enough to the true difference between the target and source parameter by having an appropriate target sample size. Therefore, if both the target and source sample sizes are large enough, the error of Algorithm 4 would be well controlled without Assumption 3.4.

## 4 Numerical studies

In this section, we evaluate the performance of our proposed algorithms via numerical experiments. The methods in the following section include Smoothed Quantile Regression (SQR) on target data, the Oracle-Trans-SQR, $\mathcal{A}_m$-Trans-SQR and the Naive-Trans-SQR, which naïvely assumes $\mathcal{A}_m = 1, \ldots, \mathcal{K}$ in the Oracle Trans-SQR. The purpose of including the Naive-Trans-Lasso is to understand the overall informative level of the auxiliary samples.

### 4.1 Transfer learning on an $\ell_1$-normed $\mathcal{A}_m$

We consider $p = 500$, $n_0 = 200$, and $n_1, \ldots, n_{10} = 150$. The covariates from target $\boldsymbol{x}_i^{(0)}$ are i.i.d. Gaussian with mean zero and covariance matrix $\boldsymbol{\Sigma}$ with $\Sigma_{jj'} = 0.5^{|j-j'|}$ for all $i = 1, \ldots, n_0$ and $\epsilon_i^{(0)}$ are i.i.d. Gaussian with mean zero and variance one for all $i$. For $k \in \mathcal{A}_m$, $\boldsymbol{x}_i^{(k)} \sim \mathcal{N}(\boldsymbol{0}_p, \boldsymbol{\Sigma} + \boldsymbol{\epsilon}\boldsymbol{\epsilon}^{\mathrm{T}})$, where $\boldsymbol{\epsilon} \sim \mathcal{N}(\boldsymbol{0}_p, 0.3^2\boldsymbol{I}_p)$. For the target, the true parameter $\boldsymbol{\beta}^*$, we set $s = 5$, $\beta_j = 0.5$ for $j \in \{1, \ldots, s\}$, and $\beta_j = 0$ otherwise. Denote $\boldsymbol{\mathcal{R}}_p^{(k)}$ as $p$ independent Rademacher variables. $\boldsymbol{\mathcal{R}}_p^{(k)}$ is independent with $\boldsymbol{\mathcal{R}}_p^{(k')}$ for any $k \neq k'$. For any source data $k$ in $\mathcal{A}_m$, we let the true parameter $\boldsymbol{\omega}^{(k)} = \boldsymbol{\beta}^* + (m/p)\boldsymbol{\mathcal{R}}_p^{(k)}$, where $m \in \{5, 10\}$. For any source data $k'$ not in $\mathcal{A}_m$, the true parameter $\boldsymbol{\omega}^{(k')} = \boldsymbol{\beta}^* + (2m/p)\boldsymbol{\mathcal{R}}_p^{(k')}$. We train the four methods with 100 reproductions and record their average $\ell_2$-estimation errors under different settings of $\tau$. Figure 2 shows the changes of the estimation errors along with the amount of the transferable sources.

We observe from Figure 2 that the Oracle-Trans-SQR has the best performance among all the methods and $\mathcal{A}_m$-Trans-SQR has almost the same performance as the Oracle-Trans-SQR, which indicates that the transferable source detection algorithm still works under the smoothed quantile regression models. Meanwhile, compared with SQR on target, the estimation errors of the Oracle-Trans-SQR and $\mathcal{A}_m$-Trans-SQR are always smaller, which means that the source data which share some similarities in $\ell_1$-norm with the target data could improve the estimation. Another observation is that the performance of $\mathcal{A}_m$-Trans-SQR consistently improves as more and more source data are transferable. This matches the theoretical $\ell_2$-estimation error bounds which become sharper as $n_{\mathcal{A}_m}$ grows.

### 4.2 Transfer learning on an $\ell_0$-normed $\mathcal{A}_m$

We consider $p = 500$, $n_0 = 200$, and assume that there are $2, 4, 6, 8, 10$ transferable sources with the sample sizes 400. The covariates from target $\boldsymbol{x}_i^{(0)}$ are i.i.d. Gaussian with mean zero and covariance matrix $\boldsymbol{\Sigma}$ with $\Sigma_{jj'} = 0.5^{|j-j'|}$. The covariates from source $\boldsymbol{x}_i^{(k)}$ are also i.i.d. Gaussian with mean zero, but with covariance matrix $\boldsymbol{\Sigma} + \boldsymbol{\epsilon}\boldsymbol{\epsilon}^{\mathrm{T}}$, where $\boldsymbol{\epsilon} \sim \mathcal{N}(\boldsymbol{0}_p, 0.3^2\boldsymbol{I}_p)$. For the target, the true parameter $\boldsymbol{\beta}^*$, we set $s = 5$, $\beta_j = 1$ for $j \in \{1, \ldots, s\}$, and $\beta_j = 0$ otherwise. For the source, their true parameter $\boldsymbol{w}^{(k)}$ is generated from $w_j^{(k)} = \beta_j^* + \Delta\mathbb{1}(j \in M)$, where $M$ is a random subset of $[p]$ with $|M| = m$. We take $m \in \{2, 4\}$, and $\Delta = 2$. Figure 4 and 5 show the $\ell_2$-estimation errors in different settings of $m$.

From the results, Trans-SQR with $\ell_0$-norm constrained transferring set has better performances than SQR only on target and SQR on all sources and target. Meanwhile, when the target data sample size $n_0$ becomes larger, the performance of Trans-SQR increases quickly, which accords with our results that the estimation error is depend on the target sample size. There are considerable decreases in estimation errors of Trans-SQR when the transferable level increases or $\Delta$ increases, which corresponds to the difference on components between target and source populations.

## 5 Conclusion

This paper studies transfer learning for high-dimensional quantile regression models, employing convolution-type smoothing techniques. The proposed algorithms focus on leveraging $\ell_1/\ell_0$-normed transferable source populations to improve estimation accuracy of the target regression coefficients. We derive error bounds for the estimators in terms of $\ell_1/\ell_2$-norms for the algorithms. Theoretical analysis reveals that these error bounds surpass those of the classical penalized quantile regression estimator, which only utilizes the target

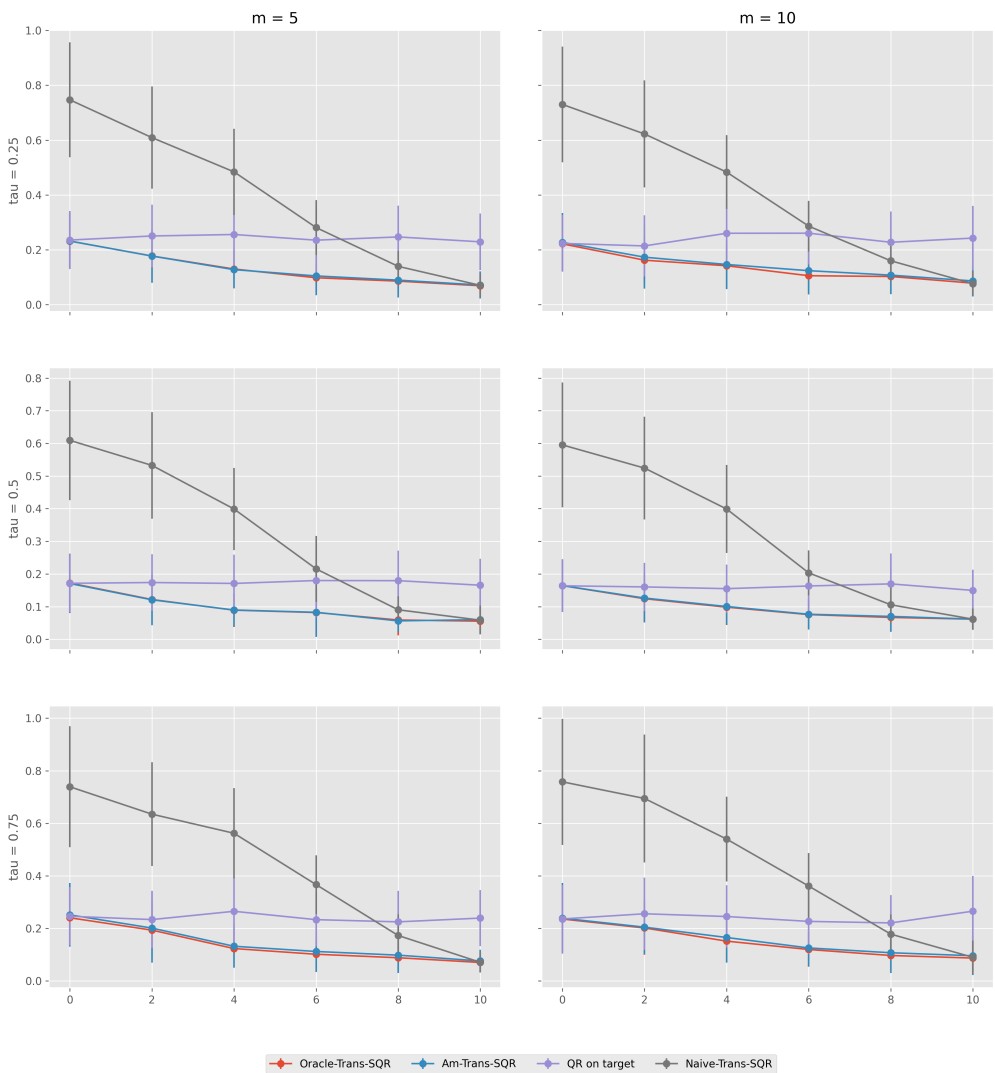

Figure 2: $\ell_2$ estimation errors of several methods under quantile levels $\tau = 0.25, 0.5, 0.75$, over 100 repetitions, where Oracle-Trans-SQR is Algorithm 1.

data, provided that the target and source populations exhibit sufficient similarity. Furthermore, we propose a transferable source detection algorithm to identify informative sources from the available sources when the set of informative sources is unknown. Numerical experiments validate our theoretical results.

## Acknowledgments

Wen-Xin Zhou is supported by the NSF grant NSF DMS-2401268. Yong He is supported by NSF China (12171282) and Qilu Young Scholars Program of Shandong University.

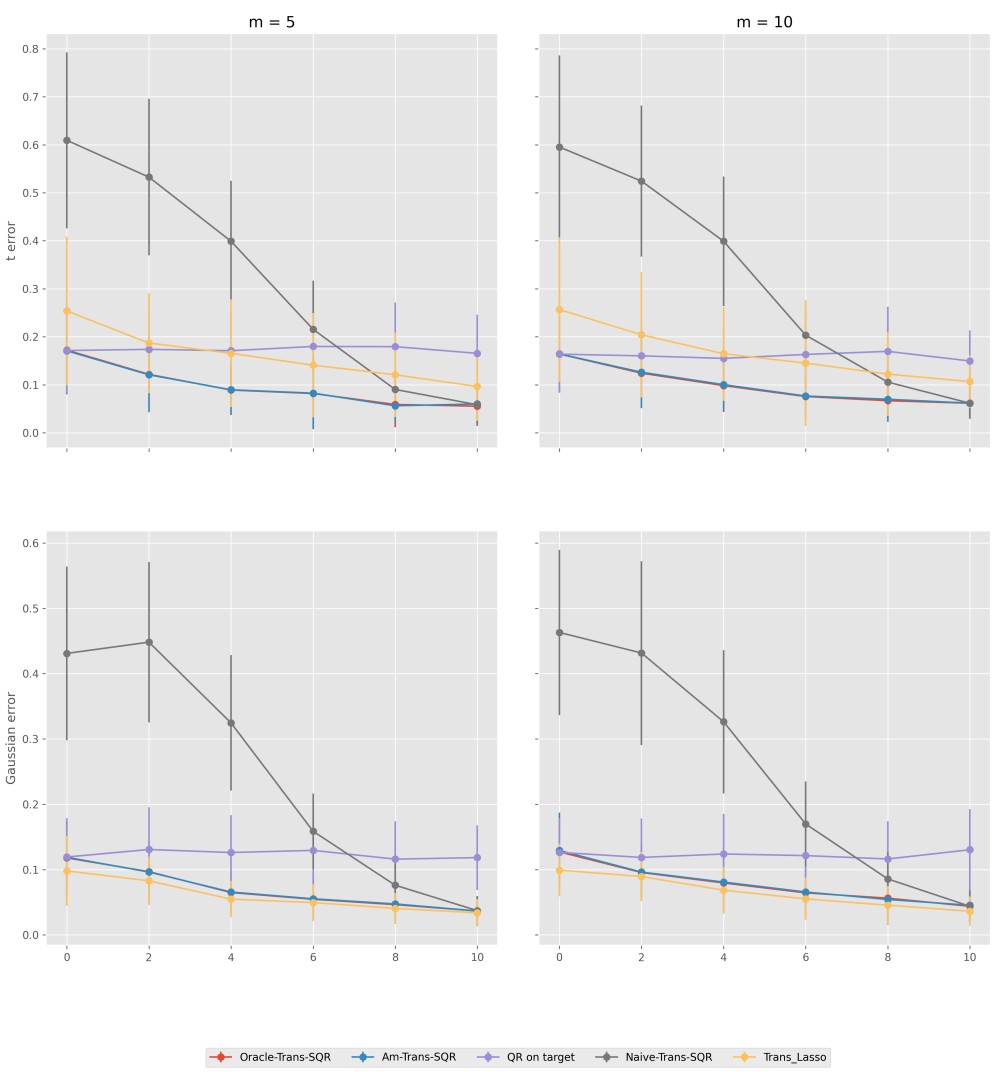

Figure 3: $\ell_2$ estimation errors of several methods for Gaussian and $t_2$ errors, over 100 repetitions.

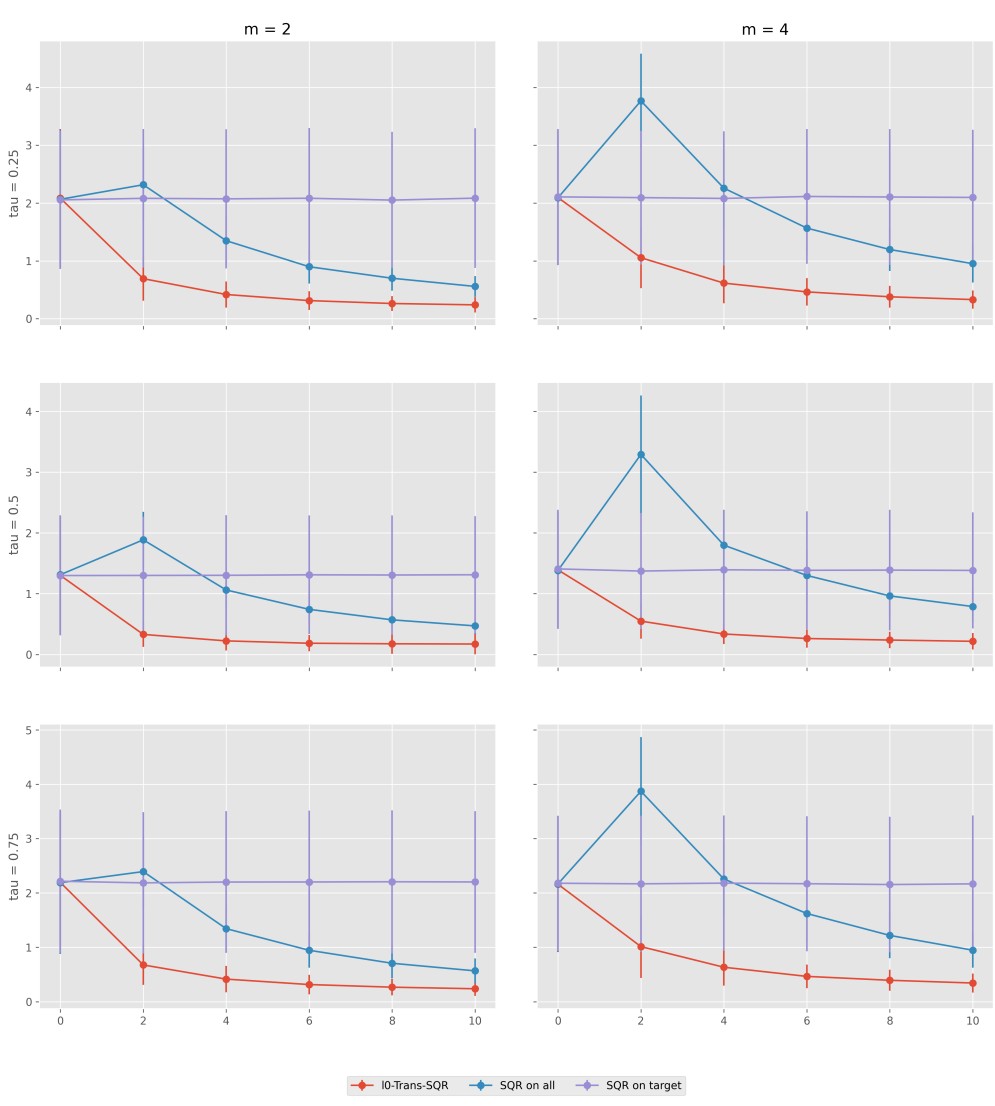

Figure 4: $\ell_2$ estimation errors of several methods with $\ell_0$ constraints for $t_2$ errors, over 100 repetitions.

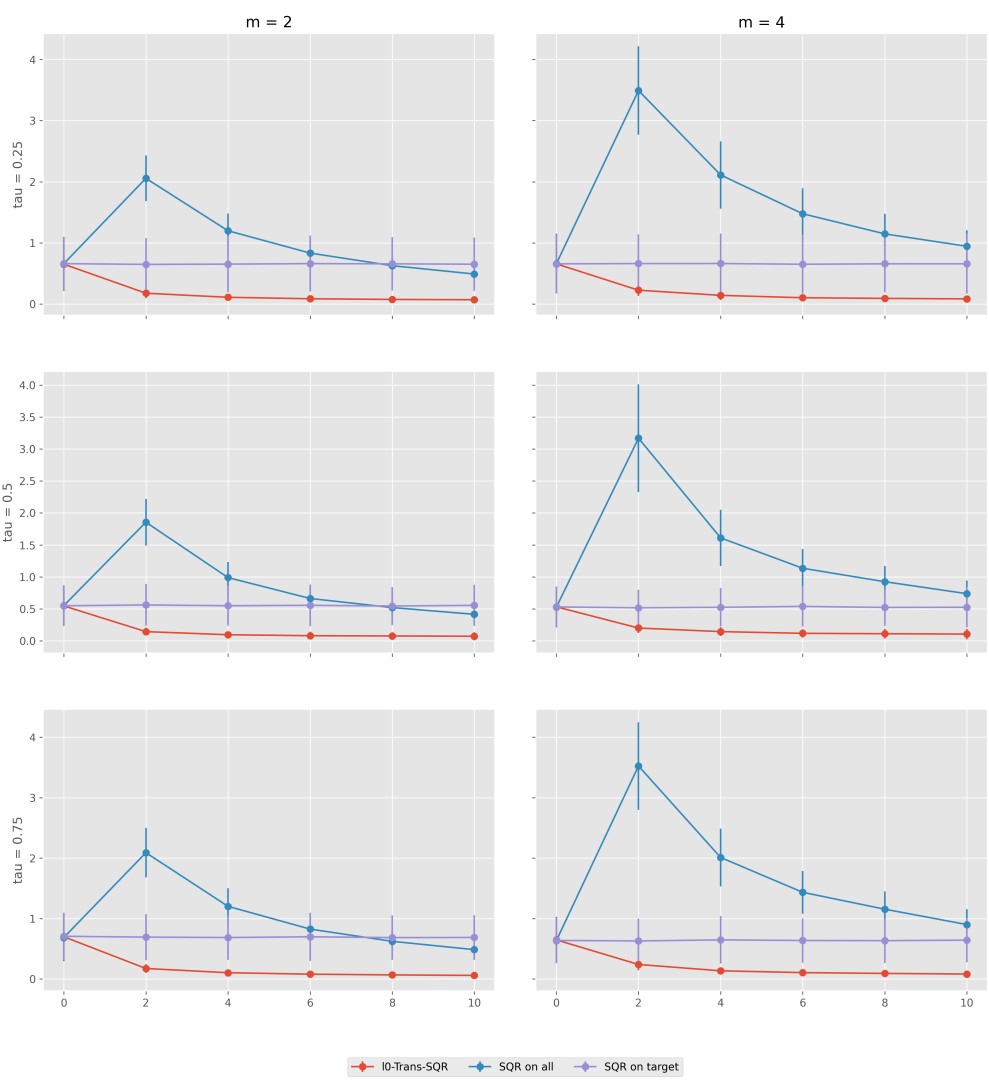

Figure 5: $\ell_2$ estimation errors of several methods with $\ell_0$ constraints for Gaussian errors, over 100 repetitions.

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

# A   Appendix: Proofs of the main results

## A.1   Technical Lemmas

For $\boldsymbol{\omega} \in \mathbb{R}^p$, suppose $\Delta = \boldsymbol{\omega} - \boldsymbol{\omega}^*$. Define

$$\hat{R}_h(\Delta) = \hat{Q}_h(\boldsymbol{\omega}) - \hat{Q}_h(\boldsymbol{\omega}^*) - \big(\nabla \hat{Q}_h(\boldsymbol{\omega}^*)\big)^{\mathrm{T}}(\boldsymbol{\omega} - \boldsymbol{\omega}^*),$$
$$\hat{D}_h(\Delta) = \hat{Q}_h(\boldsymbol{\omega}) - \hat{Q}_h(\boldsymbol{\omega}^*),$$

and their population counterparts $R_h(\Delta) = \mathbb{E}\{\hat{R}_h(\Delta)\}$ and $D_h(\Delta) = \mathbb{E}\{\hat{D}_h(\Delta)\}$, where $\boldsymbol{\omega}^*$ is the true parameter of the transferring step in the algorithm.

**Lemma A.1.** Let $\boldsymbol{\beta}^*$ be the true target parameter, then $||\boldsymbol{\omega}^* - \boldsymbol{\beta}^*||_1 \le C_1 m$, where $C_1 = \sup_k ||\tilde{\Sigma}^{-1}\tilde{\Sigma}^{(k)}||_1$ and $\tilde{\Sigma}^{-1}, \tilde{\Sigma}^{(k)}$ are given in Assumption 3.4.

Note that $\boldsymbol{w}^*$ has the explicit form, $\boldsymbol{w}^* = \boldsymbol{\beta}^* + \boldsymbol{\delta}^*$. Lemma A.1 gives an upper bound of the distance between the true $\boldsymbol{\beta}^*$ and the true estimate in transferring step. In other words, the $\ell_1$-norm of $\boldsymbol{\delta}^*$ is controlled by $m$.

**Lemma A.2.** Define $\boldsymbol{\pi}_h^* = \boldsymbol{\pi}_h(\boldsymbol{\beta}^*) \in \mathbb{R}^p$, where $\boldsymbol{\pi}_h(\boldsymbol{\beta}) = \nabla \hat{Q}_h(\boldsymbol{\beta}) - \nabla Q_h(\boldsymbol{\beta})$. Assumptions 3.1 - 3.3 ensure that for any $t > 0$,

$$||\boldsymbol{\pi}_h^*||_\infty \le \sigma \sqrt{\{\tau(1-\tau) + Ch^2\} \frac{2t}{n_{\mathcal{A}_m} + n_0}} + \max(1-\tau, \tau) \frac{t}{n_{\mathcal{A}_m} + n_0},$$

with probability at least $1 - 2pe^{-t}$, where $C = (\tau + 1)l_0 \kappa_2$ and $\sigma = \max_{1 \le j \le p} \sigma_{jj}$.

In both transferring and debiased steps, we need to restrict the regularization parameters $\lambda_\omega$ (or $\lambda_\delta$) to be no smaller than $2||\boldsymbol{\pi}_h^*||_\infty$ (or $2||\boldsymbol{\pi}_g^*||_\infty$). This Lemma helps to specify the choice of the parameters.

**Lemma A.3.** Define $b_h^* = ||\Sigma^{-1/2} \nabla Q_h(\boldsymbol{\omega}^*)||_2$, which quantifies the smoothing bias, then for some $\kappa_2 > 0$

$$b_h^* \le l_0 \kappa_2 \frac{h^2}{2},$$

where $l_0$ is the Lipschitz constant of the density $f_{\epsilon|\boldsymbol{x}}(\cdot)$.

**Lemma A.4.** For $r, l > 0$, define

$$\psi(r, l) = \sup_{\boldsymbol{\beta} \in \boldsymbol{\beta}^* + \mathbb{B}_\Sigma(r) \cap \mathbb{B}_1(l)} \left\| \frac{1}{n} \sum_{i=1}^n \big\{ \bar{K}_h(\boldsymbol{x}_i^{\mathrm{T}}\boldsymbol{\beta} - y_i) - \bar{K}_h(\boldsymbol{x}_i^{\mathrm{T}}\boldsymbol{\beta}^* - y_i) \big\} \boldsymbol{x}_i \right\|_\infty.$$

For any $t > 0$, with probability at least $1 - e^{-t}$,

$$\psi(r, l) \lesssim \frac{l}{h} \sqrt{\frac{\log p}{n}} + \bar{f}^{1/2} r \sqrt{\frac{t + \log p}{nh}} + \frac{t + \log p}{n}.$$

## A.2   Proof of Proposition 3.1

Define the Taylor error

$$\mathcal{T}(\boldsymbol{\omega}, \boldsymbol{\omega}^*) = \hat{Q}_h(\boldsymbol{\omega}) - \hat{Q}_h(\boldsymbol{\omega}^*) - \big(\nabla \hat{Q}_h(\boldsymbol{\omega}^*)\big)^{\mathrm{T}}(\boldsymbol{\omega} - \boldsymbol{\omega}^*).$$

In the following proofs, we consider the subset of $\boldsymbol{\omega}$, $\boldsymbol{\omega} \in \boldsymbol{\omega}^* + \mathbb{B}_\Sigma(r) \cap \mathbb{C}_\Sigma(l)$, and $\boldsymbol{\omega}^* \in \boldsymbol{\omega}^{(k)} + \mathbb{B}_1(d)$.

It follows from a second-order Taylor expansion that

$$
\begin{aligned}
&\mathcal{T}(\boldsymbol{\omega}, \boldsymbol{\omega}^*) \\
&= \frac{1}{2}(\boldsymbol{\omega} - \boldsymbol{\omega}^*)^{\mathrm{T}} \nabla^2 \hat{Q}_h \big(t\boldsymbol{\omega} + (1-t)\boldsymbol{\omega}^*\big)(\boldsymbol{\omega} - \boldsymbol{\omega}^*) \\
&= \frac{1}{2(n_{\mathcal{A}_m} + n_0)} \sum_{k \in \mathcal{A}_m \cup \{0\}} \sum_{i=1}^{n_k} K_h \big\{ y_i^{(k)} - (\boldsymbol{x}_i^{(k)})^{\mathrm{T}}(t_i^{(k)}\boldsymbol{\omega} + (1-t_i^{(k)})\boldsymbol{\omega}^*) \big\} \big( (\boldsymbol{x}_i^{(k)})^{\mathrm{T}}(\boldsymbol{\omega} - \boldsymbol{\omega}^*) \big)^2 \\
&= \frac{1}{2(n_{\mathcal{A}_m} + n_0)} \sum_{k \in \mathcal{A}_m \cup \{0\}} \sum_{i=1}^{n_k} K_h \big\{ \epsilon_i - t_i^{(k)}(\boldsymbol{x}_i^{(k)})^{\mathrm{T}}(\boldsymbol{\omega} - \boldsymbol{\omega}^*) - (\boldsymbol{x}_i^{(k)})^{\mathrm{T}}(\boldsymbol{\omega}^* - \boldsymbol{\omega}^{(k)}) \big\} \\
&\quad \cdot \big( (\boldsymbol{x}_i^{(k)})^{\mathrm{T}}(\boldsymbol{\omega} - \boldsymbol{\omega}^*) \big)^2,
\end{aligned}
$$

for some $t_i^{(k)} \in [0,1]$. For each $i$ and $k$, define the event $\mathcal{F}_{i,k}$,

$$
\mathcal{F}_{i,k} = \big\{ |\epsilon_i| \le h/4 \big\} \cap \big\{ |(\boldsymbol{x}_i^{(k)})^{\mathrm{T}}(\boldsymbol{\omega} - \boldsymbol{\omega}^*)| \le \|\boldsymbol{\omega} - \boldsymbol{\omega}^*\|_{\Sigma} \cdot h/(2r) \big\} \cap \big\{ |(\boldsymbol{x}_i^{(k)})^{\mathrm{T}}(\boldsymbol{\omega}^* - \boldsymbol{\omega}^{(k)})| \le h/4 \big\},
$$

for all $\boldsymbol{\omega} - \boldsymbol{\omega}^* \in \mathbb{B}_{\Sigma}(r)$. Thus

$$
\mathcal{T}(\boldsymbol{\omega}, \boldsymbol{\omega}^*) \ge \frac{\underline{\kappa}}{2(n_{\mathcal{A}_m} + n_0)h} \sum_{k \in \mathcal{A}_m \cup \{0\}} \sum_{i=1}^{n_k} \big( (\boldsymbol{x}_i^{(k)})^{\mathrm{T}}(\boldsymbol{\omega} - \boldsymbol{\omega}^*) \big)^2 \mathbb{1}_{\mathcal{F}_{i,k}}, \tag{9}
$$

where $\underline{\kappa} = \min_{|u| \le 1} K(u)$. For a truncation level $R > 0$, define functions

$$
\varphi_R(u) = \begin{cases} u^2 & |u| \le \frac{R}{2}, \\ (R - |u|)^2 & \frac{R}{2} < |u| \le R, \\ 0 & |u| > R. \end{cases}
$$

By this construction, $\varphi_R(u) \le u^2 \cdot \mathbb{1}\{|u| \le R\}$, $\varphi_{cR}(cu) = c^2 \varphi_R(u)$ and $\varphi_R$ is R-Lipschitz. In addition, we define the trapezoidal function

$$
\psi_R(u) = \begin{cases} 1 & |u| \le \frac{R}{2}, \\ 2 - \frac{2}{R}|u| & \frac{R}{2} < |u| \le R, \\ 0 & |u| > R, \end{cases}
$$

and note that $\psi_R$ is $(2/R)$-Lipschitz and $\psi_R(u) \le \mathbb{1}\{|u| \le R\}$.

With the two new-defined function and the notation $\Delta = \boldsymbol{\omega} - \boldsymbol{\omega}^*$, $n = n_{\mathcal{A}_m} + n_0$, we have established the lower bound of (9)

$$
\begin{aligned}
&\mathcal{T}(\boldsymbol{\omega}, \boldsymbol{\omega}^*) \\
&\ge \frac{\underline{\kappa}}{2nh} \|\Delta\|_{\Sigma}^2 \sum_{k \in \mathcal{A}_m \cup \{0\}} \sum_{i=1}^{n_k} \mathbb{1}\{|\epsilon_i| \le h/4\} \varphi_{\|\Delta\|_{\Sigma} \cdot h/(2r)} \big( (\boldsymbol{x}_i^{(k)})^{\mathrm{T}} \Delta \big) \psi_{h/4} \big( (\boldsymbol{x}_i^{(k)})^{\mathrm{T}}(\boldsymbol{\omega}^* - \boldsymbol{\omega}^{(k)}) \big) \\
&\ge \frac{\underline{\kappa}}{2} \|\Delta\|_{\Sigma}^2 \cdot \underbrace{\frac{1}{nh} \sum_{k,i} \mathbb{1}\{|\epsilon_i| \le h/4\} \varphi_{h/(2r)} \big( (\boldsymbol{x}_i^{(k)})^{\mathrm{T}} \Delta / \|\Delta\|_{\Sigma} \big) \psi_{h/4} \big( (\boldsymbol{x}_i^{(k)})^{\mathrm{T}}(\boldsymbol{\omega}^* - \boldsymbol{\omega}^{(k)}) \big)}_{D_0(\boldsymbol{\omega}, \boldsymbol{\omega}^*)}
\end{aligned} \tag{10}
$$

In the following proofs, we bound $\mathbb{E}D_0(\boldsymbol{\omega}, \boldsymbol{\omega}^*)$ and $D_0(\boldsymbol{\omega}, \boldsymbol{\omega}^*) - \mathbb{E}D_0(\boldsymbol{\omega}, \boldsymbol{\omega}^*)$, respectively. First, we show that

$$
\mathbb{E}D_0(\boldsymbol{\omega}, \boldsymbol{\omega}^*) \ge 0.21\underline{f}. \tag{11}
$$

Note that

$$\left|\frac{h}{2}f_{\epsilon|\boldsymbol{x}}(0)\right| - \left|\mathbb{E}\big[\mathbb{1}\{|\epsilon_i| \le h/4\}|\boldsymbol{x}_i^{(k)}\big]\right| \le \left|\mathbb{E}\big[\mathbb{1}\{|\epsilon_i| \le h/4\}|\boldsymbol{x}_i^{(k)}\big] - \frac{h}{2}f_{\epsilon|\boldsymbol{x}}(0)\right|$$
$$\le \int_{-h/4}^{h/4} |f_{\epsilon|\boldsymbol{x}}(t) - f_{\epsilon|\boldsymbol{x}}(0)|dt$$
$$\le \frac{l_0 h^2}{16}.$$

Hence we obtain

$$\left|\mathbb{E}[\mathbb{1}\{|\epsilon_i| \le h/4\}|\boldsymbol{x}_i^{(k)}]\right| \ge \frac{h}{2}\underline{f} - \frac{l_0 h^2}{16}.$$

Provided $h \le \underline{f}/l_0 \le \bar{f}/l_0$, we have

$$\left|\mathbb{E}[\mathbb{1}\{|\epsilon_i| \le h/4\}|\boldsymbol{x}_i^{(k)}]\right| \ge \frac{7\underline{f}h}{16}.$$

Meanwhile

$$\left|\mathbb{E}[\mathbb{1}\{|\epsilon_i| \le h/4\}|\boldsymbol{x}_i^{(k)}]\right| - \left|\frac{h}{2}f_{\epsilon|\boldsymbol{x}}(0)\right| \le \int_{-h/4}^{h/4} |f_{\epsilon|\boldsymbol{x}}(t) - f_{\epsilon|\boldsymbol{x}}(0)|dt$$

implies

$$\left|\mathbb{E}[\mathbb{1}\{|\epsilon_i| \le h/4\}|\boldsymbol{x}_i^{(k)}]\right| \le \frac{9\bar{f}h}{16}.$$

Then

$$\mathbb{E}D_0(\boldsymbol{\omega}, \boldsymbol{\omega}^*)$$
$$= \frac{1}{nh}\sum_{k,i}\mathbb{E}\Big[\mathbb{E}\big[\mathbb{1}\{|\epsilon_i| \le h/4\}|\boldsymbol{x}_i^{(k)}\big]\varphi_{h/(2r)}\big((\boldsymbol{x}_i^{(k)})^{\mathrm{T}}\Delta/||\Delta||_\Sigma\big)\psi_{h/4}\big((\boldsymbol{x}_i^{(k)})^{\mathrm{T}}(\boldsymbol{\omega}^* - \boldsymbol{\omega}^{(k)}))\big)\Big]$$
$$\ge \frac{7\underline{f}}{16}\mathbb{E}\Big[\varphi_{h/(2r)}\big(\boldsymbol{x}^{\mathrm{T}}\Delta/||\Delta||_\Sigma\big)\psi_{h/4}\big(\boldsymbol{x}^{\mathrm{T}}(\boldsymbol{\omega}^* - \boldsymbol{\omega}^{(k)}))\big)\Big]$$
$$\ge \frac{7\underline{f}}{16}\mathbb{E}\Big[\big(\boldsymbol{x}^{\mathrm{T}}\Delta/||\Delta||_\Sigma\big)^2\mathbb{1}\{|\boldsymbol{x}^{\mathrm{T}}\Delta/||\Delta||_\Sigma| \le h/(4r)\} \cdot \psi_{h/4}\big(\boldsymbol{x}^{\mathrm{T}}(\boldsymbol{\omega}^* - \boldsymbol{\omega}^{(k)}))\big)\Big]$$
$$\ge \frac{7\underline{f}}{16}\Big\{1 - \mathbb{E}\Big[\big(\boldsymbol{x}^{\mathrm{T}}\Delta/||\Delta||_\Sigma\big)^2\mathbb{1}\{|\boldsymbol{x}^{\mathrm{T}}\Delta/||\Delta||_\Sigma| > h/(4r)\}\Big]$$
$$\qquad - \mathbb{E}\Big[\big(\boldsymbol{x}^{\mathrm{T}}\Delta/||\Delta||_\Sigma\big)^2\mathbb{1}\{|\boldsymbol{x}^{\mathrm{T}}(\boldsymbol{\omega}^* - \boldsymbol{\omega}^{(k)})| > h/8\}\Big]\Big\}$$
$$\ge \frac{7\underline{f}}{16}\Big\{1 - \mathbb{E}\Big[\big(\boldsymbol{x}^{\mathrm{T}}\Delta/||\Delta||_\Sigma\big)^2\mathbb{1}\{|\boldsymbol{x}^{\mathrm{T}}\Delta/||\Delta||_\Sigma| > h/(4r)\}\Big] - \mu_4^{1/2}\mathbb{P}\big(|\boldsymbol{x}^{\mathrm{T}}(\boldsymbol{\omega}^* - \boldsymbol{\omega}^{(k)})| > h/8\big)^{1/2}\Big\}.$$

By the definition of $\eta_\delta$, as long as $0 < r \le h/(4\eta_{1/4})$,

$$\sup_{\Delta \in \mathbb{B}_\Sigma(r)}\mathbb{E}\Big[\big(\boldsymbol{x}^{\mathrm{T}}\Delta/||\Delta||_\Sigma\big)^2\mathbb{1}\{|\boldsymbol{x}^{\mathrm{T}}\Delta/||\Delta||_\Sigma| > h/(4r)\}\Big] \le \frac{1}{4}.$$

Moreover, $\boldsymbol{\omega}^* \in \boldsymbol{\omega}^{(k)} + \mathbb{B}_1(d)$. Hence

$$\big\|\Sigma^{1/2}(\boldsymbol{\omega}^* - \boldsymbol{\omega}^{(k)})\big\|_2 \le \big\|\Sigma^{1/2}\big\|_2\big\|\boldsymbol{\omega}^* - \boldsymbol{\omega}^{(k)}\big\|_2 \le \gamma_1^{1/2}d.$$

Under Assumption 3.3 with $v_0 \ge 1$, the tail bounds of sub-exponential $\boldsymbol{z} = \Sigma^{-1/2}\boldsymbol{x}$ implys that

$$\mathbb{P}\big(|\boldsymbol{x}^{\mathrm{T}}(\boldsymbol{\omega}^* - \boldsymbol{\omega}^{(k)})| > h/8\big) \le 2\exp\Big\{-\frac{h}{8v_0\gamma_1^{1/2}d}\Big\}.$$

Let $h \geq 32v_0\gamma_1^{1/2}d\mu_4^{1/2}$. It then follows from a numerical calculation that

$$\left\{1 - \mathbb{E}\left[(\boldsymbol{x}^{\mathrm{T}}\Delta/||\Delta||_\Sigma)^2 \mathbb{1}\{|\boldsymbol{x}^{\mathrm{T}}\Delta/||\Delta||_\Sigma| > h/(4r)\}\right] - \mu_4^{1/2}\mathbb{P}(|\boldsymbol{x}^{\mathrm{T}}(\boldsymbol{\omega}^* - \boldsymbol{\omega}^{(k)})| > h/8)^{1/2}\right\} \geq 0.49$$

holds uniformly over $\Delta \in \mathbb{B}_\Sigma(r) \cap \mathbb{C}_\Sigma(l)$. Putting together the pieces yields

$$\mathbb{E}D_0(\boldsymbol{\omega}, \boldsymbol{\omega}^*) > 0.21\underline{f}.$$

Next we find a lower bound of $D_0(\boldsymbol{\omega}, \boldsymbol{\omega}^*) - \mathbb{E}D_0(\boldsymbol{\omega}, \boldsymbol{\omega}^*)$ over $\boldsymbol{\omega} \in \boldsymbol{\omega}^* + \mathbb{B}_\Sigma(r) \cap \mathbb{C}_\Sigma(l)$. Define

$$\Omega(r, l) = \sup_{\boldsymbol{\omega} \in \boldsymbol{\omega}^* + \mathbb{B}_\Sigma(r) \cap \mathbb{C}_\Sigma(l)} \{-D_0(\boldsymbol{\omega}, \boldsymbol{\omega}^*) + \mathbb{E}D_0(\boldsymbol{\omega}, \boldsymbol{\omega}^*)\}.$$

Write $D_0(\boldsymbol{\omega}, \boldsymbol{\omega}^*) = n^{-1} \sum_{k \in \mathcal{A}_m \cup \{0\}} \sum_{i=1}^{n_k} w_{i,k}(\boldsymbol{\omega}, \boldsymbol{\omega}^*)$, where

$$w_{i,k}(\boldsymbol{\omega}, \boldsymbol{\omega}^*) = h^{-1}\mathbb{1}\{|\epsilon_i| \leq h/4\}\varphi_{h/(2r)}\big((\boldsymbol{x}_i^{(k)})^{\mathrm{T}}\Delta/||\Delta||_\Sigma\big)\psi_{h/4}\big((\boldsymbol{x}_i^{(k)})^{\mathrm{T}}(\boldsymbol{\omega}^* - \boldsymbol{\omega}^{(k)})\big)$$

satisfies $0 \leq w_{i,k}(\boldsymbol{\omega}, \boldsymbol{\omega}^*) \leq h/(4r)^2$, since $\varphi_R(u) \leq (R/2)^2$ and $\psi_R(u) \in [0, 1]$. Moreover,

$$\begin{aligned}
\mathbb{E}w_{i,k}^2(\boldsymbol{\omega}, \boldsymbol{\omega}^*) &= \mathbb{E}\left[h^{-2}\mathbb{1}\{|\epsilon_i| \leq h/4\}\varphi_{h/(2r)}^2\big((\boldsymbol{x}_i^{(k)})^{\mathrm{T}}\Delta/||\Delta||_\Sigma\big)\psi_{h/4}^2\big((\boldsymbol{x}_i^{(k)})^{\mathrm{T}}(\boldsymbol{\omega}^* - \boldsymbol{\omega}^{(k)})\big)\right] \\
&\leq \frac{9\bar{f}}{16h} \cdot \mathbb{E}\varphi_{h/(2r)}^2\big((\boldsymbol{x}_i^{(k)})^{\mathrm{T}}\Delta/||\Delta||_\Sigma\big) \\
&\leq \frac{9\bar{f}}{16h} \cdot \mathbb{E}\big((\boldsymbol{x}_i^{(k)})^{\mathrm{T}}\Delta/||\Delta||_\Sigma\big)^4 = \frac{9\bar{f}\mu_4}{16h}.
\end{aligned}$$

Using Bousquet's version of Talagrand's inequality yields that, for any $z > 0$,

$$\begin{aligned}
\Omega(r, l) &\leq \mathbb{E}\Omega(r, l) + \{\mathbb{E}\Omega(r, l)\}^{1/2}\frac{1}{2r}\sqrt{\frac{hz}{n}} + \frac{3\sqrt{2}\mu_4^{1/2}}{4}\sqrt{\frac{\bar{f}z}{nh}} + \frac{h}{(4r)^2}\frac{z}{3n} \\
&\leq \mathbb{E}\Omega(r, l) + \frac{1}{4}\mathbb{E}\Omega(r, l) + \frac{1}{4r^2}\frac{hz}{n} + \frac{3\sqrt{2}\mu_4^{1/2}}{4}\sqrt{\frac{\bar{f}z}{nh}} + \frac{h}{(4r)^2}\frac{z}{3n} \\
&\leq \frac{5}{4}\mathbb{E}\Omega(r, l) + \frac{3\sqrt{2}\mu_4^{1/2}}{4}\sqrt{\frac{\bar{f}z}{nh}} + \frac{13}{3}\frac{hz}{(4r)^2n}
\end{aligned} \qquad (12)$$

holds with probability at least $1 - e^{-z}$. To bound the expectation $\mathbb{E}\Omega(r, l)$, using Rademacher symmetrization and the connection between Gaussian and Rademacher complexities, Lemma 5.5 in Ledoux & Talagrand (1991), we have

$$\mathbb{E}\Omega(r, l) \leq 2\sqrt{\frac{\pi}{2}}\mathbb{E}\left[\sup_{(\boldsymbol{\omega}, \boldsymbol{\omega}^*) \in \Lambda(r, l)} \mathbb{G}_{\boldsymbol{\omega}, \boldsymbol{\omega}^*}\right], \qquad (13)$$

where

$$\begin{aligned}
&\mathbb{G}_{\boldsymbol{\omega}, \boldsymbol{\omega}^*} \\
&:= \frac{1}{nh} \sum_{k \in \mathcal{A}_m \cup \{0\}} \sum_{i=1}^{n_k} e_i \mathbb{1}\{|\epsilon_i| \leq h/4\}\varphi_{h/(2r)}\big((\boldsymbol{x}_i^{(k)})^{\mathrm{T}}\Delta/||\Delta||_\Sigma\big)\psi_{h/4}\big((\boldsymbol{x}_i^{(k)})^{\mathrm{T}}(\boldsymbol{\omega}^* - \boldsymbol{\omega}^{(k)})\big),
\end{aligned}$$

and $e_i$ are independent standard normal variables. Note that $\mathbb{G}_{\boldsymbol{\omega}, \boldsymbol{\omega}^*}$ is a Gaussian process conditioned on $\{(y_i^{(k)}, \boldsymbol{x}_i^{(k)})\}_{i=1}^{n_k}$ for $k \in \mathcal{A}_m \cup \{0\}$. For $(\boldsymbol{\omega}, \boldsymbol{\omega}^*)$ and $(\boldsymbol{\omega}', \boldsymbol{\omega}'^*)$, write $\Delta = \boldsymbol{\omega} - \boldsymbol{\omega}^*$, $\Delta' = \boldsymbol{\omega}' - \boldsymbol{\omega}'^*$ and

$\chi_i = \mathbb{1}\{|\epsilon_i| \le h/4\}$, then

$$
\begin{aligned}
&\mathbb{G}_{\boldsymbol{\omega},\boldsymbol{\omega}^*} - \mathbb{G}_{\boldsymbol{\omega}',\boldsymbol{\omega}'^*} \\
&= \mathbb{G}_{\boldsymbol{\omega},\boldsymbol{\omega}^*} - \mathbb{G}_{\boldsymbol{\omega}',\boldsymbol{\omega}'+\Delta} + \mathbb{G}_{\boldsymbol{\omega}',\boldsymbol{\omega}'+\Delta} - \mathbb{G}_{\boldsymbol{\omega}',\boldsymbol{\omega}'^*} \\
&= \frac{1}{nh} \sum_{k,i} e_i \chi_i \varphi_{h/(2r)}\big((\boldsymbol{x}_i^{(k)})^{\mathrm{T}} \Delta / \|\Delta\|_{\Sigma}\big) \big\{ \psi_{h/4}\big((\boldsymbol{x}_i^{(k)})^{\mathrm{T}}(\boldsymbol{\omega}^* - \boldsymbol{\omega}^{(k)})\big) \\
&\qquad - \psi_{h/4}\big((\boldsymbol{x}_i^{(k)})^{\mathrm{T}}(\boldsymbol{\omega}'^* - \boldsymbol{\omega}^{(k)})\big) \big\} \\
&\quad + \frac{1}{nh} \sum_{k,i} e_i \chi_i \psi_{h/4}\big((\boldsymbol{x}_i^{(k)})^{\mathrm{T}}(\boldsymbol{\omega}'^* - \boldsymbol{\omega}^{(k)})\big) \big\{ \varphi_{h/(2r)}\big((\boldsymbol{x}_i^{(k)})^{\mathrm{T}} \Delta / \|\Delta\|_{\Sigma}\big) \\
&\qquad - \varphi_{h/(2r)}\big((\boldsymbol{x}_i^{(k)})^{\mathrm{T}} \Delta' / \|\Delta'\|_{\Sigma}\big) \big\}.
\end{aligned}
$$

Note that $\varphi_R$ and $\psi_R$ are Lipschitz continuous, and $\varphi_R(u) \le (R/2)^2$. Let $\mathbb{E}^*$ be the conditional expectation given $\{(y_i^{(k)}, \boldsymbol{x}_i^{(k)})\}_{i=1}^{n_k}$. Consequently,

$$
\begin{aligned}
\mathbb{E}^* \big(\mathbb{G}_{\boldsymbol{\omega},\boldsymbol{\omega}^*} - \mathbb{G}_{\boldsymbol{\omega}',\boldsymbol{\omega}'+\Delta}\big)^2 &\le \frac{1}{(nh)^2} \left(\frac{8}{h}\right)^2 \left(\frac{h}{4r}\right)^4 \sum_{k \in \mathcal{A}_m \cup \{0\}} \sum_{i=1}^{n_k} \chi_i \big((\boldsymbol{x}_i^{(k)})^{\mathrm{T}}(\boldsymbol{\omega} - \boldsymbol{\omega}')\big)^2 \\
&= \frac{1}{4r^4 n^2} \sum_{k \in \mathcal{A}_m \cup \{0\}} \sum_{i=1}^{n_k} \chi_i \big((\boldsymbol{x}_i^{(k)})^{\mathrm{T}}(\boldsymbol{\omega} - \boldsymbol{\omega}')\big)^2
\end{aligned}
\tag{14}
$$

and

$$
\mathbb{E}^* \big(\mathbb{G}_{\boldsymbol{\omega}',\boldsymbol{\omega}'+\Delta} - \mathbb{G}_{\boldsymbol{\omega}',\boldsymbol{\omega}'^*}\big)^2
\tag{15}
$$

$$
\le \frac{1}{(nh)^2} \left(\frac{h}{2r}\right)^2 \sum_{k \in \mathcal{A}_m \cup \{0\}} \sum_{i=1}^{n_k} \chi_i \big\{ (\boldsymbol{x}_i^{(k)})^{\mathrm{T}} \Delta / \|\Delta\|_{\Sigma} - (\boldsymbol{x}_i^{(k)})^{\mathrm{T}} \Delta' / \|\Delta'\|_{\Sigma} \big\}^2
$$

$$
= \frac{1}{4r^2 n^2} \sum_{k \in \mathcal{A}_m \cup \{0\}} \sum_{i=1}^{n_k} \chi_i \big\{ (\boldsymbol{x}_i^{(k)})^{\mathrm{T}} (\Delta / \|\Delta\|_{\Sigma} - \Delta' / \|\Delta'\|_{\Sigma}) \big\}^2.
\tag{16}
$$

Motivated by the last two inequalities, we have

$$
\begin{aligned}
\mathbb{E}^* \big(\mathbb{G}_{\boldsymbol{\omega},\boldsymbol{\omega}^*} - \mathbb{G}_{\boldsymbol{\omega}',\boldsymbol{\omega}'^*}\big)^2 &\le \frac{1}{2r^4 n^2} \sum_{k \in \mathcal{A}_m \cup \{0\}} \sum_{i=1}^{n_k} \chi_i \big((\boldsymbol{x}_i^{(k)})^{\mathrm{T}}(\boldsymbol{\omega} - \boldsymbol{\omega}')\big)^2 \\
&\quad + \frac{1}{2r^2 n^2} \sum_{k \in \mathcal{A}_m \cup \{0\}} \sum_{i=1}^{n_k} \chi_i \big\{ (\boldsymbol{x}_i^{(k)})^{\mathrm{T}} (\Delta / \|\Delta\|_{\Sigma} - \Delta' / \|\Delta'\|_{\Sigma}) \big\}^2.
\end{aligned}
$$

Define another Gaussian process $\mathbb{Z}_{\boldsymbol{\omega},\boldsymbol{\omega}^*}$ as

$$
\begin{aligned}
\mathbb{Z}_{\boldsymbol{\omega},\boldsymbol{\omega}^*} &= \frac{1}{2^{1/2} r^2 n} \sum_{k \in \mathcal{A}_m \cup \{0\}} \sum_{i=1}^{n_k} e_i' \chi_i (\boldsymbol{x}_i^{(k)})^{\mathrm{T}}(\boldsymbol{\omega}^* - \boldsymbol{\omega}^{(k)}) \\
&\quad + \frac{1}{2^{1/2} r n} \sum_{k \in \mathcal{A}_m \cup \{0\}} \sum_{i=1}^{n_k} e_i'' \chi_i (\boldsymbol{x}_i^{(k)})^{\mathrm{T}} \Delta / \|\Delta\|_{\Sigma}
\end{aligned}
$$

such that $\mathbb{E}^*(\mathbb{G}_{\boldsymbol{\omega},\boldsymbol{\omega}^*} - \mathbb{G}_{\boldsymbol{\omega}',\boldsymbol{\omega}'^*})^2 \le \mathbb{E}^*(\mathbb{Z}_{\boldsymbol{\omega},\boldsymbol{\omega}^*} - \mathbb{Z}_{\boldsymbol{\omega}',\boldsymbol{\omega}'^*})^2$, where $\{e_i'\}$ and $\{e_i''\}$ are two dependent copies of $\{e_i\}$. Applying Theorem 7.2.11 in Vershynin (2018), we obtain

$$
\mathbb{E}^* \Big( \sup_{\boldsymbol{\omega},\boldsymbol{\omega}^*} \mathbb{G}_{\boldsymbol{\omega},\boldsymbol{\omega}^*} \Big) \le \mathbb{E}^* \Big( \sup_{\boldsymbol{\omega},\boldsymbol{\omega}^*} \mathbb{Z}_{\boldsymbol{\omega},\boldsymbol{\omega}^*} \Big).
\tag{17}
$$

To bound the supremum of $\mathbb{Z}_{\boldsymbol{\omega},\boldsymbol{\omega}^*}$, using the cone constraint and $\|\boldsymbol{\omega}^* - \boldsymbol{\omega}^{(k)}\|_1 \le d$, we have

$$
\mathbb{E}^*\left(\sup_{\boldsymbol{\omega},\boldsymbol{\omega}^*} \mathbb{Z}_{\boldsymbol{\omega},\boldsymbol{\omega}^*}\right) \le \frac{\sqrt{2}d}{2r^2}\mathbb{E}\left\|\frac{1}{n}\sum_{k\in\mathcal{A}_m\cup\{0\}}\sum_{i=1}^{n_k} e'_i\chi_i \boldsymbol{x}_i^{(k)}\right\|_\infty
$$
$$
+ \frac{\sqrt{2}l}{2r}\mathbb{E}\left\|\frac{1}{n}\sum_{k\in\mathcal{A}_m\cup\{0\}}\sum_{i=1}^{n_k} e''_i\chi_i \boldsymbol{x}_i^{(k)}\right\|_\infty. \tag{18}
$$

Thus, by (13) (17) and (18), we have

$$
\mathbb{E}\Omega(r,l) \le \sqrt{\pi}\left\{\frac{d}{r^2}\mathbb{E}\left\|\frac{1}{n}\sum_{k\in\mathcal{A}_m\cup\{0\}}\sum_{i=1}^{n_k} e'_i\chi_i \boldsymbol{x}_i^{(k)}\right\|_\infty + \frac{l}{r}\mathbb{E}\left\|\frac{1}{n}\sum_{k\in\mathcal{A}_m\cup\{0\}}\sum_{i=1}^{n_k} e''_i\chi_i \boldsymbol{x}_i^{(k)}\right\|_\infty\right\}. \tag{19}
$$

It remains to find the bound of the two $\ell_\infty$-norm terms on the right-hand side of (19). Note that the variable $|n^{-1}\sum_{k\in\mathcal{A}_m\cup\{0\}}\sum_{i=1}^{n_k} e'_i\chi_i x_{ij}^{(k)}|$ is zero-mean for $j = 1,\ldots,p$.

$$
\mathbb{E}\left[\exp\left(\lambda\left|\frac{1}{n}\sum_k\sum_{i=1}^{n_k} e'_i\chi_i x_{ij}^{(k)}\right|\right)\right] \le \prod_k\prod_{i=1}^{n_k}\mathbb{E}\left[\exp\left(\left|\frac{\lambda e'_i\chi_i x_{ij}^{(k)}}{n}\right|\right)\right]
$$
$$
\le \prod_k\prod_{i=1}^{n_k}\exp\left\{\left|\frac{\lambda^2\chi_i^2(x_{ij}^{(k)})^2}{2n^2}\right|\right\}
$$
$$
= \exp\left\{\frac{\lambda^2}{2n^2}\sum_k\sum_{i=1}^{n_k}\chi_i^2(x_{ij}^{(k)})^2\right\}.
$$

Thus, $|n^{-1}\sum_{k\in\mathcal{A}_m\cup\{0\}}\sum_{i=1}^{n_k} e'_i\chi_i x_{ij}^{(k)}|$ is sub-Gaussian with parameter $n^{-1}\sqrt{\sum_k\sum_{i=1}^{n_k}\chi_i^2(x_{ij}^{(k)})^2}$.

Applying Lemma 15 in Loh & Wainwright (2015), we have

$$
\mathbb{E}\left[\left\|\frac{1}{n}\sum_k\sum_{i=1}^{n_k} e'_i\chi_i \boldsymbol{x}_i^{(k)}\right\|_\infty \middle| \boldsymbol{x}_i^{(k)}\right] \le \frac{c}{n}\cdot\max_{j=1,\ldots,p}\sqrt{\sum_k\sum_{i=1}^{n_k}\chi_i^2(x_{ij}^{(k)})^2}\cdot\sqrt{\log p}
$$
$$
\le \frac{c\sqrt{\log p}}{n}\sqrt{\sum_k\sum_{i=1}^{n_k}\chi_i^2(x_{ij}^{(k)})^2},
$$

implying that

$$
\mathbb{E}\left\|\frac{1}{n}\sum_k\sum_{i=1}^{n_k} e'_i\chi_i \boldsymbol{x}_i^{(k)}\right\|_\infty \le c\sqrt{\frac{\log p}{n}}\cdot\mathbb{E}\left[\sqrt{\frac{\sum_k\sum_{i=1}^{n_k}\chi_i^2(x_{ij}^{(k)})^2}{n}}\right]
$$
$$
\le c\sqrt{\frac{\log p}{n}}\cdot\sqrt{\mathbb{E}\left[\frac{\sum_k\sum_{i=1}^{n_k}\chi_i^2(x_{ij}^{(k)})^2}{n}\right]}
$$
$$
\le c\sigma_x\sqrt{\frac{\log p}{n}}\cdot\sqrt{\frac{9\bar{f}h}{16}} \tag{20}
$$

Similarly,

$$
\mathbb{E}\left\|\frac{1}{n}\sum_k\sum_{i=1}^{n_k} e''_i\chi_i \boldsymbol{x}_i^{(k)}\right\|_\infty \le c\sigma_x\sqrt{\frac{\log p}{n}}\cdot\sqrt{\frac{9\bar{f}h}{16}}. \tag{21}
$$

Finally, if we take $r = h/(4\eta_{1/4})$, $d = h(\gamma_1\mu_4)^{-1/2}/(32v_0)$ and $z = t + \log p$, combining (19), (20), (21) with Bousquet's version inequality (12), we conclude that

$$
\Omega(r,l) \le 0.01\underline{f} + c'l\sqrt{(t+\log p)/(nh)}
$$

with probability at least $1 - p^{-1}e^{-t}$ for any $t > 0$, as long as

$$nh \gtrsim \bar{f}\underline{f}^{-2}\eta_{1/4}^2\mu_4\sigma_x^2\log(p).$$

This, together with (9), (10) and (11), we have

$$\mathcal{T}(\boldsymbol{\omega}, \boldsymbol{\omega}^*) \geq \frac{\kappa}{2}||\Delta||_{\Sigma}^2\left[0.21\underline{f} - \left(0.01\underline{f} + c'l\sqrt{\frac{t + \log p}{nh}}\right)\right]$$

$$\geq \frac{\kappa}{2}||\Delta||_{\Sigma}^2\left(0.2\underline{f} - c'l\sqrt{\frac{t + \log p}{nh}}\right) \tag{22}$$

with probability at least $1 - p^{-1}e^{-t}$.

It remains to extend this bound to one that is uniform in the ratio $||\Delta||_1/||\Delta||_{\Sigma}$, which we do via a peeling argument. Consider the inequality

$$\frac{1}{||\Delta||_{\Sigma}^2}\mathcal{T}(\boldsymbol{\omega}, \boldsymbol{\omega}^*) \geq \frac{\kappa \cdot \underline{f}}{10} - \frac{2c''||\Delta||_1}{||\Delta||_{\Sigma}}\sqrt{\frac{t + \log p}{nh}}. \tag{23}$$

Since

$$\gamma_1^{-1/2} \leq \frac{||\Delta||_1}{||\Delta||_{\Sigma}} \leq \frac{\kappa \cdot \underline{f}}{20c''}\sqrt{\frac{nh}{t + \log p}} := \zeta,$$

we define the set

$$\Theta_k = \left\{\Delta \in \mathbb{R}^p : \gamma_1^{-1/2}2^{k-1} \leq ||\Delta||_1/||\Delta||_{\Sigma} \leq \gamma_1^{-1/2}2^k\right\},$$

for each $k = 1, \ldots, N := \lceil\log_2(c\sqrt{nh/\log p})\rceil$ to let

$$\left\{\Delta \in \mathbb{R}^p : \gamma_1^{-1/2} \leq \frac{||\Delta||_1}{||\Delta||_{\Sigma}} \leq \zeta\right\} \subseteq \cup_{k=1}^N \Theta_k.$$

By the union bound, we then have

$$\mathbb{P}\left\{\exists\Delta \in \left\{\Delta \in \mathbb{R}^p : \gamma_1^{-1/2} \leq ||\Delta||_1/||\Delta||_{\Sigma} \leq \zeta\right\} \text{ s.t. } \frac{\kappa \cdot \underline{f}}{10} - \frac{1}{||\Delta||_{\Sigma}^2}\mathcal{T}(\boldsymbol{\omega}, \boldsymbol{\omega}^*) > \frac{2c''||\Delta||_1}{||\Delta||_{\Sigma}}\sqrt{\frac{t + \log p}{nh}}\right\}$$

$$\leq \sum_{k=1}^N \mathbb{P}\left\{\exists\Delta \in \Theta_k \text{ s.t. } \frac{\kappa \cdot \underline{f}}{10} - \frac{1}{||\Delta||_{\Sigma}^2}\mathcal{T}(\boldsymbol{\omega}, \boldsymbol{\omega}^*) > c''\gamma_1^{-1/2}2^k\sqrt{\frac{t + \log p}{nh}}\right\}$$

$$\leq \sum_{k=1}^N \mathbb{P}\left\{\sup_{||\Delta||_1/||\Delta||_{\Sigma} \leq \gamma_1^{-1/2}2^k}\frac{\kappa \cdot \underline{f}}{10} - \frac{1}{||\Delta||_{\Sigma}^2}\mathcal{T}(\boldsymbol{\omega}, \boldsymbol{\omega}^*) > c''\gamma_1^{-1/2}2^k\sqrt{\frac{t + \log p}{nh}}\right\}$$

$$\leq \sum_{k=1}^N p^{-1}e^{-t} \leq N \cdot p^{-1}e^{-t}.$$

Taking $t = \log\{\log_2(c\sqrt{nh/\log p})\} + u$ yields that with probability at least $1 - p^{-1}e^{-u}$,

$$\frac{\kappa \cdot \underline{f}}{10} - \frac{1}{||\Delta||_{\Sigma}^2}\mathcal{T}(\boldsymbol{\omega}, \boldsymbol{\omega}^*) \leq \frac{2c''||\Delta||_1}{||\Delta||_{\Sigma}}\sqrt{\frac{\log p + \log\{\log_2(c\sqrt{nh/\log p})\} + u}{nh}}.$$

Multiplying by $||\Delta||_{\Sigma}^2$ on both sides yields

$$\mathcal{T}(\boldsymbol{\omega}, \boldsymbol{\omega}^*) \geq \frac{\kappa \cdot \underline{f}}{10}||\Delta||_{\Sigma}^2 - 2c''||\Delta||_1||\Delta||_{\Sigma}\sqrt{\frac{\log p + \log\{\log_2(c\sqrt{nh/\log p})\} + u}{nh}},$$

where $c'' > 0$ is a constant depending only on $(\kappa, \underline{f})$.

## A.3 Proof of Proposition 3.2

The Taylor error around $\boldsymbol{\beta}_2$ in the direction $\boldsymbol{\beta}_1 - \boldsymbol{\beta}_2$ is given by

$$\mathcal{T}(\boldsymbol{\beta}_1, \boldsymbol{\beta}_2) = \hat{Q}_g^{(0)}(\boldsymbol{\beta}_1) - \hat{Q}_g^{(0)}(\boldsymbol{\beta}_2) - \left(\nabla \hat{Q}_g^{(0)}(\boldsymbol{\beta}_2)\right)^{\mathrm{T}}(\boldsymbol{\beta}_1 - \boldsymbol{\beta}_2).$$

For a given kernel function $K(\cdot)$ and bandwidth $g > 0$, the smoothed quantile loss $\hat{Q}_g^{(0)}$ can be written as $(n_0 g)^{-1} \sum_{i=1}^{n_0} \int_{-\infty}^{\infty} \rho_\tau(u) K\{(u + (\boldsymbol{x}_i^{(0)})^{\mathrm{T}}\boldsymbol{\beta} - y_i^{(0)})/g\} du$. Therefore

$$\mathcal{T}(\boldsymbol{\beta}_1, \boldsymbol{\beta}_2) = \frac{1}{2n_0} \sum_{i=1}^{n_0} K_g\{\epsilon_i - (\boldsymbol{x}_i^{(0)})^{\mathrm{T}}((1-t)\boldsymbol{\beta}_1 + t\boldsymbol{\beta}_2 - \boldsymbol{\beta}^*)\}\{(\boldsymbol{x}_i^{(0)})^{\mathrm{T}}(\boldsymbol{\beta}_1 - \boldsymbol{\beta}_2)\}^2$$

$$= \frac{1}{2n_0} \sum_{i=1}^{n_0} K_g\{\epsilon_i - t(\boldsymbol{x}_i^{(0)})^{\mathrm{T}}(\boldsymbol{\beta}_2 - \boldsymbol{\beta}_1) - (\boldsymbol{x}_i^{(0)})^{\mathrm{T}}(\boldsymbol{\beta}_1 - \boldsymbol{\beta}^*)\}\{(\boldsymbol{x}_i^{(0)})^{\mathrm{T}}(\boldsymbol{\beta}_1 - \boldsymbol{\beta}_2)\}^2,$$

for some $t \in [0, 1]$, For each $i$, define the event $\mathcal{E}_i$,

$$\mathcal{E}_i = \{|\epsilon_i| \le g/4\} \cap \{|(\boldsymbol{x}_i^{(0)})^{\mathrm{T}}(\boldsymbol{\beta}_1 - \boldsymbol{\beta}_2)| \le g\|\boldsymbol{\beta}_1 - \boldsymbol{\beta}_2\|_\Sigma/(2r)\} \cap \{|(\boldsymbol{x}_i^{(0)})^{\mathrm{T}}(\boldsymbol{\beta}_1 - \boldsymbol{\beta}^*)| \le g/4\},$$

for all $\boldsymbol{\beta}_1 - \boldsymbol{\beta}_2 \in \mathbb{B}_\Sigma(r)$. Thus

$$\mathcal{T}(\boldsymbol{\beta}_1, \boldsymbol{\beta}_2) \ge \frac{\kappa}{2n_0 g} \sum_{i=1}^{n_0} \{(\boldsymbol{x}_i^{(0)})^{\mathrm{T}}\boldsymbol{\delta}\}^2 \mathbb{1}_{\mathcal{E}_i}, \tag{24}$$

where $\boldsymbol{\delta} = \boldsymbol{\beta}_1 - \boldsymbol{\beta}_2$. For a truncation level $R > 0$, define functions

$$\varphi_R(u) = \begin{cases} u^2 & |u| \le \frac{R}{2}, \\ (R - |u|)^2 & \frac{R}{2} < |u| \le R, \\ 0 & |u| > R. \end{cases}$$

By this construction, $\varphi_R(u) \le u^2 \cdot \mathbb{1}\{|u| \le R\}$, $\varphi_{cR}(cu) = c^2 \varphi_R(u)$ and $\varphi_R$ is $R$-Lipschitz. In addition, we define the trapezoidal function

$$\psi_R(u) = \begin{cases} 1 & |u| \le \frac{R}{2}, \\ 2 - \frac{2}{R}|u| & \frac{R}{2} < |u| \le R, \\ 0 & |u| > R, \end{cases}$$

and note that $\psi_R$ is $(2/R)$-Lipschitz and $\psi_R(u) \le \mathbb{1}\{|u| \le R\}$.
From these two new-defined function, (24) implies

$$\mathcal{T}(\boldsymbol{\beta}_1, \boldsymbol{\beta}_2) \ge \frac{\kappa}{2n_0 g}\|\boldsymbol{\delta}\|_\Sigma^2 \sum_{i=1}^{n_0} \mathbb{1}\{|\epsilon_i| \le g/4\} \varphi_{g\|\boldsymbol{\delta}\|_\Sigma/(2r)}\left((\boldsymbol{x}_i^{(0)})^{\mathrm{T}}\boldsymbol{\delta}\right) \psi_{g/4}\left((\boldsymbol{x}_i^{(0)})^{\mathrm{T}}(\boldsymbol{\beta}_1 - \boldsymbol{\beta}^*)\right)$$

$$\ge \frac{\kappa}{2}\|\boldsymbol{\delta}\|_\Sigma^2 \cdot \underbrace{\frac{1}{n_0 g} \sum_{i=1}^{n_0} \mathbb{1}\{|\epsilon_i| \le g/4\} \varphi_{g/(2r)}\left((\boldsymbol{x}_i^{(0)})^{\mathrm{T}}\boldsymbol{\delta}/\|\boldsymbol{\delta}\|_\Sigma\right) \psi_{g/4}\left((\boldsymbol{x}_i^{(0)})^{\mathrm{T}}(\boldsymbol{\beta}_1 - \boldsymbol{\beta}^*)\right)}_{D_0(\boldsymbol{\beta}_1, \boldsymbol{\beta}_2)} \tag{25}$$

In the following proofs, we bound $\mathbb{E}D_0(\boldsymbol{\beta}_1, \boldsymbol{\beta}_2)$ and $D_0(\boldsymbol{\beta}_1, \boldsymbol{\beta}_2) - \mathbb{E}D_0(\boldsymbol{\beta}_1, \boldsymbol{\beta}_2)$, respectively. Note that

$$\left|\frac{g}{2}f_{\epsilon|\boldsymbol{x}}(0)\right| - \left|\mathbb{E}[\mathbb{1}\{|\epsilon_i| \le g/4\}|\boldsymbol{x}_i^{(0)}]\right| \le \left|\mathbb{E}[\mathbb{1}\{|\epsilon_i| \le g/4\}|\boldsymbol{x}_i^{(0)}] - \frac{g}{2}f_{\epsilon|\boldsymbol{x}}(0)\right|$$

$$\le \int_{-g/4}^{g/4} |f_{\epsilon|\boldsymbol{x}}(t) - f_{\epsilon|\boldsymbol{x}}(0)| dt.$$

Then,

$$\left|\frac{g}{2}f_{\epsilon|\boldsymbol{x}}(0)\right| - \left|\mathbb{E}[\mathbb{1}\{|\epsilon_i| \leq g/4\}|\boldsymbol{x}_i^{(0)}]\right| \leq \frac{l_0 g^2}{16}$$

$$\left|\mathbb{E}[\mathbb{1}\{|\epsilon_i| \leq g/4\}|\boldsymbol{x}_i^{(0)}]\right| \geq \frac{g}{2}\underline{f} - \frac{l_0 g^2}{16}.$$

Provided $g \leq \underline{f}/l_0 \leq \bar{f}/l_0$, we have

$$\left|\mathbb{E}[\mathbb{1}\{|\epsilon_i| \leq g/4\}|\boldsymbol{x}_i^{(0)}]\right| \geq \frac{7\underline{f}g}{16}.$$

Meanwhile

$$\left|\mathbb{E}[\mathbb{1}\{|\epsilon_i| \leq g/4\}|\boldsymbol{x}_i^{(0)}]\right| - \left|\frac{g}{2}f_{\epsilon|\boldsymbol{x}}(0)\right| \leq \int_{-g/4}^{g/4} |f_{\epsilon|\boldsymbol{x}}(t) - f_{\epsilon|\boldsymbol{x}}(0)|dt$$

implies

$$\left|\mathbb{E}[\mathbb{1}\{|\epsilon_i| \leq g/4\}|\boldsymbol{x}_i^{(0)}]\right| \leq \frac{9\bar{f}g}{16}.$$

Then

$$\mathbb{E}D_0(\boldsymbol{\beta}_1, \boldsymbol{\beta}_2)$$
$$= \frac{1}{n_0 g} \sum_{i=1}^{n_0} \mathbb{E}\left[\mathbb{E}[\mathbb{1}\{|\epsilon_i| \leq g/4\}|\boldsymbol{x}_i^{(0)}]\varphi_{g/(2r)}\big((\boldsymbol{x}_i^{(0)})^{\mathrm{T}}\boldsymbol{\delta}/\|\boldsymbol{\delta}\|_\Sigma\big)\psi_{g/4}\big((\boldsymbol{x}_i^{(0)})^{\mathrm{T}}(\boldsymbol{\beta}_1 - \boldsymbol{\beta}^*)\big)\right]$$
$$\geq \frac{7\underline{f}}{16}\mathbb{E}\left[\varphi_{g/(2r)}(\boldsymbol{x}^{\mathrm{T}}\boldsymbol{\delta}/\|\boldsymbol{\delta}\|_\Sigma)\psi_{g/4}\big(\boldsymbol{x}^{\mathrm{T}}(\boldsymbol{\beta}_1 - \boldsymbol{\beta}^*)\big)\right]$$
$$\geq \frac{7\underline{f}}{16}\left\{1 - \mathbb{E}\left[(\boldsymbol{x}^{\mathrm{T}}\boldsymbol{\delta}/\|\boldsymbol{\delta}\|_\Sigma)^2\mathbb{1}\{|\boldsymbol{x}^{\mathrm{T}}\boldsymbol{\delta}/\|\boldsymbol{\delta}\|_\Sigma| > g/(4r)\}\right]\right.$$
$$\left. - \mathbb{E}\left[(\boldsymbol{x}^{\mathrm{T}}\boldsymbol{\delta}/\|\boldsymbol{\delta}\|_\Sigma)^2\mathbb{1}\{|\boldsymbol{x}^{\mathrm{T}}(\boldsymbol{\beta}_1 - \boldsymbol{\beta}^*)| > g/8\}\right]\right\}.$$
$$\geq \frac{7\underline{f}}{16}\left\{1 - \mathbb{E}\left[(\boldsymbol{x}^{\mathrm{T}}\boldsymbol{\delta}/\|\boldsymbol{\delta}\|_\Sigma)^2\mathbb{1}\{|\boldsymbol{x}^{\mathrm{T}}\boldsymbol{\delta}/\|\boldsymbol{\delta}\|_\Sigma| > g/(4r)\}\right] - \mu_4^{1/2}\mathbb{P}\big(|\boldsymbol{x}^{\mathrm{T}}(\boldsymbol{\beta}_1 - \boldsymbol{\beta}^*)| > g/8\big)^{1/2}\right\}.$$

By the definition of $\eta_\delta$, as long as $0 < r \leq g/(4\eta_{1/4})$,

$$\sup_{\boldsymbol{\delta}\in\mathbb{B}_\Sigma(r)} \mathbb{E}\left[(\boldsymbol{x}^{\mathrm{T}}\boldsymbol{\delta}/\|\boldsymbol{\delta}\|_\Sigma)^2\mathbb{1}\{|\boldsymbol{x}^{\mathrm{T}}\boldsymbol{\delta}/\|\boldsymbol{\delta}\|_\Sigma| > g/(4r)\}\right] \leq \frac{1}{4}.$$

Moreover, $\boldsymbol{\beta}_1 \in \boldsymbol{\beta}^* + \mathbb{B}_\Sigma(r/2)$. Under Assumption 3.3 with $v_0 \geq 1$, the tail bounds of sub-exponential $\boldsymbol{z} = \Sigma^{-1/2}\boldsymbol{x}$ implys that

$$\mathbb{P}\big(|\boldsymbol{x}^{\mathrm{T}}(\boldsymbol{\beta}_1 - \boldsymbol{\beta}^*)| > g/8\big) \leq 2\exp\left\{-\frac{g}{4v_0 r}\right\}.$$

Let $g \geq 8v_0 r\mu_4^{1/2}$ and then it follows from a numerical calculation that

$$\left\{1 - \mathbb{E}\left[(\boldsymbol{x}^{\mathrm{T}}\boldsymbol{\delta}/\|\boldsymbol{\delta}\|_\Sigma)^2\mathbb{1}\{|\boldsymbol{x}^{\mathrm{T}}\boldsymbol{\delta}/\|\boldsymbol{\delta}\|_\Sigma| > g/(4r)\}\right] - \mu_4^{1/2}\mathbb{P}\big(|\boldsymbol{x}^{\mathrm{T}}(\boldsymbol{\beta}_1 - \boldsymbol{\beta}^*)| > g/8\big)^{1/2}\right\} \geq 0.23$$

holds uniformly over $\Delta \in \mathbb{B}_\Sigma(r) \cap \mathbb{C}_\Sigma(l)$. Putting together the pieces yields

$$\mathbb{E}D_0(\boldsymbol{\beta}_1, \boldsymbol{\beta}_2) > 0.1\underline{f}. \tag{26}$$

Next we find a lower bound of $D_0(\boldsymbol{\beta}_1, \boldsymbol{\beta}_2) - \mathbb{E}D_0(\boldsymbol{\beta}_1, \boldsymbol{\beta}_2)$ over $\Lambda(r, l) := \{(\boldsymbol{\beta}_1, \boldsymbol{\beta}_2) : \boldsymbol{\beta}_1 \in \boldsymbol{\beta}^* + \mathbb{B}_\Sigma(r/2), \boldsymbol{\beta}_2 \in \boldsymbol{\beta}_1 + \mathbb{B}_\Sigma(r) \cap \mathbb{C}_\Sigma(l), \mathrm{supp}(\boldsymbol{\beta}_1) \subseteq \mathcal{S}\}$. Define

$$\Omega(r, l) = \sup_{(\boldsymbol{\beta}_1, \boldsymbol{\beta}_2)\in\Lambda(r,l)} \{-D_0(\boldsymbol{\beta}_1, \boldsymbol{\beta}_2) + \mathbb{E}D_0(\boldsymbol{\beta}_1, \boldsymbol{\beta}_2)\}.$$

Write $D_0(\boldsymbol{\beta}_1, \boldsymbol{\beta}_2) = (1/n_0)\sum_{i=1}^{n_0} w_i(\boldsymbol{\beta}_1, \boldsymbol{\beta}_2)$, where

$$w_i(\boldsymbol{\beta}_1, \boldsymbol{\beta}_2) = g^{-1}\mathbb{1}\{|\epsilon_i| \le g/4\}\varphi_{g/(2r)}\big((\boldsymbol{x}_i^{(0)})^{\mathrm{T}}\boldsymbol{\delta}/\|\boldsymbol{\delta}\|_\Sigma\big)\psi_{g/4}\big((\boldsymbol{x}_i^{(0)})^{\mathrm{T}}(\boldsymbol{\beta}_1 - \boldsymbol{\beta}^*)\big)$$

satisfies $0 \le w_i(\boldsymbol{\beta}_1, \boldsymbol{\beta}_2) \le g/(4r)^2$, since $\varphi_R(u) \le (R/2)^2$ and $\psi_R(u) \in [0, 1]$. Moreover,

$$\mathbb{E}w_i^2(\boldsymbol{\beta}_1, \boldsymbol{\beta}_2) = \mathbb{E}\Big[g^{-2}\mathbb{1}\{|\epsilon_i| \le g/4\}\varphi_{g/(2r)}^2\big((\boldsymbol{x}_i^{(0)})^{\mathrm{T}}\boldsymbol{\delta}/\|\boldsymbol{\delta}\|_\Sigma\big)\psi_{g/4}^2\big((\boldsymbol{x}_i^{(0)})^{\mathrm{T}}(\boldsymbol{\beta}_1 - \boldsymbol{\beta}^*)\big)\Big]$$

$$\le \frac{9\bar{f}}{16g} \cdot \mathbb{E}\varphi_{g/(2r)}^2\big((\boldsymbol{x}_i^{(0)})^{\mathrm{T}}\boldsymbol{\delta}/\|\boldsymbol{\delta}\|_\Sigma\big)$$

$$\le \frac{9\bar{f}}{16g} \cdot \mathbb{E}\big((\boldsymbol{x}_i^{(0)})^{\mathrm{T}}\boldsymbol{\delta}/\|\boldsymbol{\delta}\|_\Sigma\big)^4 = \frac{9\bar{f}\mu_4}{16g}.$$

Using Bousquet's version of Talagrand's inequality yields that, for any $z > 0$,

$$\Omega(r, l) \le \mathbb{E}\Omega(r, l) + \{\mathbb{E}\Omega(r, l)\}^{1/2}\frac{1}{2r}\sqrt{\frac{gz}{n_0}} + \frac{3\sqrt{2}\mu_4^{1/2}}{4}\sqrt{\frac{\bar{f}z}{n_0 g}} + \frac{g}{(4r)^2}\frac{z}{3n_0}$$

$$\le \mathbb{E}\Omega(r, l) + \frac{1}{4}\mathbb{E}\Omega(r, l) + \frac{1}{4r^2}\frac{gz}{n_0} + \frac{3\sqrt{2}\mu_4^{1/2}}{4}\sqrt{\frac{\bar{f}z}{n_0 g}} + \frac{g}{(4r)^2}\frac{z}{3n_0}$$

$$\le \frac{5}{4}\mathbb{E}\Omega(r, l) + \frac{3\sqrt{2}\mu_4^{1/2}}{4}\sqrt{\frac{\bar{f}z}{n_0 g}} + \frac{13}{3}\frac{gz}{(4r)^2 n_0} \tag{27}$$

holds with probability at least $1 - e^{-z}$. To bound the expectation $\mathbb{E}\Omega(r, l)$, using Rademacher symmetrization and the connection between Gaussian and Rademacher complexities, Lemma 5.5 in Ledoux & Talagrand (1991), we have

$$\mathbb{E}\Omega(r, l) \le 2\sqrt{\frac{\pi}{2}}\mathbb{E}\left[\sup_{(\boldsymbol{\beta}_1, \boldsymbol{\beta}_2) \in \Lambda(r, l)} \mathbb{G}_{\boldsymbol{\beta}_1, \boldsymbol{\beta}_2}\right], \tag{28}$$

where $\mathbb{G}_{\boldsymbol{\beta}_1, \boldsymbol{\beta}_2} := (n_0 g)^{-1}\sum_{i=1}^{n_0} e_i\mathbb{1}\{|\epsilon_i| \le g/4\}\varphi_{g/(2r)}\big((\boldsymbol{x}_i^{(0)})^{\mathrm{T}}\boldsymbol{\delta}/\|\boldsymbol{\delta}\|_\Sigma\big)\psi_{g/4}\big((\boldsymbol{x}_i^{(0)})^{\mathrm{T}}(\boldsymbol{\beta}_1 - \boldsymbol{\beta}^*)\big)$ and $e_i$ are independent standard normal variables. Note that $\mathbb{G}_{\boldsymbol{\beta}_1, \boldsymbol{\beta}_2}$ is a Gaussian process conditioned on $\{(y_i^{(0)}, \boldsymbol{x}_i^{(0)})\}_{i=1}^{n_0}$ and $\mathbb{G}_{\boldsymbol{\beta}^*, \boldsymbol{\beta}^*} = 0$. For $(\boldsymbol{\beta}_1, \boldsymbol{\beta}_2)$ and $(\boldsymbol{\beta}_1', \boldsymbol{\beta}_2')$, write $\boldsymbol{\delta} = \boldsymbol{\beta}_1 - \boldsymbol{\beta}_2$, $\boldsymbol{\delta}' = \boldsymbol{\beta}_1' - \boldsymbol{\beta}_2'$ and $\chi_i = \mathbb{1}\{|\epsilon_i| \le g/4\}$, then

$$\mathbb{G}_{\boldsymbol{\beta}_1, \boldsymbol{\beta}_2} - \mathbb{G}_{\boldsymbol{\beta}_1', \boldsymbol{\beta}_2'}$$
$$= \mathbb{G}_{\boldsymbol{\beta}_1, \boldsymbol{\beta}_2} - \mathbb{G}_{\boldsymbol{\beta}_1', \boldsymbol{\beta}_1' + \boldsymbol{\delta}} + \mathbb{G}_{\boldsymbol{\beta}_1', \boldsymbol{\beta}_1' + \boldsymbol{\delta}} - \mathbb{G}_{\boldsymbol{\beta}_1', \boldsymbol{\beta}_2'}$$
$$= \frac{1}{n_0 g}\sum_{i=1}^{n_0} e_i\chi_i\varphi_{g/(2r)}\big((\boldsymbol{x}_i^{(0)})^{\mathrm{T}}\boldsymbol{\delta}/\|\boldsymbol{\delta}\|_\Sigma\big)\Big\{\psi_{g/4}\big((\boldsymbol{x}_i^{(0)})^{\mathrm{T}}(\boldsymbol{\beta}_1 - \boldsymbol{\beta}^*)\big) - \psi_{g/4}\big((\boldsymbol{x}_i^{(0)})^{\mathrm{T}}(\boldsymbol{\beta}_1' - \boldsymbol{\beta}^*)\big)\Big\}$$
$$+ \frac{1}{n_0 g}\sum_{i=1}^{n_0} e_i\chi_i\psi_{g/4}\big((\boldsymbol{x}_i^{(0)})^{\mathrm{T}}(\boldsymbol{\beta}_1' - \boldsymbol{\beta}^*)\big)\Big\{\varphi_{g/(2r)}\big((\boldsymbol{x}_i^{(0)})^{\mathrm{T}}\boldsymbol{\delta}/\|\boldsymbol{\delta}\|_\Sigma\big) - \varphi_{g/(2r)}\big((\boldsymbol{x}_i^{(0)})^{\mathrm{T}}\boldsymbol{\delta}'/\|\boldsymbol{\delta}'\|_\Sigma\big)\Big\}.$$

Note that $\varphi_R$ and $\psi_R$ are Lipschitz continuous, and $\varphi_R(u) \le (R/2)^2$. Let $\mathbb{E}^*$ be the conditional expectation given $\{(y_i^{(0)}, \boldsymbol{x}_i^{(0)})\}_{i=1}^{n_0}$. Consequently,

$$\mathbb{E}^*\big(\mathbb{G}_{\boldsymbol{\beta}_1, \boldsymbol{\beta}_2} - \mathbb{G}_{\boldsymbol{\beta}_1', \boldsymbol{\beta}_1' + \boldsymbol{\delta}}\big)^2 \le \frac{1}{(n_0 g)^2}\left(\frac{8}{g}\right)^2\left(\frac{g}{4r}\right)^4\sum_{i=1}^{n_0}\chi_i\big((\boldsymbol{x}_i^{(0)})^{\mathrm{T}}(\boldsymbol{\beta}_1 - \boldsymbol{\beta}_1')\big)^2$$

$$= \frac{1}{4r^4 n_0^2}\sum_{i=1}^{n_0}\chi_i\big((\boldsymbol{x}_i^{(0)})^{\mathrm{T}}(\boldsymbol{\beta}_1 - \boldsymbol{\beta}_1')\big)^2 \tag{29}$$

and

$$\mathbb{E}^*\big(\mathbb{G}_{\boldsymbol{\beta}_1', \boldsymbol{\beta}_1' + \boldsymbol{\delta}} - \mathbb{G}_{\boldsymbol{\beta}_1', \boldsymbol{\beta}_2'}\big)^2 \le \frac{1}{(n_0 g)^2}\left(\frac{g}{2r}\right)^2\sum_{i=1}^{n_0}\chi_i\big((\boldsymbol{x}_i^{(0)})^{\mathrm{T}}\boldsymbol{\delta}/\|\boldsymbol{\delta}\|_\Sigma - (\boldsymbol{x}_i^{(0)})^{\mathrm{T}}\boldsymbol{\delta}'/\|\boldsymbol{\delta}'\|_\Sigma\big)^2$$

$$= \frac{1}{4r^2 n_0^2}\sum_{i=1}^{n_0}\chi_i\big((\boldsymbol{x}_i^{(0)})^{\mathrm{T}}(\boldsymbol{\delta}/\|\boldsymbol{\delta}\|_\Sigma - \boldsymbol{\delta}'/\|\boldsymbol{\delta}'\|_\Sigma)\big)^2. \tag{30}$$

Motivated by the last two inequalities, we have

$$
\begin{aligned}
&\mathbb{E}^*\big(\mathbb{G}_{\boldsymbol{\beta}_1,\boldsymbol{\beta}_2} - \mathbb{G}_{\boldsymbol{\beta}_1',\boldsymbol{\beta}_2'}\big)^2 \\
&\leq \frac{1}{2r^4 n_0^2}\sum_{i=1}^{n_0}\chi_i\big((\boldsymbol{x}_i^{(0)})^{\mathrm{T}}(\boldsymbol{\beta}_1-\boldsymbol{\beta}_1')\big)^2 + \frac{1}{2r^2 n_0^2}\sum_{i=1}^{n_0}\chi_i\big((\boldsymbol{x}_i^{(0)})^{\mathrm{T}}(\boldsymbol{\delta}/||\boldsymbol{\delta}||_\Sigma - \boldsymbol{\delta}'/||\boldsymbol{\delta}'||_\Sigma)\big)^2.
\end{aligned}
$$

Define another Gaussian process $\mathbb{Z}_{\boldsymbol{\beta}_1,\boldsymbol{\beta}_2}$ as

$$
\begin{aligned}
\mathbb{Z}_{\boldsymbol{\beta}_1,\boldsymbol{\beta}_2} &= \frac{1}{\sqrt{2}r^2 n_0}\sum_{i=1}^{n_0} e_i'\chi_i(\boldsymbol{x}_i^{(0)})^{\mathrm{T}}(\boldsymbol{\beta}_1-\boldsymbol{\beta}^*) + \frac{1}{\sqrt{2}r n_0}\sum_{i=1}^{n_0} e_i''\chi_i(\boldsymbol{x}_i^{(0)})^{\mathrm{T}}(\boldsymbol{\beta}_2-\boldsymbol{\beta}_1)/||\boldsymbol{\delta}||_\Sigma \\
&= \frac{1}{\sqrt{2}r^2 n_0}\sum_{i=1}^{n_0} e_i'\chi_i(\boldsymbol{x}_{i,\mathcal{S}}^{(0)})^{\mathrm{T}}(\boldsymbol{\beta}_1-\boldsymbol{\beta}^*)_\mathcal{S} + \frac{1}{\sqrt{2}r n_0}\sum_{i=1}^{n_0} e_i''\chi_i(\boldsymbol{x}_i^{(0)})^{\mathrm{T}}(\boldsymbol{\beta}_2-\boldsymbol{\beta}_1)/||\boldsymbol{\delta}||_\Sigma
\end{aligned}
$$

such that $\mathbb{E}^*(\mathbb{G}_{\boldsymbol{\beta}_1,\boldsymbol{\beta}_2} - \mathbb{G}_{\boldsymbol{\beta}_1',\boldsymbol{\beta}_2'})^2 \leq \mathbb{E}^*(\mathbb{Z}_{\boldsymbol{\beta}_1,\boldsymbol{\beta}_2} - \mathbb{Z}_{\boldsymbol{\beta}_1',\boldsymbol{\beta}_2'})^2$, where $\{e_i'\}$ and $\{e_i''\}$ are two dependent copies of $\{e_i\}$. The second equlity holds since $\text{supp}(\boldsymbol{\beta}_1)$, $\text{supp}(\boldsymbol{\beta}^*) \subseteq \mathcal{S}$. Applying Theorem 7.2.11 in Vershynin (2018), we obtain

$$
\mathbb{E}^*\Big(\sup_{\boldsymbol{\beta}_1,\boldsymbol{\beta}_2}\mathbb{G}_{\boldsymbol{\beta}_1,\boldsymbol{\beta}_2}\Big) \leq \mathbb{E}^*\Big(\sup_{\boldsymbol{\beta}_1,\boldsymbol{\beta}_2}\mathbb{Z}_{\boldsymbol{\beta}_1,\boldsymbol{\beta}_2}\Big). \tag{31}
$$

To bound the supremum of $\mathbb{Z}_{\boldsymbol{\beta}_1,\boldsymbol{\beta}_2}$, using the cone constraint and $\boldsymbol{\beta}_1 \in \boldsymbol{\beta}^* + \mathbb{B}_\Sigma(r/2)$, we have

$$
\begin{aligned}
\mathbb{E}^*\Big(\sup_{\boldsymbol{\beta}_1,\boldsymbol{\beta}_2}\mathbb{Z}_{\boldsymbol{\beta}_1,\boldsymbol{\beta}_2}\Big) &\leq \frac{\sqrt{2}}{4r}\mathbb{E}\Big\|\frac{1}{n_0}\sum_{i=1}^{n_0} e_i'\chi_i \boldsymbol{S}^{-1/2}\boldsymbol{x}_{i,\mathcal{S}}^{(0)}\Big\|_2 + \frac{\sqrt{2}l}{2r}\mathbb{E}\Big\|\frac{1}{n_0}\sum_{i=1}^{n_0} e_i''\chi_i \boldsymbol{x}_i^{(0)}\Big\|_\infty \\
&\leq \frac{\sqrt{2}}{4r}\sqrt{\frac{9\bar{f}g}{16}\frac{s}{n_0}} + \frac{\sqrt{2}l}{2r}\mathbb{E}\Big\|\frac{1}{n_0}\sum_{i=1}^{n_0} e_i''\chi_i \boldsymbol{x}_i^{(0)}\Big\|_\infty,
\end{aligned} \tag{32}
$$

where $\boldsymbol{S} = \Sigma_{\mathcal{S}\mathcal{S}} = \mathbb{E}(\boldsymbol{x}_\mathcal{S}\boldsymbol{x}_\mathcal{S}^{\mathrm{T}})$. Thus, by (28) (31) and (32), we have

$$
\mathbb{E}\Omega(r,l) \leq \sqrt{\pi}\Big\{\frac{3}{8r}\sqrt{\frac{\bar{f}gs}{n_0}} + \frac{l}{r}\mathbb{E}\Big\|\frac{1}{n_0}\sum_{i=1}^{n_0} e_i''\chi_i \boldsymbol{x}_i^{(0)}\Big\|_\infty\Big\}. \tag{33}
$$

It remains to find the bound of the second term on the right-hand side of (33). Note that the variable $|n_0^{-1}\sum_{i=1}^{n_0} e_i''\chi_i x_{ij}^{(0)}|$ is zero-mean for $j = 1, \ldots, p$.

$$
\begin{aligned}
\mathbb{E}\Big[\exp\Big(\lambda\Big|\frac{1}{n_0}\sum_{i=1}^{n_0} e_i''\chi_i x_{ij}^{(0)}\Big|\Big)\Big] &\leq \prod_{i=1}^{n_0}\mathbb{E}\Big[\exp\Big(\Big|\frac{\lambda e_i''\chi_i x_{ij}^{(0)}}{n_0}\Big|\Big)\Big] \\
&\leq \prod_{i=1}^{n_0}\exp\Big\{\Big|\frac{\lambda^2\chi_i^2(x_{ij}^{(0)})^2}{2n_0^2}\Big|\Big\} \\
&= \exp\Big\{\frac{\lambda^2}{2n_0^2}\sum_{i=1}^{n_0}\chi_i^2(x_{ij}^{(0)})^2\Big\}.
\end{aligned}
$$

Thus, $|n_0^{-1}\sum_{i=1}^{n_0} e_i''\chi_i x_{ij}^{(0)}|$ is sub-Gaussian with parameter $n_0^{-1}\sqrt{\sum_{i=1}^{n_0}\chi_i^2(x_{ij}^{(0)})^2}$. Applying Lemma 15 in Loh & Wainwright (2015), for some universal constant $c_1 > 0$, we have

$$
\begin{aligned}
\mathbb{E}\Big[\Big\|\frac{1}{n_0}\sum_{i=1}^{n_0} e_i''\chi_i \boldsymbol{x}_i^{(0)}\Big\|_\infty \Big| \boldsymbol{x}_i^{(0)}\Big] &\leq \frac{c_1}{n_0}\cdot\max_{j=1,\ldots,p}\sqrt{\sum_{i=1}^{n_0}\chi_i^2(x_{ij}^{(0)})^2}\cdot\sqrt{\log p} \\
&\leq \frac{c_1\sqrt{\log p}}{n_0}\sqrt{\sum_{i=1}^{n_0}\chi_i^2(x_{ij}^{(0)})^2},
\end{aligned}
$$

implying that

$$\mathbb{E}\left\|\frac{1}{n_0}\sum_{i=1}^{n_0}e_i''\chi_i\boldsymbol{x}_i^{(0)}\right\|_\infty \le c_1\sqrt{\frac{\log p}{n_0}}\cdot\mathbb{E}\left[\sqrt{\frac{\sum_{i=1}^{n_0}\chi_i^2(x_{ij}^{(0)})^2}{n_0}}\right]$$

$$\le c_1\sqrt{\frac{\log p}{n_0}}\cdot\sqrt{\mathbb{E}\left[\frac{\sum_{i=1}^{n_0}\chi_i^2(x_{ij}^{(0)})^2}{n_0}\right]}$$

$$\le c_1\sigma_x\sqrt{\frac{\log p}{n_0}}\cdot\sqrt{\frac{9\bar{f}g}{16}}$$

Therefore,

$$\mathbb{E}\left\|\frac{1}{n_0}\sum_{i=1}^{n_0}e_i''\chi_i\boldsymbol{x}_i^{(0)}\right\|_\infty \le \frac{3c_1\sigma_x}{4}\sqrt{\frac{\bar{f}g\log p}{n_0}}. \tag{34}$$

Plug this bound to (33), we obtain

$$\mathbb{E}\Omega(r,l) \le \sqrt{\pi}\left\{\frac{3}{8r}\sqrt{\frac{\bar{f}gs}{n_0}}+\frac{3c_1\sigma_x l}{4r}\sqrt{\frac{\bar{f}g\log p}{n_0}}\right\}. \tag{35}$$

Finally, if we take $r=\min\{g/(4\eta_{1/4}),g/(8v_0\mu_4^{1/2})\}$ and $z=t+\log p$, combining (35) with Bousquet's version inequality (27), we conclude that

$$\Omega(r,l)\le\sqrt{\pi}\left\{\frac{15}{32r}\sqrt{\frac{\bar{f}gs}{n_0}}+\frac{15c_1\sigma_x l}{16r}\sqrt{\frac{\bar{f}g\log p}{n_0}}\right\}+\frac{3\sqrt{2}\mu_4^{1/2}}{4}\sqrt{\frac{\bar{f}z}{n_0 g}}+\frac{13}{3}\frac{gz}{(4r)^2 n_0}$$

$$\le 0.02\underline{f}+c'l\sqrt{(t+\log p)/(n_0 g)}$$

with probability at least $1-p^{-1}e^{-t}$ as long as $n_0 g\gtrsim\bar{f}\underline{f}^{-2}\eta_{1/4}^2\mu_4\sigma_x^2 s$. This, together with (24), (25) and (26), we have

$$\mathcal{T}(\boldsymbol{\beta}_1,\boldsymbol{\beta}_2)\ge\frac{\kappa}{2}\|\boldsymbol{\delta}\|_\Sigma^2\left[0.1\underline{f}-\left(0.02\underline{f}+c'l\sqrt{\frac{t+\log p}{n_0 g}}\right)\right]$$

$$\ge\frac{\kappa}{2}\|\boldsymbol{\delta}\|_\Sigma^2\left(0.08\underline{f}-c'l\sqrt{\frac{t+\log p}{n_0 g}}\right) \tag{36}$$

with probability at least $1-p^{-1}e^{-t}$.

It remains to extend this bound to one that is uniform in the ratio $\|\boldsymbol{\delta}\|_1/\|\boldsymbol{\delta}\|_\Sigma$, which we do via a peeling argument. Consider the inequality

$$\mathcal{T}(\boldsymbol{\beta}_1,\boldsymbol{\beta}_2)\ge\frac{\kappa\cdot\underline{f}}{25}\|\boldsymbol{\delta}\|_\Sigma^2-2c''\|\boldsymbol{\delta}\|_1\|\boldsymbol{\delta}\|_\Sigma\sqrt{\frac{t+\log p}{n_0 g}}\right)$$

$$\frac{1}{\|\boldsymbol{\delta}\|_\Sigma^2}\mathcal{T}(\boldsymbol{\beta}_1,\boldsymbol{\beta}_2)\ge\frac{\kappa\cdot\underline{f}}{25}-\frac{2c''\|\boldsymbol{\delta}\|_1}{\|\boldsymbol{\delta}\|_\Sigma}\sqrt{\frac{t+\log p}{n_0 g}}$$

$$\frac{\kappa\cdot\underline{f}}{25}-\frac{1}{\|\boldsymbol{\delta}\|_\Sigma^2}\mathcal{T}(\boldsymbol{\beta}_1,\boldsymbol{\beta}_2)\le\frac{2c''\|\boldsymbol{\delta}\|_1}{\|\boldsymbol{\delta}\|_\Sigma}\sqrt{\frac{t+\log p}{n_0 g}}.$$

For positive integers $k=1,\ldots,N:=\lceil\log_2(c\sqrt{n_0 g/\log p})\rceil$, define the set $\Theta_k=\{\boldsymbol{\delta}\in\mathbb{R}^p:\gamma_1^{-1/2}2^{k-1}\le\|\boldsymbol{\delta}\|_1/\|\boldsymbol{\delta}\|_\Sigma\le\gamma_1^{-1/2}2^k\}$, so that

$$\left\{\boldsymbol{\delta}\in\mathbb{R}^p:\gamma_1^{-1/2}\le\frac{\|\boldsymbol{\delta}\|_1}{\|\boldsymbol{\delta}\|_\Sigma}\le\frac{\kappa\cdot\underline{f}}{50c''}\sqrt{\frac{n_0 g}{t+\log p}}:=\zeta'\right\}\subseteq\cup_{k=1}^N\Theta_k.$$

Then

$$\mathbb{P}\Bigg\{\exists \boldsymbol{\delta} \in \big\{\boldsymbol{\delta} \in \mathbb{R}^p : \gamma_1^{-1/2} \leq ||\boldsymbol{\delta}||_1/||\boldsymbol{\delta}||_\Sigma \leq \zeta'\big\} \text{ s.t.}$$

$$\frac{\kappa \cdot f}{25} - \frac{1}{||\boldsymbol{\delta}||_\Sigma^2}\mathcal{T}(\boldsymbol{\beta}_1, \boldsymbol{\beta}_2) > \frac{2c''||\boldsymbol{\delta}||_1}{||\boldsymbol{\delta}||_\Sigma}\sqrt{\frac{t + \log p}{n_0 g}}\Bigg\}$$

$$\leq \sum_{k=1}^N \mathbb{P}\Bigg\{\exists \boldsymbol{\delta} \in \Theta_k \text{ s.t. } \frac{\kappa \cdot f}{25} - \frac{1}{||\boldsymbol{\delta}||_\Sigma^2}\mathcal{T}(\boldsymbol{\beta}_1, \boldsymbol{\beta}_2) > c''\gamma_1^{-1/2}2^k\sqrt{\frac{t + \log p}{n_0 g}}\Bigg\}$$

$$\leq \sum_{k=1}^N \mathbb{P}\Bigg\{\sup_{||\boldsymbol{\delta}||_1/||\boldsymbol{\delta}||_\Sigma \leq \gamma_1^{-1/2}2^k} \frac{\kappa \cdot f}{25} - \frac{1}{||\boldsymbol{\delta}||_\Sigma^2}\mathcal{T}(\boldsymbol{\beta}_1, \boldsymbol{\beta}_2) > c''\gamma_1^{-1/2}2^k\sqrt{\frac{t + \log p}{n_0 g}}\Bigg\}$$

$$\leq \sum_{k=1}^N p^{-1}e^{-t} \leq \lceil\log_2(c\sqrt{n_0 g/\log p})\rceil p^{-1}e^{-t}.$$

Taking $t = \log\{\log_2(c\sqrt{n_0 g/\log p})\} + u$ yields that with probability at least $1 - p^{-1}e^{-u}$,

$$\frac{\kappa \cdot f}{25} - \frac{1}{||\boldsymbol{\delta}||_\Sigma^2}\mathcal{T}(\boldsymbol{\beta}_1, \boldsymbol{\beta}_2) \leq \frac{2c''||\boldsymbol{\delta}||_1}{||\boldsymbol{\delta}||_\Sigma}\sqrt{\frac{\log p + \log\{\log_2(c\sqrt{n_0 g/\log p})\} + u}{n_0 g}}.$$

Multiplying by $||\boldsymbol{\delta}||_\Sigma^2$ on both sides yields

$$\mathcal{T}(\boldsymbol{\beta}_1, \boldsymbol{\beta}_2) \geq \frac{\kappa \cdot f}{25}||\boldsymbol{\delta}||_\Sigma^2 - 2c''||\boldsymbol{\delta}||_1||\boldsymbol{\delta}||_\Sigma\sqrt{\frac{\log p + \log\{\log_2(c\sqrt{n_0 g/\log p})\} + u}{n_0 g}},$$

where $c'' > 0$ is a constant depending only on $(\underline{\kappa}, \underline{f})$.

### A.4  Proof of Theorem 3.1

**Transferring step:** Let $\boldsymbol{\omega}^*$ be the true parameter of the transferring step and $\mathcal{S}$ be the active set of $\boldsymbol{\beta}^*$ with cardinality $s$. The symmetric Bregman divergence between $\hat{\boldsymbol{\omega}}^{\mathcal{A}_m}$ and $\boldsymbol{\omega}^*$ is defined as

$$\left(\nabla\hat{Q}_h(\hat{\boldsymbol{\omega}}^{\mathcal{A}_m}) - \nabla\hat{Q}_h(\boldsymbol{\omega}^*)\right)^{\mathrm{T}}(\hat{\boldsymbol{\omega}}^{\mathcal{A}_m} - \boldsymbol{\omega}^*) \geq 0. \tag{37}$$

Let $\hat{\Delta} = \hat{\boldsymbol{\omega}}^{\mathcal{A}_m} - \boldsymbol{\omega}^*$. By optimality, there exists a subgradient $\hat{\boldsymbol{\nu}} \in \partial \sum_{i=1}^n q_{\lambda_\omega}(|\omega_i|)$, such that

$$\nabla\hat{Q}_h(\hat{\boldsymbol{\omega}}^{\mathcal{A}_m}) + \lambda_\omega\hat{\boldsymbol{\nu}} = 0.$$

Then (37) is equivalent to

$$-\left(\nabla\hat{Q}_h(\boldsymbol{\omega}^*)\right)^{\mathrm{T}}\hat{\Delta} - \lambda_\omega\hat{\boldsymbol{\nu}}^{\mathrm{T}}\hat{\Delta} \geq 0. \tag{38}$$

Note that

$$\hat{\boldsymbol{\nu}}^{\mathrm{T}}(\boldsymbol{\omega}^* - \hat{\boldsymbol{\omega}}^{\mathcal{A}_m}) \leq ||\boldsymbol{\omega}^*||_1 - ||\hat{\boldsymbol{\omega}}^{\mathcal{A}_m}||_1 = ||\boldsymbol{\omega}_{\mathcal{S}}^*||_1 + ||\boldsymbol{\omega}_{\mathcal{S}^c}^*||_1 - ||\hat{\Delta} + \boldsymbol{\omega}^*||_1$$

$$= ||\boldsymbol{\omega}_{\mathcal{S}}^*||_1 + ||\boldsymbol{\omega}_{\mathcal{S}^c}^*||_1 - ||\hat{\Delta}_{\mathcal{S}} + \boldsymbol{\omega}_{\mathcal{S}}^*||_1 - ||\hat{\Delta}_{\mathcal{S}^c} + \boldsymbol{\omega}_{\mathcal{S}^c}^*||_1$$

$$\leq ||\hat{\Delta}_{\mathcal{S}}||_1 - ||\hat{\Delta}_{\mathcal{S}^c}||_1 + 2||\boldsymbol{\omega}_{\mathcal{S}^c}^*||_1.$$

Then,

$$
\begin{aligned}
&\big(\nabla \hat{Q}_h(\hat{\boldsymbol{\omega}}^{\mathcal{A}_m}) - \nabla \hat{Q}_h(\boldsymbol{\omega}^*)\big)^{\mathrm{T}} \hat{\Delta} \\
&= \lambda_\omega \hat{\boldsymbol{\nu}}^{\mathrm{T}}(\boldsymbol{\omega}^* - \hat{\boldsymbol{\omega}}^{\mathcal{A}_m}) + \big(\nabla \hat{Q}_h(\boldsymbol{\omega}^*) - \nabla Q_h(\boldsymbol{\omega}^*)\big)^{\mathrm{T}}(\boldsymbol{\omega}^* - \hat{\boldsymbol{\omega}}^{\mathcal{A}_m}) + \big(\nabla Q_h(\boldsymbol{\omega}^*)\big)^{\mathrm{T}}(\boldsymbol{\omega}^* - \hat{\boldsymbol{\omega}}^{\mathcal{A}_m}) \\
&\leq \lambda_\omega \Big( ||\hat{\Delta}_{\mathcal{S}}||_1 - ||\hat{\Delta}_{\mathcal{S}^c}||_1 + 2||\boldsymbol{\omega}^*_{\mathcal{S}^c}||_1 \Big) + \underbrace{||\nabla \hat{Q}_h(\boldsymbol{\omega}^*) - \nabla Q_h(\boldsymbol{\omega}^*)||_\infty}_{||\boldsymbol{\pi}^*_h||_\infty} ||\hat{\Delta}||_1 \\
&\quad + \underbrace{||\Sigma^{-1/2} \nabla Q_h(\boldsymbol{\omega}^*)||_2}_{b^*_h} ||\hat{\Delta}||_\Sigma.
\end{aligned}
\tag{39}
$$

Conditioned on the event $\{\lambda_\omega \geq 2||\boldsymbol{\pi}^*_h||_\infty\}$, (39) becomes

$$
\big(\nabla \hat{Q}_h(\hat{\boldsymbol{\omega}}^{\mathcal{A}_m}) - \nabla \hat{Q}_h(\boldsymbol{\omega}^*)\big)^{\mathrm{T}} \hat{\Delta} \leq \lambda_{\boldsymbol{\omega}} \Big( ||\hat{\Delta}_{\mathcal{S}}||_1 - ||\hat{\Delta}_{\mathcal{S}^c}||_1 + 2||\boldsymbol{\omega}^*_{\mathcal{S}^c}||_1 \Big) + \frac{\lambda_{\boldsymbol{\omega}}}{2} ||\hat{\Delta}||_1 + b^*_h ||\hat{\Delta}||_\Sigma.
$$

Lemma A.1 implies $||\boldsymbol{\omega}^*_{\mathcal{S}^c}||_1 \leq C_1 m$, so we have

$$
\big(\nabla \hat{Q}_h(\hat{\boldsymbol{\omega}}^{\mathcal{A}_m}) - \nabla \hat{Q}_h(\boldsymbol{\omega}^*)\big)^{\mathrm{T}} \hat{\Delta} \leq \frac{3}{2}\lambda_{\boldsymbol{\omega}}||\hat{\Delta}_{\mathcal{S}}||_1 - \frac{1}{2}\lambda_{\boldsymbol{\omega}}||\hat{\Delta}_{\mathcal{S}^c}||_1 + 2\lambda_{\boldsymbol{\omega}} C_1 m + b^*_h ||\hat{\Delta}||_\Sigma.
$$

Since $(\nabla \hat{Q}_h(\hat{\boldsymbol{\omega}}^{\mathcal{A}_m}) - \nabla \hat{Q}_h(\boldsymbol{\omega}^*))^{\mathrm{T}} \hat{\Delta} \geq 0$, $\widehat{\Delta}$ satisfies the constraint $||\hat{\Delta}_{\mathcal{S}^c}||_1 \leq 3||\hat{\Delta}_{\mathcal{S}}||_1 + 4C_1 m + 2\lambda_\omega^{-1} b^*_h ||\hat{\Delta}||_\Sigma$, from which it follows that

$$
||\hat{\Delta}||_1 \leq 4s^{1/2}||\hat{\Delta}||_2 + 4C_1 m + 2\lambda_\omega^{-1} b^*_h ||\hat{\Delta}||_\Sigma.
\tag{40}
$$

Now we claim that when $\lambda_\omega \geq 2||\boldsymbol{\pi}^*_h||_\infty$, with probability at least $1 - p^{-1}$, it holds that

$$
||\hat{\Delta}||_\Sigma \leq \frac{8\phi_2 C_1 m}{\phi_1} \sqrt{\frac{\log p + \log(n_{\mathcal{A}_m} + n_0)}{n_{\mathcal{A}_m} + n_0}} + \frac{3\lambda_{\boldsymbol{\omega}} \gamma_1^{1/2} s^{1/2} + 2b^*_h}{\phi_1} + 2\sqrt{\frac{C_1 \lambda_{\boldsymbol{\omega}} m}{\phi_1}}.
\tag{41}
$$

If the claim does not hold , consider $\mathbb{C} = \{\Delta : 1.5\lambda_{\boldsymbol{\omega}}||\Delta_{\mathcal{S}}||_1 - 0.5\lambda_{\boldsymbol{\omega}}||\Delta_{\mathcal{S}^c}||_1 + 2\lambda_{\boldsymbol{\omega}} C_1 m + b^*_h ||\Delta||_\Sigma \geq 0\}$. For any $t \in (0, 1)$,

$$
\begin{aligned}
\frac{1}{2}\lambda_{\boldsymbol{\omega}}||t\hat{\Delta}_{\mathcal{S}^c}||_1 = t \cdot \frac{1}{2}\lambda_{\boldsymbol{\omega}}||\hat{\Delta}_{\mathcal{S}^c}||_1 &\leq t \cdot \left( \frac{3}{2}\lambda_{\boldsymbol{\omega}}||\hat{\Delta}_{\mathcal{S}}||_1 + 2\lambda_{\boldsymbol{\omega}} C_1 m + b^*_h ||\hat{\Delta}||_\Sigma \right) \\
&\leq \frac{3}{2}\lambda_{\boldsymbol{\omega}}||t\hat{\Delta}_{\mathcal{S}}||_1 + 2\lambda_{\boldsymbol{\omega}} C_1 m + b^*_h ||t\hat{\Delta}||_\Sigma,
\end{aligned}
$$

which implies that $t\hat{\Delta} \in \mathbb{C}$. We could find some $t$ satisfying that $||t\widehat{\Delta}||_\Sigma \leq 1$ and

$$
||t\hat{\Delta}||_\Sigma > \frac{8\phi_2 C_1 m}{\phi_1} \sqrt{\frac{\log p + \log(n_{\mathcal{A}_m} + n_0)}{n_{\mathcal{A}_m} + n_0}} + \frac{3\lambda_{\boldsymbol{\omega}} \gamma_1^{1/2} s^{1/2} + 2b^*_h}{\phi_1} + 2\sqrt{\frac{C_1 \lambda_{\boldsymbol{\omega}} m}{\phi_1}}.
$$

Denote $\tilde{\Delta} = t\hat{\Delta}$ and $F(\Delta) = \hat{Q}_h(\boldsymbol{\omega}^* + \Delta) - \hat{Q}_h(\boldsymbol{\omega}^*) + \lambda_{\boldsymbol{\omega}}(||\boldsymbol{\omega}^* + \Delta||_1 - ||\boldsymbol{\omega}^*||_1)$. Since $F(\mathbf{0}) = 0$ and $F(\hat{\Delta}) \leq 0$, by convexity,

$$
F(\tilde{\Delta}) = F(t\hat{\Delta} + (1 - t)\mathbf{0}) \leq tF(\hat{\Delta}) \leq 0.
$$

However,

$$
\begin{aligned}
F(\tilde{\Delta}) &= \hat{D}_h(\tilde{\Delta}) - \lambda_\omega||\boldsymbol{\omega}^*||_1 + \lambda_\omega||\boldsymbol{\omega}^* + \tilde{\Delta}||_1 \\
&= \hat{R}_h(\tilde{\Delta}) + \big(\nabla \hat{Q}_h(\boldsymbol{\omega}^*)\big)^{\mathrm{T}} \tilde{\Delta} - \lambda_\omega||\boldsymbol{\omega}^*||_1 + \lambda_\omega||\boldsymbol{\omega}^* + \tilde{\Delta}||_1 \\
&= \hat{R}_h(\tilde{\Delta}) - \Big( \lambda_\omega||\boldsymbol{\omega}^*||_1 - \lambda_\omega||\boldsymbol{\omega}^* + \tilde{\Delta}||_1 - \big(\nabla \hat{Q}_h(\boldsymbol{\omega}^*) - \nabla Q_h(\boldsymbol{\omega}^*)\big)^{\mathrm{T}} \tilde{\Delta} \\
&\quad - \big(\nabla Q_h(\boldsymbol{\omega}^*)\big)^{\mathrm{T}} \tilde{\Delta} \Big).
\end{aligned}
$$

Then by Proposition 3.1 and (39),

$$F(\tilde{\Delta}) \geq \phi_1||\tilde{\Delta}||_\Sigma^2 - \phi_2\sqrt{\frac{\log p + \log n}{nh}}||\tilde{\Delta}||_1||\tilde{\Delta}||_\Sigma - \frac{3}{2}\lambda_\omega||\tilde{\Delta}_\mathcal{S}||_1 + \frac{1}{2}\lambda_\omega||\tilde{\Delta}_{\mathcal{S}^c}||_1$$
$$- 2\lambda_\omega C_1 m - b_h^*||\tilde{\Delta}||_\Sigma$$
$$\geq \phi_1||\tilde{\Delta}||_\Sigma^2 - \phi_2\sqrt{\frac{\log p + \log n}{nh}}||\tilde{\Delta}||_1||\tilde{\Delta}||_\Sigma - \frac{3}{2}\lambda_\omega||\tilde{\Delta}_\mathcal{S}||_1 - 2\lambda_\omega C_1 m - b_h^*||\tilde{\Delta}||_\Sigma.$$

Note that $||\tilde{\Delta}_\mathcal{S}||_1 \leq s^{1/2}||\tilde{\Delta}||_2 \leq \gamma_1^{1/2}s^{1/2}||\tilde{\Delta}||_\Sigma$ and (40). Therefore, when

$$(n_{\mathcal{A}_m} + n_0)h > 16\phi_1^{-2}\phi_2^2(\log p + \log n)\max\{16s\gamma_1, 4\lambda_\omega^{-2}(b_h^*)^2\},$$

we have $\phi_2\sqrt{(\log p + \log n)/(nh)}(4\sqrt{s}\gamma_1^{1/2} + 2\lambda_\omega^{-1}b_h^*) \leq \phi_1/2$. Then it follows that,

$$F(\tilde{\Delta}) \geq \frac{1}{2}\phi_1||\tilde{\Delta}||_\Sigma^2 - \left(4\phi_2\sqrt{\frac{\log p + \log n}{nh}}C_1 m + \frac{3}{2}\lambda_\omega\gamma_1^{1/2}s^{1/2} + b_h^*\right)||\tilde{\Delta}||_\Sigma - 2\lambda_\omega C_1 m$$
$$> 0,$$

which contradicts with $F(\tilde{\Delta}) \leq 0$. Therefore the claim holds.

It remains to control the probability of the event $\{\lambda_\omega \geq ||\pi_h^*||_\infty\}$ and the probability of the local RSC condition. By Lemma A.2, we pick

$$\lambda_\omega = 2\left[\sigma\sqrt{\{\tau(1-\tau) + Ch^2\}\frac{4\log(2p)}{n_{\mathcal{A}_m} + n_0}} + \max(1-\tau, \tau)\frac{2\log(2p)}{n_{\mathcal{A}_m} + n_0}\right],$$

so that $\{\lambda_\omega \geq 2||\pi_h^*||_\infty\}$. From Lemma A.3, we have $b_h^* \leq Ch^2$. Now with probability at least $1 - (pn)^{-1}$,

$$||\hat{\Delta}||_\Sigma \lesssim m\sqrt{\frac{\log p + \log(n_{\mathcal{A}_m} + n_0)}{n_{\mathcal{A}_m} + n_0}} + \sqrt{\frac{s\log p}{n_{\mathcal{A}_m} + n_0}} + h^2 + \left(\frac{\log p}{n_{\mathcal{A}_m} + n_0}\right)^{\frac{1}{4}}\sqrt{m}.$$

We then let $h^2 \leq s^{1/2}\lambda_\omega$, so that

$$||\hat{\Delta}||_\Sigma \lesssim m\sqrt{\frac{\log p + \log(n_{\mathcal{A}_m} + n_0)}{n_{\mathcal{A}_m} + n_0}} + \sqrt{\frac{s\log p}{n_{\mathcal{A}_m} + n_0}} + \left(\frac{\log p}{n_{\mathcal{A}_m} + n_0}\right)^{\frac{1}{4}}\sqrt{m}, \tag{42}$$

with probability at least $1 - (pn)^{-1}$.

Note that

$$||\hat{\Delta}||_1 \leq 4||\hat{\Delta}_\mathcal{S}||_1 + 4C_1 m + 2\lambda_\omega^{-1}b_h^*||\hat{\Delta}||_\Sigma$$
$$\leq 4\sqrt{s}||\hat{\Delta}||_2 + 4C_1 m + 2\lambda_\omega^{-1}b_h^*||\hat{\Delta}||_\Sigma$$
$$\leq (4\sqrt{s}\gamma_1^{1/2} + l_0\kappa_2\lambda_\omega^{-1}h^2)||\hat{\Delta}||_\Sigma + 4C_1 m,$$

which encloses

$$||\hat{\Delta}||_1 \lesssim m\sqrt{\frac{s\log(p) + s\log(n_{\mathcal{A}_m} + n_0)}{n_{\mathcal{A}_m} + n_0}} + s\sqrt{\frac{\log p}{n_{\mathcal{A}_m} + n_0}} + \left(\frac{\log p}{n_{\mathcal{A}_m} + n_0}\right)^{\frac{1}{4}}\sqrt{sm} + m,$$

with probability at least $1 - (pn)^{-1}$.

**Debiasing step:** Denote $\boldsymbol{\delta}^* = \boldsymbol{\beta}^* - \boldsymbol{\omega}^*$, $\hat{\boldsymbol{\delta}}^{\mathcal{A}_m} = \hat{\boldsymbol{\beta}} - \hat{\boldsymbol{\omega}}^{\mathcal{A}_m}$ and $\hat{\boldsymbol{v}}^{\mathcal{A}_m} = \hat{\boldsymbol{\delta}}^{\mathcal{A}_m} - \boldsymbol{\delta}^*$.

Similar to (39), we have

$$
\begin{aligned}
&\left(\nabla\hat{Q}_g^{(0)}(\hat{\boldsymbol{\omega}}^{\mathcal{A}_m} + \hat{\boldsymbol{\delta}}^{\mathcal{A}_m}) - \nabla\hat{Q}_g^{(0)}(\boldsymbol{\beta}^*)\right)^{\mathrm{T}}(\hat{\boldsymbol{\beta}} - \boldsymbol{\beta}^*) \\
&\leq \lambda_\delta\left(||\boldsymbol{\beta}^* - \hat{\boldsymbol{\omega}}^{\mathcal{A}_m}||_1 - ||\hat{\boldsymbol{\beta}} - \hat{\boldsymbol{\omega}}^{\mathcal{A}_m}||_1\right) + \underbrace{\left\|\nabla\hat{Q}_g^{(0)}(\boldsymbol{\beta}^*) - \nabla Q_g^{(0)}(\boldsymbol{\beta}^*)\right\|_\infty}_{||\boldsymbol{\pi}_g^*||_\infty}\left\|\hat{\boldsymbol{\beta}} - \boldsymbol{\beta}^*\right\|_1 \\
&\quad + \underbrace{\left\|\Sigma^{-1/2}\nabla Q_g^{(0)}(\boldsymbol{\beta}^*)\right\|_2}_{b_g^*}\left\|\hat{\boldsymbol{\beta}} - \boldsymbol{\beta}^*\right\|_\Sigma \\
&\leq \frac{3}{2}\lambda_\delta||\boldsymbol{\beta}^* - \hat{\boldsymbol{\omega}}^{\mathcal{A}_m}||_1 - \frac{1}{2}\lambda_\delta||\hat{\boldsymbol{\beta}} - \hat{\boldsymbol{\omega}}^{\mathcal{A}_m}||_1 + b_g^*\left\|\hat{\boldsymbol{\beta}} - \boldsymbol{\beta}^*\right\|_\Sigma \\
&\leq \frac{3}{2}\lambda_\delta||\boldsymbol{\beta}^* - \boldsymbol{\omega}^*||_1 + \frac{3}{2}\lambda_\delta||\hat{\Delta}||_1 - \frac{1}{2}\lambda_\delta||\hat{\boldsymbol{\beta}} - \hat{\boldsymbol{\omega}}^{\mathcal{A}_m}||_1 + b_g^*\left\|\hat{\boldsymbol{\beta}} - \boldsymbol{\beta}^*\right\|_\Sigma \\
&\leq \frac{3}{2}\lambda_\delta Cm + \frac{3}{2}\lambda_\delta||\hat{\Delta}||_1 - \frac{1}{2}\lambda_\delta||\hat{\boldsymbol{\beta}} - \hat{\boldsymbol{\omega}}^{\mathcal{A}_m}||_1 + b_g^*\left\|\hat{\boldsymbol{\beta}} - \boldsymbol{\beta}^*\right\|_\Sigma.
\end{aligned} \tag{43}
$$

On the other hand,

$$
\begin{aligned}
&\left(\nabla\hat{Q}_g^{(0)}(\hat{\boldsymbol{\omega}}^{\mathcal{A}_m} + \hat{\boldsymbol{\delta}}^{\mathcal{A}_m}) - \nabla\hat{Q}_g^{(0)}(\boldsymbol{\beta}^*)\right)^{\mathrm{T}}(\hat{\boldsymbol{\beta}} - \boldsymbol{\beta}^*) \\
&\leq \lambda_\delta\left(||\boldsymbol{\beta}^* - \hat{\boldsymbol{\omega}}^{\mathcal{A}_m}||_1 - ||\hat{\boldsymbol{\beta}} - \hat{\boldsymbol{\omega}}^{\mathcal{A}_m}||_1\right) + \frac{\lambda_\delta}{2}\left\|\hat{\boldsymbol{\beta}} - \boldsymbol{\beta}^*\right\|_1 + b_g^*\left\|\hat{\boldsymbol{\beta}} - \boldsymbol{\beta}^*\right\|_\Sigma \\
&\leq \lambda_\delta||\boldsymbol{\beta}_{\mathcal{S}}^* - \hat{\boldsymbol{\omega}}_{\mathcal{S}}^{\mathcal{A}_m}||_1 + \lambda_\delta||\boldsymbol{\beta}_{\mathcal{S}^c}^* - \hat{\boldsymbol{\omega}}_{\mathcal{S}^c}^{\mathcal{A}_m}||_1 - \lambda_\delta||\hat{\boldsymbol{\beta}}_{\mathcal{S}} - \hat{\boldsymbol{\omega}}_{\mathcal{S}}^{\mathcal{A}_m}||_1 - \lambda_\delta||\hat{\boldsymbol{\beta}}_{\mathcal{S}^c} - \hat{\boldsymbol{\omega}}_{\mathcal{S}^c}^{\mathcal{A}_m}||_1 \\
&\quad + \frac{\lambda_\delta}{2}\left\|\hat{\boldsymbol{\beta}} - \boldsymbol{\beta}^*\right\|_1 + b_g^*\left\|\hat{\boldsymbol{\beta}} - \boldsymbol{\beta}^*\right\|_\Sigma \\
&\leq \lambda_\delta\left(||\boldsymbol{\beta}_{\mathcal{S}}^* - \hat{\boldsymbol{\omega}}_{\mathcal{S}}^{\mathcal{A}_m}||_1 - ||\hat{\boldsymbol{\beta}}_{\mathcal{S}} - \hat{\boldsymbol{\omega}}_{\mathcal{S}}^{\mathcal{A}_m}||_1\right) - \lambda_\delta\left(||\boldsymbol{\beta}_{\mathcal{S}^c}^* - \hat{\boldsymbol{\omega}}_{\mathcal{S}^c}^{\mathcal{A}_m}||_1 + ||\hat{\boldsymbol{\beta}}_{\mathcal{S}^c} - \hat{\boldsymbol{\omega}}_{\mathcal{S}^c}^{\mathcal{A}_m}||_1\right) \\
&\quad + 2\lambda_\delta||\boldsymbol{\beta}_{\mathcal{S}^c}^* - \hat{\boldsymbol{\omega}}_{\mathcal{S}^c}^{\mathcal{A}_m}||_1 + \frac{\lambda_\delta}{2}||\hat{\boldsymbol{\beta}} - \boldsymbol{\beta}^*||_1 + b_g^*\left\|\hat{\boldsymbol{\beta}} - \boldsymbol{\beta}^*\right\|_\Sigma \\
&\leq \lambda_\delta||\boldsymbol{\beta}_{\mathcal{S}}^* - \hat{\boldsymbol{\beta}}_{\mathcal{S}}||_1 - \lambda_\delta||\boldsymbol{\beta}_{\mathcal{S}^c}^* - \hat{\boldsymbol{\beta}}_{\mathcal{S}^c}||_1 + 2\lambda_\delta||\boldsymbol{\beta}_{\mathcal{S}^c}^* - \boldsymbol{\omega}_{\mathcal{S}^c}^*||_1 + 2\lambda_\delta||\hat{\Delta}_{\mathcal{S}^c}||_1 \\
&\quad + \frac{\lambda_\delta}{2}||\hat{\boldsymbol{\beta}} - \boldsymbol{\beta}^*||_1 + b_g^*\left\|\hat{\boldsymbol{\beta}} - \boldsymbol{\beta}^*\right\|_\Sigma \\
&\leq \frac{3}{2}\lambda_\delta||\boldsymbol{\beta}_{\mathcal{S}}^* - \hat{\boldsymbol{\beta}}_{\mathcal{S}}||_1 - \frac{1}{2}\lambda_\delta||\boldsymbol{\beta}_{\mathcal{S}^c}^* - \hat{\boldsymbol{\beta}}_{\mathcal{S}^c}||_1 + 2\lambda_\delta C_1 m + 2\lambda_\delta||\hat{\Delta}_{\mathcal{S}^c}||_1 + b_g^*\left\|\hat{\boldsymbol{\beta}} - \boldsymbol{\beta}^*\right\|_\Sigma \\
&\leq \frac{3}{2}\lambda_\delta||\boldsymbol{\beta}_{\mathcal{S}}^* - \hat{\boldsymbol{\beta}}_{\mathcal{S}}||_1 - \frac{1}{2}\lambda_\delta||\boldsymbol{\beta}_{\mathcal{S}^c}^* - \hat{\boldsymbol{\beta}}_{\mathcal{S}^c}||_1 + 2\lambda_\delta C_1 m + b_g^*\left\|\hat{\boldsymbol{\beta}} - \boldsymbol{\beta}^*\right\|_\Sigma \\
&\quad + 2\lambda_\delta\left(m\sqrt{\frac{s\log(p) + s\log(n_{\mathcal{A}_m} + n_0)}{n_{\mathcal{A}_m} + n_0}} + s\sqrt{\frac{\log p}{n_{\mathcal{A}_m} + n_0}} + \left(\frac{\log p}{n_{\mathcal{A}_m} + n_0}\right)^{\frac{1}{4}}\sqrt{sm} + m\right).
\end{aligned} \tag{44}
$$

Thus

$$
\begin{aligned}
||\boldsymbol{\beta}^* - \hat{\boldsymbol{\beta}}||_1 &\leq 4\gamma_1^{1/2}\sqrt{s}||\boldsymbol{\beta}^* - \hat{\boldsymbol{\beta}}||_\Sigma + 2b_g^*\lambda_\delta^{-1}||\boldsymbol{\beta}^* - \hat{\boldsymbol{\beta}}||_\Sigma \\
&\quad + 4\left(m\sqrt{\frac{s\log(p) + s\log(n_{\mathcal{A}_m} + n_0)}{n_{\mathcal{A}_m} + n_0}} + s\sqrt{\frac{\log p}{n_{\mathcal{A}_m} + n_0}} + \left(\frac{\log p}{n_{\mathcal{A}_m} + n_0}\right)^{\frac{1}{4}}\sqrt{sm} + m\right)
\end{aligned} \tag{45}
$$

Set $r = g/(48c)$ for some $c > 0$ and $R = (4\gamma_1^{1/2}\sqrt{s} + 2b_g^*\lambda_\delta^{-1})r + 4C\sqrt{s}$, if $m \leq C\sqrt{s}$ for some positive constant $C$ and $n_{\mathcal{A}_m} + n_0 \geq s\log p$. Denote $\Theta(r, R) = \mathbb{B}_\Sigma(r) \cap \mathbb{C}_\Sigma(R)$ and $\tilde{\boldsymbol{\beta}} = (1 - \eta)\boldsymbol{\beta}^* + \eta\hat{\boldsymbol{\beta}}$, where $\eta = \sup\{u \in [0, 1] : \boldsymbol{\beta}^* + u(\hat{\boldsymbol{\beta}} - \boldsymbol{\beta}^*) \in \boldsymbol{\beta}^* + \Theta(r, R)\}$. If $\hat{\boldsymbol{\beta}} \notin \Theta(r, R)$, then $\eta \in (0, 1)$ and $\tilde{\boldsymbol{\beta}}$ falls onto the boundary of $\Theta(r, R)$; otherwise $\tilde{\boldsymbol{\beta}} = \hat{\boldsymbol{\beta}}$.

Combining (43) and Proposition 3.2, we have

$$
\frac{\alpha_1}{2}||\tilde{\boldsymbol{\beta}} - \boldsymbol{\beta}^*||_\Sigma^2 - \alpha_2 \cdot \frac{\log p + \log n_0}{n_0 g}||\tilde{\boldsymbol{\beta}} - \boldsymbol{\beta}^*||_1^2 \leq \frac{3}{2}\lambda_\delta Cm + \frac{3}{2}\lambda_\delta||\tilde{\Delta}||_1 + b_g^*\left\|\tilde{\boldsymbol{\beta}} - \boldsymbol{\beta}^*\right\|_\Sigma. \tag{46}
$$

Besides, (43) implies

$$||\tilde{\boldsymbol{\beta}} - \hat{\boldsymbol{\omega}}^{\mathcal{A}_m}||_1 \le 3Cm + 3||\hat{\Delta}||_1 + \frac{2b_g^*}{\lambda_\delta}||\tilde{\boldsymbol{\beta}} - \boldsymbol{\beta}^*||_\Sigma.$$

As a result,

$$||\tilde{\boldsymbol{\beta}} - \boldsymbol{\beta}^*||_1 \le ||\hat{\boldsymbol{\beta}} - \hat{\boldsymbol{\omega}}^{\mathcal{A}_m}||_1 + ||\hat{\boldsymbol{\omega}}^{\mathcal{A}_m} - \boldsymbol{\beta}^*||_1$$
$$\le 4Cm + 4||\hat{\Delta}||_1 + \frac{2b_g^*}{\lambda_\delta}||\tilde{\boldsymbol{\beta}} - \boldsymbol{\beta}^*||_\Sigma.$$

Let $\alpha = \alpha_1 - 4b_g^{*2}\lambda_\delta^{-2}$, then (46) becomes

$$\alpha_1||\tilde{\boldsymbol{\beta}} - \boldsymbol{\beta}^*||_\Sigma^2 \le 2\alpha_2 \cdot \frac{\log p + \log n_0}{n_0 g}(16Cm^2 + 16||\hat{\Delta}||_1^2 + 4b_g^{*2}\lambda_\delta^{-2}||\tilde{\boldsymbol{\beta}} - \boldsymbol{\beta}^*||_\Sigma^2)$$
$$+ 3\lambda_\delta Cm + 3\lambda_\delta ||\hat{\Delta}||_1 + 2b_g^*||\tilde{\boldsymbol{\beta}} - \boldsymbol{\beta}^*||_\Sigma$$

$$\alpha||\tilde{\boldsymbol{\beta}} - \boldsymbol{\beta}^*||_\Sigma^2 - 2b_g^*||\tilde{\boldsymbol{\beta}} - \boldsymbol{\beta}^*||_\Sigma + \frac{b_g^{*2}}{\alpha} \lesssim \frac{\log p}{n_0 g}(m^2 + ||\hat{\Delta}||_1^2) + \lambda_\delta m + \lambda_\delta ||\hat{\Delta}||_1$$

$$\alpha\left(||\tilde{\boldsymbol{\beta}} - \boldsymbol{\beta}^*||_\Sigma - \frac{b_g^*}{\alpha}\right)^2 \lesssim \frac{\log p}{n_0 g}\left(m^2 + ||\hat{\Delta}||_1^2\right) + \lambda_\delta m + \lambda_\delta ||\hat{\Delta}||_1.$$

Thus,

$$||\tilde{\boldsymbol{\beta}} - \boldsymbol{\beta}^*||_\Sigma \lesssim \sqrt{\frac{\log p}{n_0 g}}(m + ||\hat{\Delta}||_1) + \sqrt{\lambda_\delta m} + \sqrt{\lambda_\delta ||\hat{\Delta}||_1} + b_g^*.$$

Let $\lambda_\delta = C\sqrt{\log(p)/n_0}$ and $g \asymp (\log(p)/n_0)^{1/4}$, then

$$||\tilde{\boldsymbol{\beta}} - \boldsymbol{\beta}^*||_\Sigma \lesssim \left(\frac{\log p}{n_0}\right)^{3/8}||\hat{\Delta}||_1 + \sqrt{m}\left(\frac{\log p}{n_0}\right)^{1/4} + \left(\frac{\log p}{n_0}\right)^{1/4}\sqrt{||\hat{\Delta}||_1} + \left(\frac{\log p}{n_0}\right)^{1/2}$$

$$\lesssim m\left(\frac{\log p}{n_0}\right)^{3/8} + s\left(\frac{\log p}{n_0}\right)^{3/8}\sqrt{\frac{\log p}{n_{\mathcal{A}_m} + n_0}}$$

$$+ \sqrt{sm}\left(\frac{\log p}{n_0}\right)^{3/8}\left(\frac{\log p}{n_{\mathcal{A}_m} + n_0}\right)^{1/4} + \sqrt{m}\left(\frac{\log p}{n_0}\right)^{1/4}$$

$$+ \sqrt{s}\left(\frac{\log p}{n_0}\right)^{1/4}\left(\frac{\log p}{n_{\mathcal{A}_m} + n_0}\right)^{1/4} + (sm)^{1/4}\left(\frac{\log p}{n_0}\right)^{1/4}\left(\frac{\log p}{n_{\mathcal{A}_m} + n_0}\right)^{1/8}.$$

If $n_0 > s^2 \log p$, $\hat{\boldsymbol{\beta}}$ falls in the interior of $\Theta(r, R)$, so we must have $\hat{\boldsymbol{\beta}} \in \Theta(r, R)$. Consequently, $\hat{\boldsymbol{\beta}} = \tilde{\boldsymbol{\beta}}$ satisfies the claimed bound,

$$||\hat{\boldsymbol{\beta}} - \boldsymbol{\beta}^*||_\Sigma \lesssim \sqrt{m}\left(\frac{\log p}{n_0}\right)^{1/4} + \sqrt{s}\left(\frac{\log p}{n_0}\right)^{1/4}\left(\frac{\log p}{n_{\mathcal{A}_m} + n_0}\right)^{1/4}. \tag{47}$$

In addition, if $m \le s\sqrt{\log(p)/n_0}$, the above upper bound is sharper than $\sqrt{s\log(p)/n_0}$. Then by (45), we have

$$||\hat{\boldsymbol{\beta}} - \boldsymbol{\beta}^*||_1 \lesssim s\sqrt{\frac{\log p}{n_{\mathcal{A}_m} + n_0}} + \left(\frac{\log p}{n_{\mathcal{A}_m} + n_0}\right)^{\frac{1}{4}}\sqrt{sm} + m.$$

## A.5 Proof of Proposition 3.3

The method is similar to the proof of Proposition 3.2. At first, the divergence is given by

$$D(\boldsymbol{\beta}_1, \boldsymbol{\beta}_2) = \left(\nabla\hat{Q}_g^{(0)}(\boldsymbol{\beta}_1) - \nabla\hat{Q}_g^{(0)}(\boldsymbol{\beta}_2)\right)^{\mathrm{T}}(\boldsymbol{\beta}_1 - \boldsymbol{\beta}_2).$$

For a given kernel function $K(\cdot)$ and bandwidth $g > 0$, the smoothed quantile loss $\hat{Q}_g^{(0)}$ can be written as $(n_0 g)^{-1} \sum_{i=1}^{n_0} \int_{-\infty}^{\infty} \rho_\tau(u) K\{(u + (\boldsymbol{x}_i^{(0)})^{\mathrm{T}} \boldsymbol{\beta} - y_i^{(0)})/g\} du$. Therefore

$$D(\boldsymbol{\beta}_1, \boldsymbol{\beta}_2) \geq \frac{\kappa}{n_0 g} \sum_{i=1}^{n_0} \left((\boldsymbol{x}_i^{(0)})^{\mathrm{T}}(\boldsymbol{\beta}_1 - \boldsymbol{\beta}_2)\right)^2 \mathbb{1}_{\mathcal{E}_i},$$

where the event $\mathcal{E}_i$ is defined by,

$$\mathcal{E}_i = \{|\epsilon_i| \leq g/4\} \cap \left\{ \left|(\boldsymbol{x}_i^{(0)})^{\mathrm{T}}(\boldsymbol{\beta}_1 - \boldsymbol{\beta}_2)\right| \leq g||\boldsymbol{\beta}_1 - \boldsymbol{\beta}_2||_\Sigma/(2r) \right\} \cap \left\{ \left|(\boldsymbol{x}_i^{(0)})^{\mathrm{T}}(\boldsymbol{\beta}_1 - \boldsymbol{\beta}^*)\right| \leq g/4 \right\}.$$

for all $\boldsymbol{\beta}_1 - \boldsymbol{\beta}_2 \in \mathbb{B}_\Sigma(r)$. For a truncation level $R > 0$, define functions $\varphi_R(u)$ and $\psi_R(u)$ as previous proof. By this construction, $\varphi_R(u) \leq u^2 \cdot \mathbb{1}\{|u| \leq R\}$, $\varphi_{cR}(cu) = c^2 \varphi_R(u)$, $\varphi_R$ is R-Lipschitz, $\psi_R$ is $(2/R)$-Lipschitz and $\psi_R(u) \leq \mathbb{1}\{|u| \leq R\}$.

From these two new-defined function, we have

$$D(\boldsymbol{\beta}_1, \boldsymbol{\beta}_2) \geq \frac{\kappa}{n_0 g}||\boldsymbol{\delta}||_\Sigma^2 \sum_{i=1}^{n_0} \mathbb{1}\{|\epsilon_i| \leq g/4\} \varphi_{g||\boldsymbol{\delta}||_\Sigma/(2r)}\left((\boldsymbol{x}_i^{(0)})^{\mathrm{T}} \boldsymbol{\delta}\right) \psi_{g/4}\left((\boldsymbol{x}_i^{(0)})^{\mathrm{T}}(\boldsymbol{\beta}_1 - \boldsymbol{\beta}^*)\right)$$

$$\geq \underline{\kappa}||\boldsymbol{\delta}||_\Sigma^2 \cdot \underbrace{\frac{1}{n_0 g} \sum_{i=1}^{n_0} \mathbb{1}\{|\epsilon_i| \leq g/4\} \varphi_{g/(2r)}\left((\boldsymbol{x}_i^{(0)})^{\mathrm{T}} \boldsymbol{\delta}/||\boldsymbol{\delta}||_\Sigma\right) \psi_{g/4}\left((\boldsymbol{x}_i^{(0)})^{\mathrm{T}}(\boldsymbol{\beta}_1 - \boldsymbol{\beta}^*)\right)}_{D_0(\boldsymbol{\beta}_1, \boldsymbol{\beta}_2)}, \quad (48)$$

where $\boldsymbol{\delta} = \boldsymbol{\beta}_1 - \boldsymbol{\beta}_2$. Finally, with a similar proof as Proposition 3.2, if $r = \min\{g/(4\eta_{1/4}), g/(32 v_0 \mu_4^{1/2})\}$ and $n_0 g \gtrsim \bar{f} \underline{f}^{-2} \eta_{1/4}^2 \mu_4 \sigma_x^2 s \log p$, then

$$D_0(\boldsymbol{\beta}_1, \boldsymbol{\beta}_2) \geq 0.1 \underline{f},$$

with probability at least $1 - (pn_0)^{-1}$. Therefore,

$$D(\boldsymbol{\beta}_1, \boldsymbol{\beta}_2) \geq 0.1 \underline{\kappa} \cdot \underline{f}||\boldsymbol{\beta}_1 - \boldsymbol{\beta}_2||_\Sigma^2.$$

## A.6 Proof of Theorem 3.2

For step 1, the parameter $\boldsymbol{\omega}^{(k)}$ is at most $s + m$ sparse. Therefore, similarly as Theorem 1 in Tan et al. (2022), we have

$$||\hat{\boldsymbol{\omega}}^{(k)} - \boldsymbol{\omega}^{(k)}||_2^2 \lesssim \frac{(s+m)\log p}{n^{(k)}}, \quad ||\hat{\boldsymbol{\omega}}^{(k)} - \boldsymbol{\omega}^{(k)}||_1 \lesssim (s+m)\sqrt{\frac{\log p}{n^{(k)}}}, \quad k \in \mathcal{A}_m'$$

with probability at least $1 - p^{-1}$, provided that the bandwidth $h$ satisfies

$$\max\left(\frac{\sigma_x}{\underline{f}}\sqrt{\frac{(s+m)\log p}{n^{(k)}}}, \frac{\sigma_x^2 \bar{f}}{\underline{f}^2}\frac{(s+m)\log p}{n^{(k)}}\right) \lesssim h \leq \min\{\underline{f}/(2l_0), (s^{1/2}\lambda_\omega^{(k)})\},$$

where $\sigma_x^2 = \max_{1 \geq j \geq p} \sigma_{jj}$, $\sigma_{jj}$ are the diagonal elements of $\Sigma$.

For step 2, denote $\boldsymbol{\delta}^{(k)} = \boldsymbol{\beta}^* - \boldsymbol{\omega}^{(k)}$, $\hat{\boldsymbol{\delta}}^{(k)} = \hat{\boldsymbol{\beta}} - \hat{\boldsymbol{\omega}}^{(k)}$ and $\hat{\boldsymbol{v}}^{(k)} = \hat{\boldsymbol{\delta}}^{(k)} - \boldsymbol{\delta}^{(k)}$. For each $k \in \mathcal{A}_m'$,

$$\left(\nabla \hat{Q}_g^{(0)}(\hat{\boldsymbol{\omega}}^{(k)} + \hat{\boldsymbol{\delta}}^{(k)}) - \nabla \hat{Q}_g^{(0)}(\hat{\boldsymbol{\omega}}^{(k)} + \boldsymbol{\delta}^{(k)})\right)^{\mathrm{T}} \hat{\boldsymbol{v}}^{(k)}$$

$$\leq \lambda_\delta\left(||\boldsymbol{\delta}^*||_1 - ||\hat{\boldsymbol{\delta}}^{(k)}||_1\right) + \left(\nabla \hat{Q}_g^{(0)}(\hat{\boldsymbol{\omega}}^{(k)} + \boldsymbol{\delta}^*) - \nabla \hat{Q}_g^{(0)}(\boldsymbol{\beta}^*)\right)^{\mathrm{T}}(\boldsymbol{\delta}^{(k)} - \hat{\boldsymbol{\delta}}^{(k)})$$

$$+ \underbrace{\left\|\nabla \hat{Q}_g^{(0)}(\boldsymbol{\beta}^*) - \nabla Q_g^{(0)}(\boldsymbol{\beta}^*)\right\|_\infty}_{||\boldsymbol{\pi}_g^*||_\infty} \left\|\hat{\boldsymbol{v}}^{(k)}\right\|_1 + \underbrace{\left\|\Sigma^{-1/2}\nabla Q_g^{(0)}(\boldsymbol{\beta}^*)\right\|_2}_{b_g^*} \left\|\hat{\boldsymbol{v}}^{(k)}\right\|_\Sigma. \quad (49)$$

Then by Lemma A.4 with $t = 2\log p$ and for each $k \in \mathcal{A}'_m$, we let $r_k = \sqrt{(s+m)\log(p)/n^{(k)}}$ and $l_k = (s+m)\sqrt{\log(p)/n^{(k)}}$. We have when $\lambda_\delta \geq 2\|\boldsymbol{\pi}_g^*\|_\infty$,

$$
\begin{aligned}
&\big(\nabla\hat{Q}_g^{(0)}(\hat{\boldsymbol{\omega}}^{(k)} + \hat{\boldsymbol{\delta}}^{(k)}) - \nabla\hat{Q}_g^{(0)}(\hat{\boldsymbol{\omega}}^{(k)} + \boldsymbol{\delta}^{(k)})\big)^{\mathrm{T}}\hat{\boldsymbol{v}}^{(k)} \\
&\leq \lambda_\delta\big(\|\boldsymbol{\delta}^{(k)}\|_1 - \|\hat{\boldsymbol{\delta}}^{(k)}\|_1\big) + \underbrace{C\bigg(\frac{s+m}{g}\sqrt{\frac{\log p}{n^{(k)}}}\sqrt{\frac{\log p}{n_0}} + \frac{\log p}{n_0} + \sqrt{s+m}\sqrt{\frac{\log p}{n^{(k)}}}\bigg)}_{C_v}\|\hat{\boldsymbol{v}}^{(k)}\|_1 \\
&\quad + \frac{\lambda_\delta}{2}\|\hat{\boldsymbol{v}}^{(k)}\|_1 + b_g^*\|\hat{\boldsymbol{v}}^{(k)}\|_\Sigma.
\end{aligned}
$$

Since

$$
\|\boldsymbol{\delta}_{\mathcal{S}_k}^{(k)}\|_1 - \|\hat{\boldsymbol{\delta}}_{\mathcal{S}_k}^{(k)}\|_1 \leq \|(\boldsymbol{\delta}^{(k)} - \hat{\boldsymbol{\delta}}^{(k)})_{\mathcal{S}_k}\|_1 \text{ and } \|\boldsymbol{\delta}_{\mathcal{S}_k^c}^{(k)}\|_1 - \|\hat{\boldsymbol{\delta}}_{\mathcal{S}_k^c}^{(k)}\|_1 = -\|(\hat{\boldsymbol{\delta}}^{(k)} - \boldsymbol{\delta}^{(k)})_{\mathcal{S}_k^c}\|_1,
$$

when $C_v < \lambda_\delta/2$, we obtain

$$
\begin{aligned}
\big(\nabla\hat{Q}_g^{(0)}(\hat{\boldsymbol{\omega}}^{(k)} + \hat{\boldsymbol{\delta}}^{(k)}) - \nabla\hat{Q}_g^{(0)}(\hat{\boldsymbol{\omega}}^{(k)} + \boldsymbol{\delta}^{(k)})\big)^{\mathrm{T}}\hat{\boldsymbol{v}}^{(k)} &\leq \Big(\frac{3}{2}\lambda_\delta + C_v\Big)\|(\boldsymbol{\delta}^{(k)} - \hat{\boldsymbol{\delta}}^{(k)})_{\mathcal{S}_k}\|_1 + b_g^*\|\hat{\boldsymbol{v}}^{(k)}\|_\Sigma \\
&\leq m^{1/2}\Big(\frac{3}{2}\lambda_\delta + C_v\Big)\|\hat{\boldsymbol{v}}^{(k)}\|_2 + b_g^*\|\hat{\boldsymbol{v}}^{(k)}\|_\Sigma.
\end{aligned}
$$

By Proposition 3.3, the RSC of $(\nabla\hat{Q}_g^{(0)}(\hat{\boldsymbol{\omega}}^{(k)} + \hat{\boldsymbol{\delta}}^{(k)}) - \nabla\hat{Q}_g^{(0)}(\hat{\boldsymbol{\omega}}^{(k)} + \boldsymbol{\delta}^{(k)}))^{\mathrm{T}}\hat{\boldsymbol{v}}^{(k)}$, we have

$$
0.1\underline{\kappa}\cdot\underline{f}\|\hat{\boldsymbol{v}}^{(k)}\|_\Sigma^2 \leq m^{1/2}\Big(\frac{3}{2}\lambda_\delta + C_v\Big)\|\hat{\boldsymbol{v}}^{(k)}\|_2 + b_g^*\|\hat{\boldsymbol{v}}^{(k)}\|_\Sigma,
$$

with probability at least $1 - (pn_0)^{-1}$. Therefore, if we let $g^2 \leq m^{1/2}\lambda_\delta$,

$$
\|\hat{\boldsymbol{v}}^{(k)}\|_\Sigma^2 \lesssim \frac{m\log p}{n_0}.
$$

By Lemma 17 in Yuan et al. (2018) and the condition $m \lesssim \sqrt{n_0/\log p}$, we have

$$
\|\tilde{\boldsymbol{\delta}}^{(k)} - \boldsymbol{\delta}^{(k)}\|_\Sigma^2 \lesssim \frac{m\log p}{n_0} \text{ and } \|\tilde{\boldsymbol{\delta}}^{(k)} - \boldsymbol{\delta}^{(k)}\|_1 \lesssim m\sqrt{\frac{\log p}{n_0}}.
$$

For step 3, let $\tilde{\boldsymbol{\delta}}^{(0)} = \boldsymbol{\delta}^{(0)} = 0$, then the loss function in step 3 could be written as:

$$
\frac{1}{n_0 + n_{\mathcal{A}'_m}}\sum_{k\in\{0\}\cup\mathcal{A}'_m}\sum_{i=1}^{n_k} l_w\big(y_i^{(k)} - (\boldsymbol{X}_i^{(k)})^{\mathrm{T}}(\boldsymbol{\beta} - \tilde{\boldsymbol{\delta}}^{(k)})\big) =: \sum_{k\in\{0\}\cup\mathcal{A}'_m}\hat{Q}_w^{(k)}(\boldsymbol{\beta} - \tilde{\boldsymbol{\delta}}^{(k)}).
$$

The symmetric Bregman divergence is defined as

$$
\sum_{k\in\{0\}\cup\mathcal{A}'_m}\big(\nabla\hat{Q}_w^{(k)}(\hat{\boldsymbol{\beta}} - \tilde{\boldsymbol{\delta}}^{(k)}) - \nabla\hat{Q}_w^{(k)}(\boldsymbol{\beta}^* - \tilde{\boldsymbol{\delta}}^{(k)})\big)^{\mathrm{T}}(\hat{\boldsymbol{\beta}} - \boldsymbol{\beta}^*).
$$

To simplify the notations, define $\nabla \hat{R}_w(\boldsymbol{\beta}) = \sum_{k \in \{0\} \cup \mathcal{A}'_m} \nabla \hat{Q}_w^{(k)}(\boldsymbol{\beta} - \tilde{\boldsymbol{\delta}}^{(k)})$. Similarly as above, we have an oracle inequality for $\hat{\boldsymbol{\beta}}$,

$$
\begin{aligned}
&\left(\nabla \hat{R}_w(\hat{\boldsymbol{\beta}}) - \nabla \hat{R}_w(\boldsymbol{\beta}^*)\right)^{\mathrm{T}}(\hat{\boldsymbol{\beta}} - \boldsymbol{\beta}^*) \\
&\leq \lambda_\beta \left(\|\boldsymbol{\beta}^*\|_1 - \|\hat{\boldsymbol{\beta}}\|_1\right) + \sum_{k \in \{0\} \cup \mathcal{A}'_m} \left(\nabla \hat{Q}_w^{(k)}(\boldsymbol{\omega}^{(k)} + \boldsymbol{\delta}^{(k)} - \tilde{\boldsymbol{\delta}}^{(k)}) - \nabla \hat{Q}_w^{(k)}(\boldsymbol{\omega}^{(k)})\right)^{\mathrm{T}}(\boldsymbol{\beta}^* - \hat{\boldsymbol{\beta}}) \\
&\quad + \sum_{k \in \{0\} \cup \mathcal{A}'_m} \left(\nabla \hat{Q}_w^{(k)}(\boldsymbol{\omega}^{(k)}) - \nabla Q_w^{(k)}(\boldsymbol{\omega}^{(k)})\right)^{\mathrm{T}}(\boldsymbol{\beta}^* - \hat{\boldsymbol{\beta}}) + \sum_{k \in \{0\} \cup \mathcal{A}'_m} \left(\nabla Q_w^{(k)}(\boldsymbol{\omega}^{(k)})\right)^{\mathrm{T}}(\boldsymbol{\beta}^* - \hat{\boldsymbol{\beta}}) \\
&\leq \lambda_\beta \left(\|\boldsymbol{\beta}^*\|_1 - \|\hat{\boldsymbol{\beta}}\|_1\right) + \sum_{k \in \{0\} \cup \mathcal{A}'_m} \left(\nabla \hat{Q}_w^{(k)}(\boldsymbol{\omega}^{(k)} + \boldsymbol{\delta}^{(k)} - \tilde{\boldsymbol{\delta}}^{(k)}) - \nabla \hat{Q}_w^{(k)}(\boldsymbol{\omega}^{(k)})\right)^{\mathrm{T}}(\boldsymbol{\beta}^* - \hat{\boldsymbol{\beta}}) \\
&\quad + \sum_{k \in \{0\} \cup \mathcal{A}'_m} \underbrace{\left\|\nabla \hat{Q}_w^{(k)}(\boldsymbol{\omega}^{(k)}) - \nabla Q_w^{(k)}(\boldsymbol{\omega}^{(k)})\right\|_\infty}_{\|\boldsymbol{\pi}_w^{(k)}\|_\infty} \|\boldsymbol{\beta}^* - \hat{\boldsymbol{\beta}}\|_1 \\
&\quad + \sum_{k \in \{0\} \cup \mathcal{A}'_m} \underbrace{\left\|\Sigma^{-1/2} \nabla Q_w^{(k)}(\boldsymbol{\omega}^{(k)})\right\|_2}_{b_w^*} \|\boldsymbol{\beta}^* - \hat{\boldsymbol{\beta}}\|_\Sigma .
\end{aligned}
$$

For the second term above, by Lemma A.4,

$$
\begin{aligned}
&\left(\nabla \hat{Q}_w^{(k)}(\boldsymbol{\omega}^{(k)} + \boldsymbol{\delta}^{(k)} - \tilde{\boldsymbol{\delta}}^{(k)}) - \nabla \hat{Q}_w^{(k)}(\boldsymbol{\omega}^{(k)})\right)^{\mathrm{T}}(\boldsymbol{\beta}^* - \hat{\boldsymbol{\beta}}) \\
&\leq \underbrace{C'\left(\frac{m}{w}\sqrt{\frac{\log p}{n_0}}\sqrt{\frac{\log p}{n_0 + n_{\mathcal{A}'_m}}} + \frac{\log p}{n_0 + n_{\mathcal{A}'_m}} + \sqrt{m}\sqrt{\frac{\log p}{n_0}}\right)}_{C'_v} \|\boldsymbol{\beta}^* - \hat{\boldsymbol{\beta}}\|_1 .
\end{aligned}
$$

If we set $\lambda_\beta \geq 2\|\boldsymbol{\pi}_w^*\|_\infty$ and $C'_v \leq \lambda_\beta/2$, then

$$
\begin{aligned}
&\left(\nabla \hat{R}_w(\hat{\boldsymbol{\beta}}) - \nabla \hat{R}_w(\boldsymbol{\beta}^*)\right)^{\mathrm{T}}(\hat{\boldsymbol{\beta}} - \boldsymbol{\beta}^*) \\
&\leq \left(\frac{3}{2}\lambda_\beta + C'_v\right)\|(\boldsymbol{\beta}^* - \hat{\boldsymbol{\beta}})_{\mathcal{S}}\|_1 + b_w^*\|\boldsymbol{\beta}^* - \hat{\boldsymbol{\beta}}\|_\Sigma \\
&\leq s^{1/2}\left(\frac{3}{2}\lambda_\beta + C'_v\right)\|\boldsymbol{\beta}^* - \hat{\boldsymbol{\beta}}\|_2 + b_w^*\|\boldsymbol{\beta}^* - \hat{\boldsymbol{\beta}}\|_\Sigma .
\end{aligned}
$$

Under the RSC of $(\nabla \hat{R}_w(\hat{\boldsymbol{\beta}}) - \nabla \hat{R}_w(\boldsymbol{\beta}^*))^{\mathrm{T}}(\hat{\boldsymbol{\beta}} - \boldsymbol{\beta}^*)$, we have

$$
\left(\nabla \hat{R}_w(\hat{\boldsymbol{\beta}}) - \nabla \hat{R}_w(\boldsymbol{\beta}^*)\right)^{\mathrm{T}}(\hat{\boldsymbol{\beta}} - \boldsymbol{\beta}^*) \geq c_1 \|\boldsymbol{\beta}^* - \hat{\boldsymbol{\beta}}\|_\Sigma^2,
$$

with probability as least $1 - (pn)^{-1}$, where $n = n_0 + n_{\mathcal{A}'_m}$ and $c_1$ is a positive constant. The proof of the RSC in step 3 is similar to Proposition 3.3. Thus,

$$
\|\boldsymbol{\beta}^* - \hat{\boldsymbol{\beta}}\|_\Sigma \leq s^{1/2}\left(\frac{3}{2}\lambda_\beta + C'_v\right)\gamma_p^{-1/2} + b_w^* .
$$

Through a similar proof as Lemma A.2, we obtain $\lambda_\beta \lesssim \sqrt{\log(p)/n}$. If $s\log(p)/n \leq w^2 \leq s^{1/2}\lambda_\beta$, we have

$$
\|\boldsymbol{\beta}^* - \hat{\boldsymbol{\beta}}\|_\Sigma \lesssim \sqrt{\frac{s\log p}{n}} + \sqrt{\frac{sm\log p}{n_0}} \text{ and } \|\boldsymbol{\beta}^* - \hat{\boldsymbol{\beta}}\|_1 \lesssim s\sqrt{\frac{\log p}{n}} + s\sqrt{\frac{m\log p}{n_0}},
$$

with probability at least $1 - p^{-1}$.

# B Proof of Lemmas

## B.1 Proof of Lemma A.1

Define $\boldsymbol{\omega}^{(k)}$ for all $0 \leq k \leq \mathcal{K}$ as the true parameters of each local source model, then note that $\nabla Q^{(k)}(\boldsymbol{\omega}^{(k)}) = 0$ and $\nabla Q(\boldsymbol{\omega}^*) = \sum_{k=0}^{\mathcal{K}} \alpha_k \nabla Q^{(k)}(\boldsymbol{\omega}^*) = 0$. So we have

$$\nabla Q(\boldsymbol{\omega}^*) - \nabla Q(\boldsymbol{\beta}^*) + \nabla Q(\boldsymbol{\beta}^*) - \sum_{k=1}^{\mathcal{K}} \alpha_k \nabla Q^{(k)}(\boldsymbol{\omega}^{(k)}) = 0$$

$$\nabla Q(\boldsymbol{\omega}^*) - \nabla Q(\boldsymbol{\beta}^*) = \sum_{k=1}^{\mathcal{K}} \alpha_k \nabla Q^{(k)}(\boldsymbol{\omega}^{(k)}) - \nabla Q(\boldsymbol{\beta}^*)$$

Note that $\nabla Q^{(0)}(\boldsymbol{\omega}^{(0)}) = Q^{(0)}(\boldsymbol{\beta}^*) = 0$, so

$$\sum_{k=0}^{\mathcal{K}} \alpha_k (\nabla Q^{(k)}(\boldsymbol{\omega}^*) - \nabla Q^{(k)}(\boldsymbol{\beta}^*)) = \sum_{k=1}^{\mathcal{K}} \alpha_k (\nabla Q^{(k)}(\boldsymbol{\omega}^{(k)}) - \nabla Q^{(k)}(\boldsymbol{\beta}^*))$$

By the second-order Taylor expansions and Assumption 3.4,

$$\sum_{k=0}^{\mathcal{K}} \alpha_k \int_0^1 \nabla^2 Q^{(k)}((1-t)\boldsymbol{\beta}^* + t\boldsymbol{\omega}^*) dt(\boldsymbol{\omega}^* - \boldsymbol{\beta}^*) = \sum_{k=1}^{\mathcal{K}} \alpha_k \int_0^1 \nabla^2 Q^{(k)}((1-t)\boldsymbol{\beta}^* + t\boldsymbol{\omega}^{(k)}) dt(\boldsymbol{\omega}^{(k)} - \boldsymbol{\beta}^*)$$

$$||\boldsymbol{\omega}^* - \boldsymbol{\beta}^*||_1 \leq \sum_{k=1}^{\mathcal{K}} \alpha_k ||\tilde{\Sigma}^{-1} \tilde{\Sigma}^{(k)}||_1 \cdot ||\boldsymbol{\omega}^{(k)} - \boldsymbol{\beta}^*||_1.$$

By the definition of the parameter space

$$\Theta(s, m) = \left\{ \boldsymbol{\beta}^*, \{\boldsymbol{\omega}^{(k)}\} : ||\boldsymbol{\beta}^*||_0 \leq s, \sup_{k \in \mathcal{A}_m} ||\boldsymbol{\omega}^{(k)} - \boldsymbol{\beta}^*||_1 \leq m \right\},$$

We have $||\boldsymbol{\omega}^{(k)} - \boldsymbol{\beta}^*||_1 \leq m$. Let $C_1 = \sup_k ||\tilde{\Sigma}^{-1} \tilde{\Sigma}^{(k)}||_1$. Then Lemma A.1 is proved.

## B.2 Proof of Lemma A.2

For the transferring steps,

$$\nabla \hat{Q}_h(\boldsymbol{\omega}) = \frac{1}{n_{\mathcal{A}_m} + n_0} \sum_{k=0}^{\mathcal{K}} \sum_{i=1}^{n_k} \left\{ \bar{K}\left( \frac{(\boldsymbol{x}_i^{(k)})^{\mathrm{T}} \boldsymbol{\omega} - y_i^{(k)}}{h} \right) - \tau \right\} \boldsymbol{x}_i^{(k)}$$

$$\nabla^2 \hat{Q}_h(\boldsymbol{\omega}) = \frac{1}{n_{\mathcal{A}_m} + n_0} \sum_{k=0}^{\mathcal{K}} \sum_{i=1}^{n_k} K\left( \frac{(\boldsymbol{x}_i^{(k)})^{\mathrm{T}} \boldsymbol{\omega} - y_i^{(k)}}{h} \right) \boldsymbol{x}_i^{(k)} (\boldsymbol{x}_i^{(k)})^{\mathrm{T}}.$$

Let $\xi_i^{(k)} = \bar{K}\{((\boldsymbol{x}_i^{(k)})^{\mathrm{T}} \boldsymbol{\omega} - y_i^{(k)})/h\} - \tau$, then $\nabla \hat{Q}_h(\boldsymbol{\omega}) = (n_{\mathcal{A}_m} + n_0)^{-1} \sum_{k=0}^{\mathcal{K}} \sum_{i=1}^{n_k} \xi_i^{(k)} \boldsymbol{x}_i^{(k)}$ and

$$||\boldsymbol{\pi}_h^*||_\infty = \left\| \frac{1}{n_{\mathcal{A}_m} + n_0} \sum_{k=0}^{\mathcal{K}} \sum_{i=1}^{n_k} \{\xi_i^{(k)} \boldsymbol{x}_i^{(k)} - \mathbb{E}(\xi_i^{(k)} \boldsymbol{x}_i^{(k)})\} \right\|_\infty.$$

The upper bound of $||\boldsymbol{\pi}_h^*||_\infty$ involves two quantities that are related to

$$\mathbb{E}\left[ \bar{K}^2 \left( \frac{(\boldsymbol{x}^{(k)})^{\mathrm{T}} (\boldsymbol{\omega} - \boldsymbol{\omega}^{(k)}) - \epsilon}{h} \right) \Big| \boldsymbol{x}^{(k)} \right] \text{ and } \mathbb{E}\left[ \left( \frac{(\boldsymbol{x}^{(k)})^{\mathrm{T}} (\boldsymbol{\omega} - \boldsymbol{\omega}^{(k)}) - \epsilon}{h} \right) \Big| \boldsymbol{x}^{(k)} \right].$$

For the first term, by a change of variable and integration by parts, we obtain

$$
\mathbb{E}\left[\bar{K}^2\left(\frac{(\boldsymbol{x}^{(k)})^{\mathrm{T}}(\boldsymbol{\omega} - \boldsymbol{\omega}^{(k)}) - \epsilon}{h}\right)\bigg|\boldsymbol{x}^{(k)}\right] = \int_{-\infty}^{\infty} \bar{K}^2(-u/h)f_{\epsilon|\boldsymbol{x}}(u)du
$$

$$
= h\int_{-\infty}^{\infty} \bar{K}^2(v)f_{\epsilon|\boldsymbol{x}}(-vh)dv
$$

$$
= 2\int_{-\infty}^{\infty} K(v)\bar{K}(v)F_{\epsilon|\boldsymbol{x}}(-vh)dv. \tag{50}
$$

By the fact that $F_{\epsilon|\boldsymbol{x}}(0) = \tau$, we have

$$
F_{\epsilon|\boldsymbol{x}}(-vh) = F_{\epsilon|\boldsymbol{x}}(0) + \int_0^{-vh} f_{\epsilon|\boldsymbol{x}}(t)dt
$$

$$
= \tau - hvf_{\epsilon|\boldsymbol{x}}(0) + \int_0^{-vh}\{f_{\epsilon|\boldsymbol{x}}(t) - f_{\epsilon|\boldsymbol{x}}(0)\}dt. \tag{51}
$$

Moreover, it can be shown that

$$
a_K := \int_{-\infty}^{\infty} vK(v)\bar{K}(v)dv = \int_0^{\infty} K(v)\{1 - K(v)\}dv > 0 \text{ and } a_K \leq \kappa_1, \tag{52}
$$

where $\kappa_1 = \int |u|K(u)du$.

Substituting (51) into (50), and by (52), we obtain

$$
\mathbb{E}\left[\bar{K}^2\left(\frac{(\boldsymbol{x}^{(k)})^{\mathrm{T}}(\boldsymbol{\omega} - \boldsymbol{\omega}^{(k)}) - \epsilon}{h}\right)\bigg|\boldsymbol{x}^{(k)}\right] = 2\tau\int_{-\infty}^{\infty} K(v)\bar{K}(v)dv - 2hf_{\epsilon|\boldsymbol{x}}(0)\int_{-\infty}^{\infty} vK(v)\bar{K}(v)dv
$$

$$
+ 2\int_{-\infty}^{\infty}\int_0^{-vh}\{f_{\epsilon|\boldsymbol{x}}(t) - f_{\epsilon|\boldsymbol{x}}(0)\}K(v)\bar{K}(v)dtdv
$$

$$
\leq \tau - 2a_K h f_{\epsilon|\boldsymbol{x}}(0) + l_0 h^2\int_{-\infty}^{\infty} v^2 K(v)\bar{K}(v)dv
$$

$$
\leq \tau + l_0\kappa_2 h^2,
$$

where the first inequality holds using the Lipschitz condition on $f_{\epsilon|\boldsymbol{x}}$ in Assumption 3.1 and the last inequality holds by Assumption 3.2. Through a similar calculation, the Lipschitz condition on $f_{\epsilon|\boldsymbol{x}}$ ensures that

$$
\left|\mathbb{E}\left[\left(\frac{(\boldsymbol{x}^{(k)})^{\mathrm{T}}(\boldsymbol{\omega} - \boldsymbol{\omega}^{(k)}) - \epsilon}{h}\right)\bigg|\boldsymbol{x}^{(k)}\right] - \tau\right| \leq \frac{l_0}{2}\kappa_2 h^2.
$$

Hence

$$
\mathbb{E}(\xi_i^{(k)}x_{ij}^{(k)})^2 = \mathbb{E}_x\left\{(x_{ij}^{(k)})^2 \cdot \mathbb{E}((\xi_i^{(k)})^2|\boldsymbol{x}_i^{(k)})\right\}
$$

$$
\mathbb{E}(\xi^2|\boldsymbol{x}) = \mathbb{E}\left[\left(\bar{K}\left(\frac{\boldsymbol{x}^{\mathrm{T}}\boldsymbol{\omega} - y}{h}\right) - \tau\right)^2\bigg|\boldsymbol{x}\right]
$$

$$
= \underbrace{\mathbb{E}\left[\bar{K}^2\left(\frac{\boldsymbol{x}^{\mathrm{T}}\boldsymbol{\omega} - y}{h}\right)\bigg|\boldsymbol{x}\right]}_{\leq \tau + l_0\kappa_2 h^2} - 2\tau\underbrace{\mathbb{E}\left[\bar{K}\left(\frac{\boldsymbol{x}^{\mathrm{T}}\boldsymbol{\omega} - y}{h}\right)\bigg|\boldsymbol{x}\right]}_{\geq \tau - \frac{l_0}{2}\kappa_2 h^2} + \tau^2
$$

$$
\leq \tau(1 - \tau) + Ch^2,
$$

where $C = (\tau + 1)l_0\kappa_2$. Then, by Assumption 4.3, we have

$$
\mathbb{E}(\xi_i^{(k)}x_{ij}^{(k)})^2 \leq \tau(1 - \tau)\sigma_{jj} + C\sigma_{jj}h^2.
$$

Also by Assumption 3.3 and $|\xi_i^{(k)}| \leq \max(1 - \tau, \tau)$, for $d = 3, 4, \ldots,$

$$
\begin{aligned}
\mathbb{E}(|\xi_i^{(k)} x_{ij}^{(k)}|^d) &\leq \max(1 - \tau, \tau)^{d-2} \mathbb{E}_{\boldsymbol{x}}\{|x_{ij}^{(k)}|^d \cdot \mathbb{E}[(\xi_i^{(k)})^2 | \boldsymbol{x}_i^{(k)}]\} \\
&\leq \max(1 - \tau, \tau)^{d-2}\{\tau(1 - \tau) + Ch^2\} \\
&\leq \frac{d!}{2}\{\tau(1 - \tau) + Ch^2\} \max(1 - \tau, \tau)^{d-2}.
\end{aligned}
$$

Thus it follows from Bernstein's inequality and union bound that for every $t \geq 0$,

$$
||\boldsymbol{\pi}_h^*||_\infty \leq \sigma\sqrt{\{\tau(1 - \tau) + Ch^2\}\frac{2t}{n_{\mathcal{A}_m} + n_0}} + \max(1 - \tau, \tau)\frac{t}{n_{\mathcal{A}_m} + n_0}
$$

with probability at least $1 - 2pe^{-t}$.

For the debiasing step, through the similar proof we could get same results with different sample size and smoothing bandwidth.

### B.3 Proof of Lemma A.3

Note that

$$
\begin{aligned}
b_h^* &= ||\Sigma^{-1/2}\nabla Q_h(\boldsymbol{\omega}^*)||_2 \\
&= \left\|\Sigma^{-1/2}\left(\sum_{k=0}^{\mathcal{K}} \alpha_k \mathbb{E}\left[\mathbb{E}\left\{\bar{K}\left(\frac{(\boldsymbol{x}^{(k)})^{\mathrm{T}}\boldsymbol{\omega}^* - y^{(k)}}{h}\right) - \tau \,\middle|\, \boldsymbol{x}^{(k)}\right\}\boldsymbol{x}^{(k)}\right]\right)\right\|_2 \\
&\leq \sup_{\boldsymbol{u} \in \mathbb{S}^{p-1}} \sum_{k=0}^{\mathcal{K}} \mathbb{E}\left[\bar{K}\left(\frac{(\boldsymbol{x}^{(k)})^{\mathrm{T}}\boldsymbol{\omega}^* - y^{(k)}}{h}\right) - \tau\right](\Sigma^{-1/2}\boldsymbol{x}^{(k)})^{\mathrm{T}}\boldsymbol{u} \\
&\leq \frac{l_0}{2}\kappa_2 h^2.
\end{aligned}
$$

### B.4 Proof of Lemma A.4

For $k = 1, \ldots, p$, define that

$$
\psi_k(r, l) = \sup_{\boldsymbol{v} \in \mathbb{B}_\Sigma(r) \cap \mathbb{B}_1(l)} \left|\frac{1}{n}\sum_{i=1}^{n}(1 - \mathbb{E})\underbrace{\{\bar{K}_h(\boldsymbol{x}_i^{\mathrm{T}}\boldsymbol{v} - \epsilon_i) - \bar{K}_h(-\epsilon_i)\}x_{ik}}_{=:g_{\boldsymbol{v},k}(y_i, \boldsymbol{x}_i)}\right|,
$$

where $\boldsymbol{v} = \boldsymbol{\beta} - \boldsymbol{\beta}^*$. Note that $\psi(r, l) \leq \max_{1 \leq k \leq p}\{\psi_k(r, l) + |\mathbb{E}g_{\boldsymbol{v},k}(y_i, \boldsymbol{x}_i)|\}$. In the following, we bound $\psi_k(r, l)$ and $\mathbb{E}g_{\boldsymbol{v},k}(y_i, \boldsymbol{x}_i)$, respectively.

Let $\sigma$ be any positive constant such that $\sigma^2 \geq \sup_{\boldsymbol{v} \in \mathbb{B}_\Sigma(r) \cap \mathbb{B}_1(l)} \mathbb{E}g_{\boldsymbol{v},k}^2(y_i, \boldsymbol{x}_i)$. By the bounded design, we note that $\sup_{\boldsymbol{v}} |g_{\boldsymbol{v},k}(y_i, \boldsymbol{x}_i)| \leq |x_{ik}| \leq 1$. Applying Theorem 7.3 in Bousquet (2003), Bousquet's version of Talagrand's inequality, we obtain that for any $z > 0$,

$$
\psi_k(r, l) \leq \mathbb{E}\psi_k(r, l) + \sqrt{\{\sigma^2 + 2\mathbb{E}\psi_k(r, l)\}\frac{2z}{n}} + \frac{z}{3n} \tag{53}
$$

holds with probability at least $1 - e^{-z}$. For the second moment $\mathbb{E}g_{\boldsymbol{v},k}^2(y_i, \boldsymbol{x}_i)$, by a change of variable and Minkowski's integral inequality we derive that

$$
\begin{aligned}
\mathbb{E}g_{\boldsymbol{v},k}^2(y_i, \boldsymbol{x}_i) &= \mathbb{E}\left[ x_{ik}^2 \int_{-\infty}^{\infty} \left\{ \bar{K}_h(\boldsymbol{x}_i^{\mathrm{T}}\boldsymbol{v} - t) - \bar{K}_h(-t) \right\}^2 f_{\epsilon_i|\boldsymbol{x}_i}(t)dt \right] \\
&= \mathbb{E}\left[ x_{ik}^2 \int_{-\infty}^{\infty} \left\{ \bar{K}_h(u) - \bar{K}_h(u - \boldsymbol{x}_i^{\mathrm{T}}\boldsymbol{v}) \right\}^2 f_{\epsilon_i|\boldsymbol{x}_i}(\boldsymbol{x}_i^{\mathrm{T}}\boldsymbol{v} - u)du \right] \\
&= h\mathbb{E}\left[ x_{ik}^2 \int_{-\infty}^{\infty} \left\{ \bar{K}(v) - \bar{K}(v - \boldsymbol{x}_i^{\mathrm{T}}\boldsymbol{v}/h) \right\}^2 f_{\epsilon_i|\boldsymbol{x}_i}(\boldsymbol{x}_i^{\mathrm{T}}\boldsymbol{v} - vh)dv \right] \\
&\leq \bar{f}h^{-1}\mathbb{E}\left[ x_{ik}^2 (\boldsymbol{x}_i^{\mathrm{T}}\boldsymbol{v})^2 \int_{-\infty}^{\infty} \left\{ \int_0^1 K(v - w\boldsymbol{x}_i^{\mathrm{T}}\boldsymbol{v}/h)dw \right\}^2 dv \right] \\
&\leq \bar{f}h^{-1}\mathbb{E}\left( x_{ik}^2 (\boldsymbol{x}_i^{\mathrm{T}}\boldsymbol{v})^2 \left[ \int_0^1 \left\{ \int_{-\infty}^{\infty} K^2(v - w\boldsymbol{x}_i^{\mathrm{T}}\boldsymbol{v}/h)dv \right\}^{1/2} dw \right]^2 \right) \\
&\leq \bar{\kappa}\bar{f}h^{-1}\mathbb{E}(x_{ik} \cdot \boldsymbol{x}_i^{\mathrm{T}}\boldsymbol{v})^2 \leq \bar{\kappa}\bar{f}h^{-1}r^2.
\end{aligned}
$$

It remains to bound $\mathbb{E}\psi_k(r,l)$. Note that $|g_{\boldsymbol{v},k}(y_i, \boldsymbol{x}_i) - g_{\boldsymbol{v}',k}(y_i, \boldsymbol{x}_i)| \leq (\bar{\kappa}/h)|\boldsymbol{x}_i^{\mathrm{T}}\boldsymbol{v} - \boldsymbol{x}_i^{\mathrm{T}}\boldsymbol{v}'|$, for any $\boldsymbol{v}, \boldsymbol{v}'$. Hence using Rademacher symmetrization and Talagrand's contraction principle, we have

$$
\begin{aligned}
\mathbb{E}\psi_k(r,l) &\leq 2\mathbb{E}\left[ \sup_{\boldsymbol{v}\in\mathbb{B}_\Sigma(r)\cap\mathbb{B}_1(l)} \left| \frac{1}{n}\sum_{i=1}^n e_i g_{\boldsymbol{v},k}(y_i, \boldsymbol{x}_i) \right| \right] \\
&\leq 4\bar{\kappa}\mathbb{E}\left[ \sup_{\boldsymbol{v}\in\mathbb{B}_\Sigma(r)\cap\mathbb{B}_1(l)} \left| \frac{1}{nh}\sum_{i=1}^n e_i \boldsymbol{x}_i^{\mathrm{T}}\boldsymbol{v} \right| \right] \leq 4\bar{\kappa}\frac{l}{h}\mathbb{E}\left\| \frac{1}{n}\sum_{i=1}^n e_i \boldsymbol{x}_i \right\|_\infty,
\end{aligned} \tag{54}
$$

where $e_1, \ldots, e_n$ are independent Rademacher variables. Applying Hoeffding's moment inequality,

$$
\mathbb{E}_e \left\| \frac{1}{n}\sum_{i=1}^n e_i \boldsymbol{x}_i \right\|_\infty \leq \max_{1\leq k\leq p}\left( \sum_{i=1}^n x_{ik}^2 \right)^{1/2} \frac{\sqrt{2\log(2p)}}{n}, \tag{55}
$$

where $\mathbb{E}_e$ denotes the expectation over $\{e_i\}_{i=1}^n$. By (54) and (55), we obtain

$$
\mathbb{E}\psi_k(r,l) \leq 4\bar{\kappa}\frac{l}{h}\sqrt{\frac{2\log(2p)}{n}}.
$$

Taking $z = t + \log p$ in (53), we have that

$$
\psi_k(r,l) \lesssim \frac{l}{h}\sqrt{\frac{\log p}{n}} + \bar{f}^{1/2}r\sqrt{\frac{t + \log p}{nh}} + \frac{t + \log p}{n} \tag{56}
$$

holds with probability at least $1 - e^{-t}$.

Next we find an union upper bound of $|\mathbb{E}g_{\boldsymbol{v},k}(y_i, \boldsymbol{x}_i)|$. Similarly as the method to bound the second moment, we derive that

$$
\begin{aligned}
\mathbb{E}g_{\boldsymbol{v},k}(y_i, \boldsymbol{x}_i) &= h\mathbb{E}\left[ x_{ik} \int_{-\infty}^{\infty} \left\{ \bar{K}(v) - \bar{K}(v - \boldsymbol{x}_i^{\mathrm{T}}\boldsymbol{v}/h) \right\} f_{\epsilon_i|\boldsymbol{x}_i}(\boldsymbol{x}_i^{\mathrm{T}}\boldsymbol{v} - vh)dv \right] \\
&\leq \bar{f}\mathbb{E}\left[ |x_{ik}||\boldsymbol{x}_i^{\mathrm{T}}\boldsymbol{v}| \int_{-\infty}^{\infty} \left\{ \int_0^1 K(v - w\boldsymbol{x}_i^{\mathrm{T}}\boldsymbol{v}/h)dw \right\}dv \right] \\
&\leq \bar{f}\mathbb{E}\left( |x_{ik}||\boldsymbol{x}_i^{\mathrm{T}}\boldsymbol{v}| \left[ \int_0^1 \left\{ \int_{-\infty}^{\infty} K(v - w\boldsymbol{x}_i^{\mathrm{T}}\boldsymbol{v}/h)dv \right\}dw \right] \right) \\
&\leq \bar{\kappa}\bar{f}\mathbb{E}|x_{ik}\boldsymbol{x}_i^{\mathrm{T}}\boldsymbol{v}| \leq \bar{\kappa}\bar{f}r.
\end{aligned}
$$

Finally taking the union bound, we obtain that with probability at least $1 - e^{-t}$,

$$\psi(r,l) \lesssim \frac{l}{h}\sqrt{\frac{\log p}{n}} + \bar{f}^{1/2}r\sqrt{\frac{t + \log p}{nh}} + \frac{t + \log p}{n}.$$

## C  Empirical results on the similarity of target and source data

This section presents the empirical results of the relationship between target and source data with different degrees of similarity to the target. For the similarity defined in $\ell_1$-norm of the contrast of each source, we consider $p = 500$ and the sample size of the target and each source is 400. For the transferable source $k$, we let $\boldsymbol{\omega}^{(k)} = \boldsymbol{\beta}^* + (m/p)\boldsymbol{\mathcal{R}}_p^{(k)}$ to satisfy the transferring level $||\boldsymbol{\delta}^{(k)}||_1 = ||\boldsymbol{\omega}^{(k)} - \boldsymbol{\beta}^*||_1 \leq m$, where $\boldsymbol{\beta}^*$ is the target parameter, $\boldsymbol{\mathcal{R}}_p^{(k)}$ is a vector of $p$ independent Rademacher variables, and $m = 10$. For any source data $k'$ that is not transferable, we let $\boldsymbol{\omega}^{(k')} = \boldsymbol{\beta}^* + (2m/p)\boldsymbol{\mathcal{R}}_p^{(k')}$. All the other settings are the same as the numerical studies in section 4

Figure 6 and 7 illustrate that when the contrast is relatively small in $\ell_1$-norm, there is significant overlap between the target and source data. When the contrast is relatively large, the source data would have much more frequency at the two tails, which may cause the negative transfer if those sources are used in transfer learning.

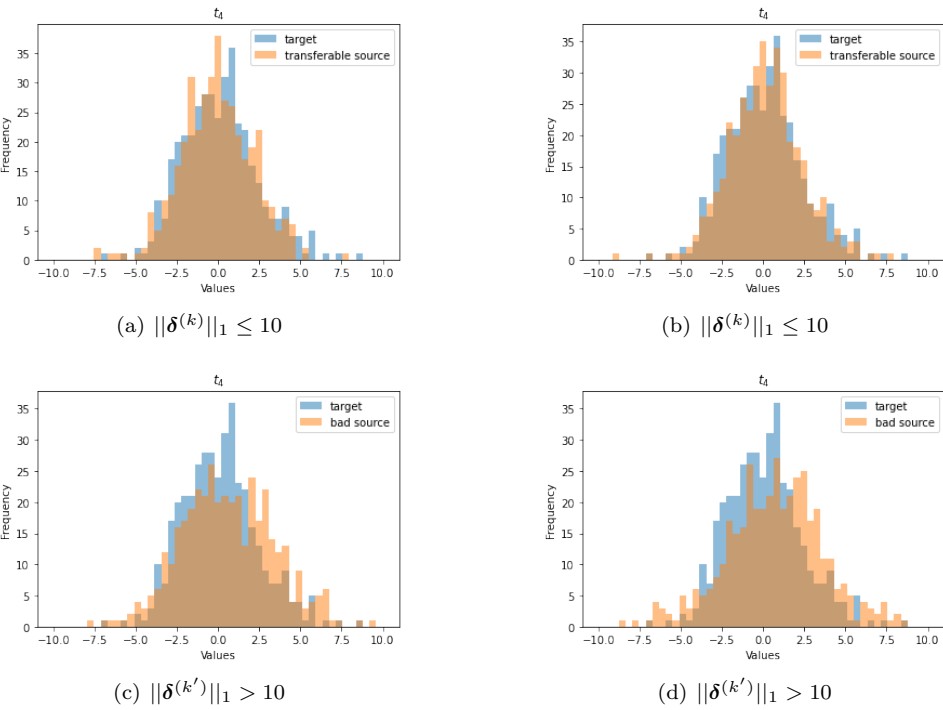

(a) $||\boldsymbol{\delta}^{(k)}||_1 \leq 10$

(b) $||\boldsymbol{\delta}^{(k)}||_1 \leq 10$

(c) $||\boldsymbol{\delta}^{(k')}||_1 > 10$

(d) $||\boldsymbol{\delta}^{(k')}||_1 > 10$

Figure 6: The predictors are from $t$-distributions with 4 degrees of freedom.

For the similarity defined in $\ell_0$-norm of the contrast of each source, we consider $p = 500$ and the sample size of the target and each source is 400. For the transferable source $k$, $\boldsymbol{\omega}^{(k)}$ is generated from $\omega_j^{(k)} = \beta_j^* + 2 \cdot \mathbb{1}(j \in M)$, where $M$ is a random subset of $[p]$ with $|M| = 2$. For the source $k'$ that is not transferable, $\boldsymbol{\omega}^{(k')}$ is generated from $\omega_j^{(k')} = \beta_j^* + 2 \cdot \mathbb{1}(j \in M')$, where $M'$ is a random subset of $[p]$ with $|M'| = 4$.

Figure 8 and 9 demonstrate that with a relatively small contrast in $\ell_0$-norm, most of the target and source data also overlap. However, it is easy to observe that the source data has a distribution with a relatively long tail. Conversely, when the contrast becomes larger, the distribution of the source data at the tail becomes more distinct from the distribution of the target.

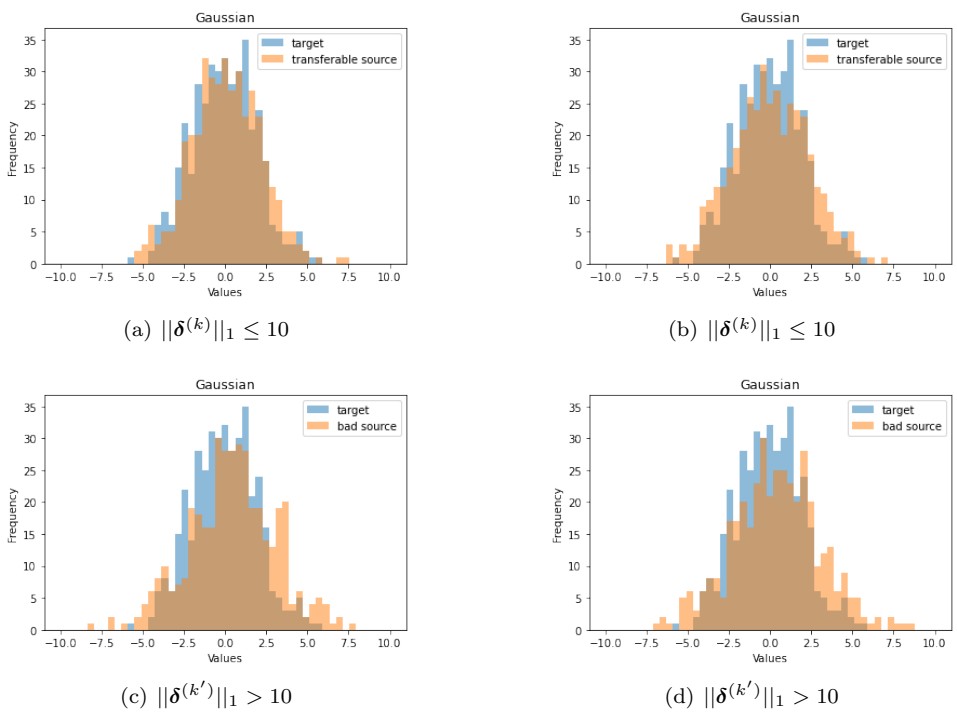

Figure 7: The predictors are from Gaussian distributions.

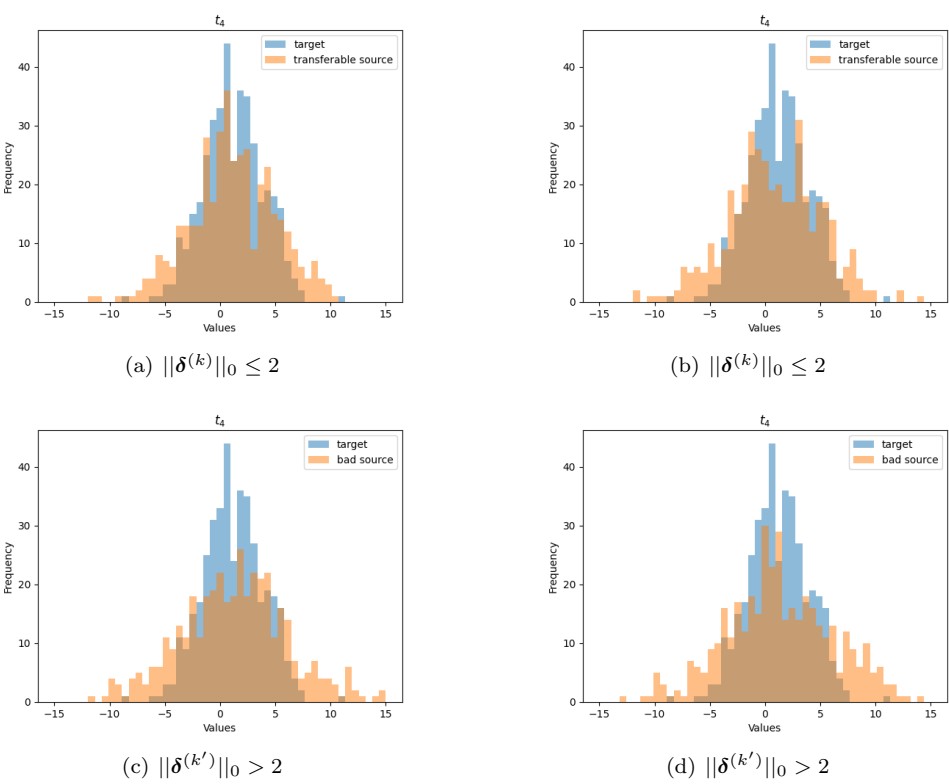

Figure 8: The predictors are from $t$-distributions with 4 degrees of freedom.

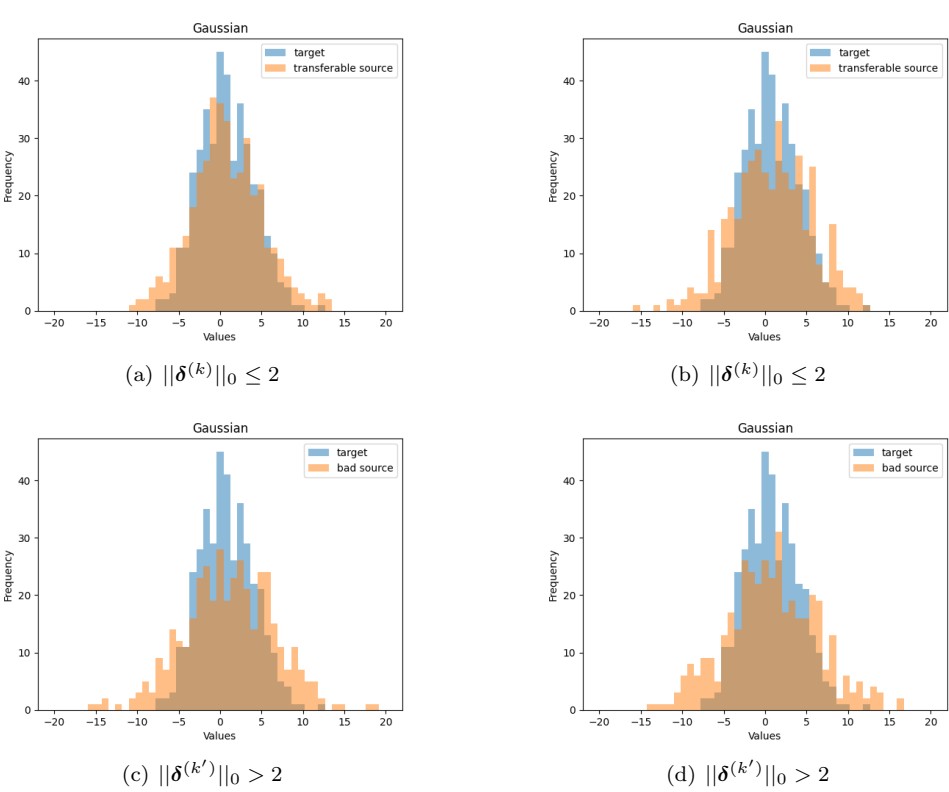

Figure 9: The predictors are from Gaussian distributions.

