# OpenReview forum: "Transfer Learning for High-dimensional Quantile Regression with Statistical Guarantee"
_TMLR — Accepted by TMLR_

### Review · Reviewer_JxJL · 2023-07-28

**Summary Of Contributions:**

The learning problem of interest here is quantile regression (high-dimensional, in a sparsity scenario), and the main research question of interest is pinning down sufficient conditions for "domain transfer" to be beneficial, i.e., to provide stronger guarantees (error bounds) than are possible with just the "target" data alone.

The main substantive content of this paper is as follows:

- Concrete algorithms for transfer learning in the context of (sparse) quantile regression. The main procedures are Algorithms 3 and 4. The key underlying difference is in the assumed "closeness" between the sources and the target; the latter assumes an $\\ell\_{0}$-norm bound rather than the more traditional $\\ell\_{1}$-norm bound. The procedures themselves are quite clear and based on similar strategies in the literature, essentially finding good solutions on the source domains, and then since domain "closeness" is characterized here by the difference in underlying weight vectors, this difference is then estimated (using target data for comparison), and finally used to correct the learned source weights before integrating into the final target estimator.

- High-probability (target domain) error bounds for estimators based on the aforementioned algorithms (I'm not certain about this; see my later comments).

- Some basic simulated numerical tests, comparing their proposed procedure with some natural benchmarks, in particular looking at the weak/strong scenarios of interest ($\\ell\_{1}$ versus $\\ell\_{0}$ bounds on the index of "good" sources, denoted $\\mathcal{A}\_{m}$).

**Audience:**

No

**Broader Impact Concerns:**

Not applicable.

**Claims And Evidence:**

No

**Requested Changes:**

Please see the comments above. Taking everything together, I feel like the authors have made an honest effort, but the presentation is way below the standard that one would expect. TMLR does not require the results to be "significant," but at this point it is not really even clear how the material here differs from Zhang and Zhu 2022, aside from treating the "lucky" situation in which we have a $\\ell_{0}$ bound on the contrast (plus oracle information?). My personal opinion is that clearing all this up is quite close to what I would call a "major revision."

**Strengths And Weaknesses:**

The problem setting of this paper is clearly described, and is a natural area of interest for machine learning researchers. The related literature is also described quite meticulously, and the place of the present paper within this context is for the most part clear. The basic formulation is also easy to understand, and the basic algorithmic strategies undertaken are intuitive.

On the other hand, I found certain parts of the overall logic underlying this paper quite puzzling, and while at a macroscopic level the paper is well-organized and clear, a more microscopic look at it yields all sorts of notational issues; the formal material is really quite sloppy in my opinion. Since the paper basically just re-appropriates existing algorithms and analytical machinery to the quantile regression setting, a certain degree of effort needs to be expended to ensure the reader understands what the results in this paper teach us that was not already known before.

I will try my best to elaborate on the points which are lacking in a constructive but concise fashion below, using some concrete examples from the text.

- One major point I tripped up on was all the talk related to *convexity*. The authors say that using the traditional "kernel smoothing method" (p.3), the loss is "still non-convex," and use this to motivate their "convolution-type" smoother, which they say is "not only convex but also differentiable." Maybe I'm missing something basic here, but the pinball loss $\\rho\_{\\tau}(\\cdot)$ is convex, correct? Is it *strong* convexity (i.e., a quadratic growth) the authors want? In any case, the logic here seems quite faulty. The loss is pretty nice in that it is Lipschitz, though. If *smoothness* (i.e., a Lipschitz gradient) is the critical point, then yes, the pinball loss here is not smooth, but no gradient-based optimizers are actually studied here, so how this technical backdrop actually links to the methods studied here is not clear to me.

- If the authors want to focus on convexity, they should be more explicit about what properties their proposed objective has, and when. Is the "empirical smoothed loss function" defined at the bottom of page 3 convex for any function $K\_{h}(\\cdot)$ that is non-negative, integrable, and symmetric about zero?

- Several quantities appear before they are defined. Here are a few examples.
  - The "contrast" $\\boldsymbol{\\delta}^{\\ast}$ (p.1).
  - Matrix $\\Sigma$ used in defining the ball and cone on page 2 (really, these aren't needed until much later...).

- The same notation is used for different objects in various locations. This should really be avoided. Here are some examples.
  - $K$ is both the number of source data sets (p.4) and the kernel function.
  - The notation $\\hat{Q}\_{h}(\\cdot)$ is used with different meanings (e.g., p.3 and p.4). This is a critical quantity; such ambiguity severely reduces clarity and is really just sloppy exposition. It appears again on p.6 with a different meaning; the one on p.6 really should be $\\hat{Q}\_{h}^{(k)}(\\cdot)$, I think this is obvious.
  - In Assumption 3.2, $u$ is used as a label on the upper bound $\\kappa\_{u}$ and also the variable being maxed over on the left-hand side of the inequality; more semantic inconsistency.

- I find it rather hard to parse the main theorems, as well as the related propositions. Here are a few things I tripped up on.
  - In Prop 3.1 and 3.2, are the quantities $r$, $l$, $\\boldsymbol{\\beta}\_{1}$, and $\\boldsymbol{\\beta}\_{2}$ arbitrary, up to the inequalities holding? Nothing is said about them. Even more frustrating is that real-valued $l$ appears alongside $f\_{l}$, where the subscript $l$ is just a label representing "lower" I assume (from Assumption 3.1). Again, same symbol, totally different meanings. Really must be avoided.
  - What is $\\alpha\_{1}$ in Prop 3.2?
  - Most critically, what is $\\hat{\\boldsymbol{\\beta}}$ in Theorems 3.1 and 3.2? From the context, one would like to assume this corresponds precisely to the outputs of Algorithms 3 and 4 respectively; is this so? Even if this is the case, it feels like there is a conceptual gap in that the Theorem 3.2 benefits not just from the stronger "closeness" assumption used in defining $\\mathcal{A}\_{m}^{\\prime}$, but that it is also an "oracle" (i.e., this index of good sources is known).

- Early in the paper, the authors compare their work with existing papers at a detailed technical level, but I think this could use some additional exposition. Here are two points:
  - First, in comparing the $s^{2}$ versus $s^{3}$ factors (paragraph 2, page 2), the authors say that "this is due to the bandwidth involved in the smoothing method." What does this mean? The main results in this paper assume the bandwidth is optimally set, correct? Naively, I think some readers will assume that the difference is due to the fact that bias is introduced by using the smoothed objective instead of the traditional quadratic regression objective; is this not the case? One expects that there is a tradeoff - a benefit due to smoothness, and a takeway due to bias - and that these are balanced in the analysis, but this is unclear at present.
  - Also on page 2, the authors say that compared with Zhang and Zhu 2022, they do not require a particular "restrictive condition" on the kernel function; why is this? What differs here? It isn't just due to the stronger $\\ell\_{0}$-norm contrast assumptions, I assume?

Minor issues:
- Quotation marks are vertical, but really should be slanted.
- "the logistic of the algorithm..." (p.4), maybe should be "logic"?
- Asterisk is bold in second-last paragraph of page 8.

---

> ### Author Response · Authors · 2023-12-16
> **Response**
>
> * "One major point I tripped up on was all the talk related to convexity. The authors say that using the traditional "kernel smoothing method" (p.3), the loss is "still non-convex," and use this to motivate their "convolution-type" smoother, which they say is "not only convex but also differentiable." Maybe I'm missing something basic here, but the pinball loss $\rho_\tau(\cdot)$ is convex, correct? Is it strong convexity (i.e., a quadratic growth) the authors want? In any case, the logic here seems quite faulty. The loss is pretty nice in that it is Lipschitz, though. If smoothness (i.e., a Lipschitz gradient) is the critical point, then yes, the pinball loss here is not smooth, but no gradient-based optimizers are actually studied here, so how this technical backdrop actually links to the methods studied here is not clear to me."
>
> We appreciate the time and effort you dedicated to reviewing our paper. Your detailed and constructive feedback has been invaluable in refining our research. Yes, the pinball loss is convex, but it lacks smoothness. We do not require the strong convexity. The traditional “kernel smoothing method”, referring to Horowitz’s smoothed quantile loss, is smooth but not convex. We have attached a picture illustrating the differences among the three types of loss functions on page 2 of the newly uploaded revised manuscript.
>
> We need a smooth loss function since our proposed method applies a gradient-based algorithm to find the minimizer in the transferring and debiasing step. Our paper focuses on a high-dimensional transfer learning model. Although Horowitz’s smoothed quantile loss is smooth, it yields a non-convex function for which global minimum is not guaranteed. This poses even more challenges in the high-dimensional setting. However, the gradient-based algorithm proposed in Tan et al. (2022) is more scalable to large-scale problems with either large sample size or high dimensionality compared with other methods for fitting high-dimensional quantile regression. That's why we combine this smoothing method with our work.
>
> * "If the authors want to focus on convexity, they should be more explicit about what properties their proposed objective has, and when. Is the "empirical smoothed loss function" defined at the bottom of page 3 convex for any function $K_h(\cdot)$ that is non-negative, integrable, and symmetric about zero?"
>
> Thanks for bringing this to our attention. We are committed to addressing these concerns in our revised manuscript. In response, we have included a remark about the explicit expressions of the smoothed empirical loss with several widely used kernel functions on page 4 of our revised manuscript. Additionally we have created a plot on p.5 to visualize the convexity of those smoothed loss. I will list those changes in the following response.
>
> Recall that $\rho_\tau(u) = |u|/2+(\tau- 1/2)u$.
> 1. (Gaussian kernel) The Gaussian kernel $K(u) = \varphi(u)$, where $\varphi(\cdot)$ is the density function of a standard normal distribution. The resulting smoothed loss is $l_h(u) = (h/2)G(u/h)+(\tau -1/2)u$, where $G(u) = (2/\pi)^{1/2}e^{-u^2/2}+u(1-2\Phi(-u))$.
>
> 2. (Uniform kernel) The uniform kernel is $K(u) = (1/2) \mathbb{1}(|u|\leq 1)$, which is the density function of the uniform distribution on $[-1,1]$. The resulting smoothed loss is $l_h(u) = (h/2)U(u/h)+(\tau -1/2)u$, where $U(u) = (u^2/2+1/2)\mathbb{1}(|u| \leq 1)+|u|\mathbb{1}(|u| > 1)$ is a Huber-type loss.
>
> 3. (Laplacian kernel) The Laplacian kernel is $K(u) = e^{-|u|}/2$. We have $l_h(u) = \rho_\tau(u)+he^{-|u|/h}/2$.
>
> 4. (Logistic kernel) The logistic kernel is $K(u) = e^{-u}/(1 + e^{-u})^2$. The resulting smoothed loss is $l_h(u) = \tau u + h \log(1 + e^{-u/h})$.
>
> 5. (Triangular kernel) The triangular kernel is $K(u) = (1 - |u|)\mathbb{1}(|u| \leq 1)$. The resulting smoothed loss is $l_h(u) = (h/2)l_{tr}(u/h)+(\tau - 1/2)u$, where $l_{tr}(u) := (u^2 - |u|^3/3+1/3)\mathbb{1}(|u|\leq 1) + |u|\mathbb{1}(|u| >1)$.
>
> 6. (Epanechnikov kernel) The Epanechnikov kernel is $K(u) = (3/4)(1 - u^2)\mathbb{1}(|u| \leq 1)$. The resulting smoothed loss is $l_h(u) = (h/2)E(u/h) + (\tau - 1/2)u$, where $E(u) := (3u^2/4 - u^4/8 +3/8)\mathbb{1}(|u| \leq 1) + |u|\mathbb{1}(|u| > 1)$.
>
> All these smoothed loss are convex.

---

> > ### Author Response · Authors · 2023-12-16
> > **Response 2**
> >
> > * "Several quantities appear before they are defined. Here are a few examples.
> >      * The "contrast" $\boldsymbol{\delta}^*$ (p.1).
> >      * Matrix $\Sigma$ used in defining the ball and cone on page 2 (really, these aren't needed until much later...)."
> >
> > We have thoroughly proofread the paper to avoid some terms appearing before we define them.
> > 1. We removed the term "contrast $\boldsymbol{\delta}$" before we give the definition of this term.
> >
> > 2. We replaced $\Sigma$ with $A$ in defining the ball and cone on page 2. (Now located on page 3 in the revised manuscript.)
> >
> > * "The same notation is used for different objects in various locations. This should really be avoided. Here are some examples.
> >      * $K$ is both the number of source data sets (p.4) and the kernel function.
> >      * The notation $\hat{Q}_h(\cdot)$ is used with different meanings (e.g., p.3 and p.4). This is a critical quantity; such ambiguity severely reduces clarity and is really just sloppy exposition. It appears again on p.6 with a different meaning; the one on p.6 really should be $\hat{Q}_h^{(k)}(\cdot)$, I think this is obvious.
> >      * In Assumption 3.2, $u$ is used as a label on the upper bound $\kappa_u$ and also the variable being maxed over on the left-hand side of the inequality; more semantic inconsistency."
> >
> > Thanks for your careful evaluation of our manuscript. Below is what we have done to address your concerns.
> > 1. Instead, we use $\mathcal{K}$ to denote the number of source data sets, while $K$ specifically refers to the kernel function.
> >
> > 2. The $\hat{Q}_h(\boldsymbol{\beta})$ on p.3 (now on p.4 in the revised manuscript) is the empirical smoothed loss function in general. However the $\hat{Q}_h(\boldsymbol{\beta})$ on p.4 (now on p.5 in the revised manuscript) is the empirical smoothed loss function of the target and source data sets in the transferable set $\mathcal{A}_m$. Since there are many sub-datasets, to distinguish between them, the expression takes the form of a double sum. However, if there is only one dataset in $\mathcal{A}_m$, it will be the same as the definition of $\hat{Q}_h(\cdot)$ on p.3. The one on p.6 should be $\hat{Q}_h^{(k)}(\cdot)$. Thank you for pointing out it.
> >
> > 3. Now $\kappa_u$ is substituted by $\bar{\kappa}$. Other similar issues have also been resolved.

---

> > > ### Author Response · Authors · 2023-12-16
> > > **Response 3**
> > >
> > > * "I find it rather hard to parse the main theorems, as well as the related propositions. Here are a few things I tripped up on.
> > >    * In Prop 3.1 and 3.2, are the quantities $r, l, \boldsymbol{\beta}_1$, and $\boldsymbol{\beta}_2$ arbitrary, up to the inequalities holding? Nothing is said about them. Even more frustrating is that real-valued $l$ appears alongside $f_l$, where the subscript $l$ is just a label representing "lower" I assume (from Assumption 3.1). Again, same symbol, totally different meanings. Really must be avoided.
> > >    * What is $\alpha_1$ in Prop 3.2?
> > >    * Most critically, what is $\hat{\boldsymbol{\beta}}$ in Theorems 3.1 and 3.2? From the context, one would like to assume this corresponds precisely to the outputs of Algorithms 3 and 4 respectively; is this so? Even if this is the case, it feels like there is a conceptual gap in that the Theorem 3.2 benefits not just from the stronger "closeness" assumption used in defining $\mathcal{A}'_m$, but that it is also an "oracle" (i.e., this index of good sources is known)."
> > >
> > > The reviewer's insightful comments regarding the interpretation of our results raises our considerations, and we are committed to providing a more nuanced and accurate discussion.
> > >
> > > 1. Firstly, those quantities are not arbitrary. Since the smoothing parameters $h$ and $g$ are small, $r$ should be also small enough by the conditions in Prop 3.1 and 3.2. Consequently, the condition that $\boldsymbol{\beta}_1 - \boldsymbol{\beta}_2$ is in a ball with radius $r$ implies the distance between $\boldsymbol{\beta}_1$ and $\boldsymbol{\beta}_2$ should also be small. We use those propositions in the proof of the error bound for transfer learning in quantile regression. In the proof, $\boldsymbol{\beta}_1$ and $\boldsymbol{\beta}_2$ are very close to $\boldsymbol{\beta}^*$ and we apply those propositions when we know that $r$ is relatively small. For example, the proof of Theorem 3.1 on p.32 states:
> > >
> > >     "We could find some $t$ satisfying $\lVert t\hat{\Delta}\rVert_\Sigma\leq 1$. Denote $\tilde{\Delta} = t\hat{\Delta}$."
> > >
> > >     So we first have a small enough $r$ and then apply Proposition 3.1 to obtain a lower bound of $F(\tilde{\Delta})$, where $F(\Delta) = \hat{Q}_h(\boldsymbol{\omega}^*+\Delta) - \hat{Q}_h(\boldsymbol{\omega}^*) + \lambda_w(\lVert\boldsymbol{\omega}^*+\Delta\rVert_1 - \lVert\boldsymbol{\omega}^*\rVert_1) $.
> > >
> > > 2. $\alpha_1$ (now $\phi_1'$) is defined as $\underline{\smash{\kappa}}\cdot \underline{\smash{f}}/25$, where $\underline{\smash{\kappa}}$ is the minimum value of the kernel function and $\underline{\smash{f}}$ is the lower bound of the conditional density of the error $\epsilon$ given $\boldsymbol{x}$ when $\epsilon = 0$.
> > >
> > > 3. The $\hat{\boldsymbol{\beta}}$ in Theorem 3.1 is the output of Algorithm 1 and the $\hat{\boldsymbol{\beta}}$ in Theorem 3.2 is the output of Algorithm 4. Both theorems are based on the oracle algorithms of the transfer learning quantile regression. They demonstrate a fast rate of convergence when the transferable set is non-empty, in which case the information from the useful source can be optimally transferred to substantially help solve the regression problem under the target model.
> > >
> > >     For the algorithm with an $\ell_1$-norm constrained transferring set, we proposed a transferable source detection algorithm to identify informative sources from all available sources. We then evaluate the empirical performance of our proposals. The results indicate that source detection algorithm works well and demonstrate the effectiveness of Algorithm 3.
> > >
> > >     In the case of the algorithm with an $\ell_0$-norm constrained transferring set, we do not have a corresponding source detection algorithm, so we only test the performance of Algorithm 4 with an increasing number of transferable source data sets in the simulations.
> > >
> > >     Theorem 3.2 is an extension to the settings where the contrast vectors are characterized in terms of the $\ell_0$-norm. However, the theorem does not benefit from the stronger "closeness" assumption since the $\ell_1$ norm of the contrasts could be large under this assumption. It also increase the difficulty of the implementation to some degree. The algorithm estimates the contrast vectors individually. This is because the $\ell_1$-norm has the sub-additive property, while $\ell_0$-norm does not.

---

> > > > ### Author Response · Authors · 2023-12-16
> > > > **Response 4**
> > > >
> > > > * "Early in the paper, the authors compare their work with existing papers at a detailed technical level, but I think this could use some additional exposition. Here are two points:
> > > >
> > > >     * First, in comparing the $s^2$ versus $s^3$ factors (paragraph 2, page 2), the authors say that "this is due to the bandwidth involved in the smoothing method." What does this mean? The main results in this paper assume the bandwidth is optimally set, correct? Naively, I think some readers will assume that the difference is due to the fact that bias is introduced by using the smoothed objective instead of the traditional quadratic regression objective; is this not the case? One expects that there is a tradeoff - a benefit due to smoothness, and a takeaway due to bias - and that these are balanced in the analysis, but this is unclear at present.
> > > >
> > > >     * Also on page 2, the authors say that compared with Zhang and Zhu 2022, they do not require a particular "restrictive condition" on the kernel function; why is this? What differs here? It isn't just due to the stronger $\ell_0$-norm contrast assumptions, I assume?"
> > > >
> > > > We are grateful for your thoughtful suggestions and valuable insights. For the comparison between the $s^2$ versus $s^3$ factors, this part has been omitted in the revised manuscript. We modified the proof, and now it only requires the sample size of the target data to be $\mathcal{O}(s^2\log p)$, aligning with the results in Li et al. (2022) and Tian \& Feng (2022).
> > > >
> > > > There is a takeaway due to bias. In the proof,
> > > > $$
> > > > \big(\nabla \hat{Q}_h(\hat{\boldsymbol{\omega}}^{\mathcal{A}_m}) - \nabla \hat{Q}_h(\boldsymbol{\omega}^*)\big)^\mathrm{T}\hat{\Delta} \leq \lambda_w \Big(\lVert \hat{\Delta}\_\mathcal{S} \rVert_1 - \lVert \hat{\Delta}\_{\mathcal{S}^c} \rVert_1 + 2\lVert\boldsymbol{\omega}^*\_{\mathcal{S}^c}\rVert_1\Big) + \frac{\lambda_w}{2} \lVert \hat{\Delta} \rVert_1 + b_h^*\lVert \hat{\Delta}\rVert\_{\Sigma}.
> > > > $$
> > > >
> > > > The term $b_h^*\lVert \hat{\Delta}\rVert_\Sigma$ represents the bias introduced by incorporating the convolution smooth.
> > > >
> > > > Regarding how our paper differs from Zhang \& Zhu (2022), firstly, Zhang \& Zhu (2022) is a concurrent work that also considers the smoothed quantile regression models under a transfer learning framework with transferable source defined in $\ell_1$-norm. We have noticed that their paper has two versions. In the first version, they introduced a condition on the kernel function $K(\cdot)$, which is $\sup_{|h|\leq 1}K(u/h)/h<M_k$ almost everywhere (in $u$). However, contraty to the paper's claim, this inequality does not hold for many kernel functions. Even the Gaussian kernel, Laplacian kernel and Epanechnikov kernel mentioned in their paper do not satisfy this condition.
> > > >
> > > > In the newest version, they removed that condition. However, their paper still requires a condition that $\lVert\boldsymbol{x}^{(k)}\rVert_\infty\leq B_k$. Our paper relaxes this condition, only requiring the covariate vector $\boldsymbol{x}^{(k)}$ to be sub-exponential. The sub-exponential tail condition is commonly assumed in high-dimensional statistics. In light of these distinctions, our approach provides a more flexible and widely applicable framework for smoothed quantile regression models under transfer learning.
> > > >
> > > > References:
> > > >     [1]. Horowitz, J.L., 1998. Bootstrap methods for median regression models. Econometrica, pp.1327-1351.
> > > >     [2]. Tan, K.M., Wang, L. and Zhou, W.X., 2022. High-dimensional quantile regression: Convolution smoothing and concave regularization. Journal of the Royal Statistical Society Series B: Statistical Methodology, 84(1), pp.205-233.
> > > >     [3]. Zhang, Y. and Zhu, Z., 2022. Transfer Learning for High-dimensional Quantile Regression via Convolution Smoothing. arXiv preprint arXiv:2212.00428.
> > > >     [4] Li, S., Cai, T.T. and Li, H., 2022. Transfer learning for high-dimensional linear regression: Prediction, estimation and minimax optimality. Journal of the Royal Statistical Society Series B: Statistical Methodology, 84(1), pp.149-173.
> > > >     [5] Tian, Y. and Feng, Y., 2022. Transfer learning under high-dimensional generalized linear models. Journal of the American Statistical Association, pp.1-14.

---

> > > > > ### Comment · Reviewer_JxJL · 2024-01-05
> > > > > **Re: Author response**
> > > > >
> > > > > I thank the authors for their detailed response, and have confirmed the revisions and improvements made to the paper.

---

### Review · Reviewer_yBAM · 2023-11-22

**Summary Of Contributions:**

The paper introduces an algorithm aimed at enhancing quantile regression through transfer learning. Building upon the optimization method for quantile regression, the authors propose a pipeline of leveraging similar datasets to benefit from utilizing extensive data. Dataset similarity is gauged by a straightforward criterion, employing the q-norm of the deviation between optimal parameters with q = 0 or 1.

The proposed pipeline includes the transfering, debiasing, and determining the transferable source. While the algorithm itself proposes a standard modules applicable to various algorithms beyond quantile regression, the accompanying theories are tailored specifically to quantile regression.

**Audience:**

Yes

**Broader Impact Concerns:**

Nothing to note.

**Claims And Evidence:**

Yes

**Requested Changes:**

I suggest minor changes:

1. Use a consistent inner product notation: use either bra-ket or transpose.

2. Include reference for negative transfer on the 7th line in the Introduction.

3. In the last equation of the Introduction (in page 3), $\Sigma$ should be $A$ for consistency of the notations.

**Strengths And Weaknesses:**

The method of determining the transferable set, detecting the deviation of parameters, and the joint optimization step looks reasonable and novel. The authors provided experimental results only for synthetic data. It would be nice if some experiments for real data have been provided, but it should be fine because the graphs show the algorithms are working as expected.

---

> ### Author Response · Authors · 2023-12-16
> **Response**
>
> Thanks for recognizing the strengths of our work. We also appreciate your constructive comments, which has pinpointed areas for improvement. We are committed to addressing these weaknesses and refining our paper accordingly.
>
> * "1. Use a consistent inner product notation: use either bracket or transpose."
>
> We have changed all the inner product notation to the bracket form in the revised manuscript.
>
> * "2. Include reference for negative transfer on the 7th line in the Introduction."
>
> We have added the reference for negative transfer in the Introduction.
>
> * "3. In the last equation of the Introduction (in page 3), $\Sigma$ should be $A$ for consistency of the notations."
>
> We replaced $\Sigma$ with $A$ in defining the ball and cone on page 2. (Now located on page 3 in the revised manuscript.)

---

### Review · Reviewer_CHYg · 2023-12-02

**Summary Of Contributions:**

Paper contributions are as follows :

1. Proposes transfer learning algorithms for high-dimensional quantile regression using convolution-type smoothing. Assumes sparsity on the contrast between target and source coefficients and shows the estimation error bounds can improve over using only the target data.

2. Adapts the transferable source detection algorithm to the quantile regression setting to identify useful sources when the informative sources are unknown. Provides theoretical analysis and simulation studies to validate the performance improvements.

**Audience:**

Yes

**Claims And Evidence:**

Yes

**Requested Changes:**

Provide some analysis or empirical results on sensitivity to key assumptions like similarity of source and target distributions. Relaxing assumptions even mildly would broaden applicability.

**Strengths And Weaknesses:**

Strengths

1) The proposed algorithms are intuitive and seem practical, leveraging smoothing techniques to handle non-smooth loss and using sparsity assumptions.
2) The theoretical analysis providing error bounds under mild conditions is solid. The bounds quantitatively show potential improvements over not using transfer learning.

Weakness:

The relaxation of not needing the condition that $sup|_h|≤1 K(u/h)/h < M_k$ seems like incremental relaxation over previous works  Zhang & Zhu (2022)

Refs:

[1] Transfer Learning for High-dimensional Quantile Regression via Convolution Smoothing

---

> ### Author Response · Authors · 2023-12-16
> **Response**
>
> * "Provide some analysis or empirical results on sensitivity to key assumptions like similarity of source and target distributions. Relaxing assumptions even mildly would broaden applicability."
>
> We are thankful to the reviewer for the positive feedback on our work. The constructive criticism has been particularly helpful in identifying the weaknesses of our paper, and we are committed to addressing these concerns to enhance the overall quality of our research.
>
> We added empirical results on the similarity of source and target distributions in the Appendix C on page 42. For more details, please refer to the graphs and explanations in the appendix. The results demonstrate that when the contrast is relatively small either in $\ell_1$ or $\ell_0$ norm, most of the target and source data are overlapped. When the contrast is relatively large, the source data would have much more frequency at the two tails, which may cause the negative transfer if those sources are used in transfer learning algorithm.

---

### Author Response · Authors · 2023-12-16
**General Response**

We express gratitude to all the reviewers for your thoughtful comments and apologize for the delayed response. The paper has been revised to incorporate requested changes. In summary:

* Corrected the use of the same notations in different objects and avoided the occurrence of terms before their definitions. For example, $f_u, \kappa_u$ have been replaced by $\bar{f}, \bar{\kappa}$, and $f_l, \kappa_l$ have been replaced by $\underline{\smash{f}}, \underline{\kappa}$.

* Included empirical results on the similarity of source and target distributions in the Appendix C.

* Revised the proof to establish that the theory of the algorithm with an $\ell_1$-norm transferable set now relies on the sub-exponential assumption on the covariate vector $\boldsymbol{x}$, eliminating the previous requirement of $\lVert \boldsymbol{x} \rVert_\infty \leq B$. This distinction is important and addresses a concern raised by the reviewers, distinguishing our paper from Zhang and Zhu (2022) in this particular aspect.

* Added the reference for negative transfer in the Introduction.

* Included a remark about the explicit expressions of the smoothed empirical loss with several widely used kernel functions and added the plots to better illustrate the convolution smoothing mechanism behind those smoothed empirical losses.

Thank you again for your valuable feedback, and please let us know if you have any additional comments or suggestions.

---

### Decision · Action_Editor_72Ky · 2024-01-10

**Recommendation:** Accept as is

**Comment:**

The authors have put an honest effort into revising the paper to address the concerns raised by reviewers, most of which are related to the presentation. The presentation issues have made it difficult to verify all of the claims in the paper, although no major issues related to correctness of the results have been raised by reviewers.

After revision to improve the presentation, all three reviewers are in favour of accepting the paper.

**Audience:**

Transfer learning is clearly relevant to the TMLR audience.

**Claims And Evidence:**

The main concern raised is the close connection with the paper by Zhang & Zhu (2022), compared to which the present work may be considered incremental. According to the acceptance criteria for TMLR, this is not a reason to reject as long as the claims are sufficiently supported. With this in mind, I recommend acceptance.

---

> ### Author Response · Authors · 2024-01-29
> **Camera Ready Uploaded**
>
> Dear Editor and all Reviewers,
>
> We appreciate your feedback and the recommendation for acceptance. The camera ready version has been uploaded.
>
> Thank you.